# Human and bacterial genetic variation shape oral microbiomes and health

Nolan Kamitaki[1,2,3,4,5,6 ✉], Robert E. Handsaker[3,4,5], Margaux L. A. Hujoel[1,2,3,7,8], Ronen E. Mukamel[1,2,3], Christina L. Usher[9], Steven A. McCarroll[3,4,5,10 ✉] & Po-Ru Loh[1,2,3 ✉]

Human genetic variation influences all aspects of our biology, including the oral cavity[1–3], through which nutrients and microbes enter the body. Yet it is largely unknown which human genetic variants shape a person's oral microbiome and potentially promote its dysbiosis[3–5]. We characterized the oral microbiomes of 12,519 people by re-analysing whole-genome sequencing reads from previously sequenced saliva-derived DNA. Human genetic variation at 11 loci (10 new) associated with variation in oral microbiome composition. Several of these related to carbohydrate availability; the strongest association ($P = 3.0 \times 10^{-188}$) involved the common *FUT2* W154X loss-of-function variant, which associated with the abundances of 58 bacterial species. Human host genetics also seemed to powerfully shape genetic variation in oral bacterial species: these 11 host genetic variants also associated with variation of gene dosages in 68 regions of bacterial genomes. Common, multi-allelic copy number variation of *AMY1*, which encodes salivary amylase, associated with oral microbiome composition ($P = 1.5 \times 10^{-53}$) and with dentures use in UK Biobank ($P = 5.9 \times 10^{-35}$, $n = 418,039$) but not with body mass index ($P = 0.85$), suggesting that salivary amylase abundance impacts health by influencing the oral microbiome. Two other microbiome composition-associated loci, *FUT2* and *PITX1*, also significantly associated with dentures risk, collectively nominating numerous host–microbial interactions that contribute to tooth decay.

When Antonie van Leeuwenhoek first observed bacteria as 'animalcules' in scrapings from his teeth in the seventeenth century, one of his first inquiries involved the extent of their variation among people[6]. Oral microbiomes are now known to vary abundantly across people[7–9], and twin studies have shown that some of this variation is heritable[1–3]. However, few human genetic polymorphisms have been associated with the abundances of specific oral microbial species[3–5]; study sizes so far ($n < 3,000$) have provided limited power to detect robust genetic effects. Larger genome-wide association studies (GWAS) of the gut microbiome ($n = 5,959–18,340$) have consistently replicated two effects of variation at the *LCT* and *ABO* loci on gut microbial abundances[10–13], and larger GWAS of oral microbiomes might yield similar discovery.

Oral pathologies, such as dental caries, result from dysbiosis of the oral microbiome[14]. Untreated pathologies can progress to oral infections which carried high mortality rates before modern dentistry and antibiotics[15]. Susceptibility to caries and other oral pathologies is also strongly influenced by genetics[16,17], and GWAS have identified 47 loci harbouring such genetic effects[18]. However, whether these or other genetic effects act by modulating the composition of the oral microbiome is at present unknown. Identifying such interactions could point to microbial drivers of cariogenesis[9].

Given the effects of human hosts and resident microbes on each other's survival and evolutionary trajectory, the human microbiome is an example of symbiosis[19,20]. The stability of the gut microbiome in individuals[21], its codiversification with humans[22] and abundant structural variation of its microbial genomes[23] all suggest intricate genetic interactions between microbiomes and their human hosts, whereby microbial genomes adapt to genetic variation across people. A recently observed example of such an interaction with the gut microbiome is a structural variant in the *Faecalibacterium prausnitzii* genome that includes genes encoding an *N*-acetylgalactosamine (GalNAc)-metabolizing pathway and interacts with human *ABO* variation[24]. Whether such specific co-adaptation commonly occurs in oral microbiomes remains an open question.

## Oral microbiome profiles of 12,519 people

To create a dataset suitable for exploring variation in the oral microbiome and the way it is shaped by human genetic variation, we analysed DNA sequencing reads previously generated from whole-genome sequencing (WGS) of saliva samples from 12,519 participants in the Simons Foundation Powering Autism Research (SPARK) cohort[25]

[1]Division of Genetics, Department of Medicine, Brigham and Women's Hospital and Harvard Medical School, Boston, MA, USA. [2]Center for Data Sciences, Brigham and Women's Hospital, Boston, MA, USA. [3]Program in Medical and Population Genetics, Broad Institute of MIT and Harvard, Cambridge, MA, USA. [4]Stanley Center for Psychiatric Research, Broad Institute of MIT and Harvard, Cambridge, MA, USA. [5]Department of Genetics, Harvard Medical School, Boston, MA, USA. [6]Department of Biomedical Informatics, Harvard Medical School, Boston, MA, USA. [7]Department of Human Genetics, David Geffen School of Medicine, University of California, Los Angeles, Los Angeles, CA, USA. [8]Department of Computational Medicine, David Geffen School of Medicine, University of California, Los Angeles, Los Angeles, CA, USA. [9]Jerome Lipper Multiple Myeloma Center, Dana-Farber Cancer Institute, Boston, MA, USA. [10]Howard Hughes Medical Institute, Harvard Medical School, Boston, MA, USA. ✉e-mail: nolan_kamitaki@hms.harvard.edu; smccarro@broadinstitute.org; poruloh@broadinstitute.org

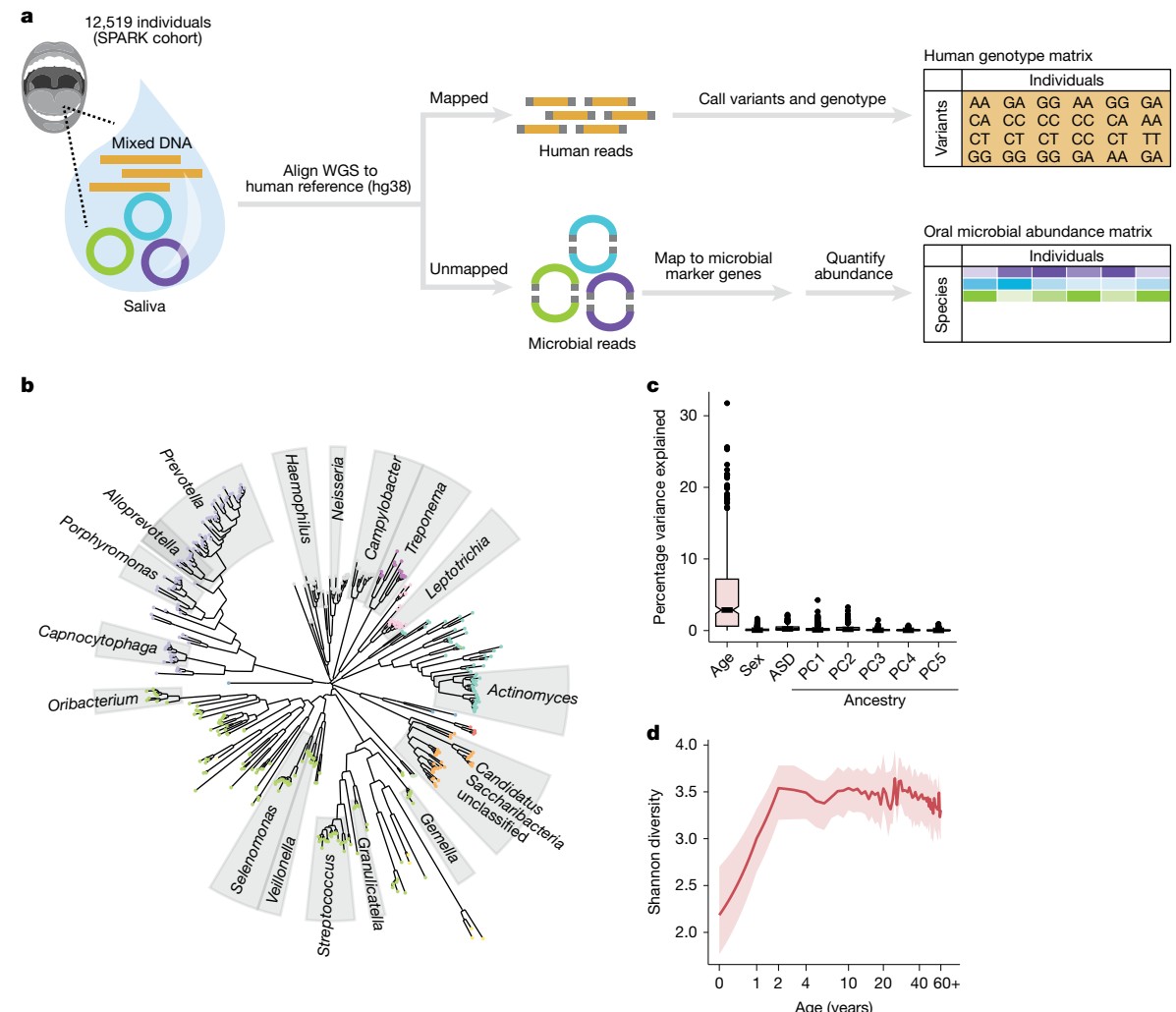

**Fig. 1 | Oral microbiomes in 12,519 individuals measured by WGS of saliva samples. a**, Generation of paired datasets of human genetic variation and oral microbiome composition from WGS of saliva samples from the SPARK cohort ($n = 12{,}519$). Human genetic variants were previously called with DeepVariant and relative abundances of microbial species were estimated with MetaPhlAn 4 (ref. 29) from sequencing reads that did not map to the human genome. **b**, Phylogenetic tree based on genomic divergence among 439 microbial species observed in ≥10% of SPARK participants. Phyla are indicated by dot colour and genera with more than five species are indicated with labelled grey sectors. **c**, Contributions of age, sex, ASD case status and genetic ancestry principal components (PC1 through PC5) to variation in oral microbial species abundances.

For each factor, the fraction of variance in species abundance explained by the factor was computed for each of the 439 species, and the box and whisker plot shows the distribution of this quantity across the 439 species. ASD status explained a median fraction of variance of 0.002. Boxes span quartiles; centres indicate medians and whiskers are drawn up to 1.5× the interquartile range. **d**, Species diversity in the oral microbiome as a function of host age. The red line indicates median Shannon entropy and the shaded region indicates the interquartile range. Oral microbial diversity increases substantially over the first few years of life, plateaus and then modestly declines in late adulthood. Images in **a** were reproduced from Pixabay (https://pixabay.com) under a CC0 1.0 Universal Public Domain Licence.

(Fig. 1a), building on previous work[26,27]. WGS captured substantial non-human genomic information[28], with a median of 8.4% ([4.6%,14.7%], quartiles) of sequencing reads not mapping to the human reference genome (Extended Data Fig. 1a). Many of these unmapped reads instead mapped to clade-specific marker genes in microbial genomes[29], enabling quantification of relative microbial abundances. This produced the largest collection of oral microbiome profiles ($n = 12{,}519$) generated so far, measuring the abundances of 645 microbial species present at >1% frequency, including 439 species (spanning 13 phyla, including one fungal commensal, *Malassezia restricta*) commonly observed in SPARK (≥10% of participants) (Fig. 1b, Extended Data Fig. 1b and Supplementary Table 1). Comparing these profiles across individuals showed that age was a major driver of interindividual variation in oral microbiome composition, unlike autism spectrum disorder (ASD) case status, sex and genetic ancestry (Fig. 1c and Extended Data Fig. 1c–g). Across the lifespan represented in SPARK (age 0–90 years), mean species diversity

sharply increased in the first few years of life (representing when the oral cavity is colonized, diet diversifies and primary teeth are acquired) and then decreased slowly with age[8] (Fig. 1d). Individual species exhibited vastly different abundance trajectories over the lifespan, with some observed predominantly in adults and others predominantly in children (Extended Data Fig. 1h–k).

## Human genetics shapes oral microbiome composition

To identify human genetic variants that influence interindividual differences in the abundances of microbial taxa, we first tested the abundances of taxa detectable in ≥10% of participants for association with common human genetic variants, accounting for family structure using a linear mixed model[30,31]. Human genetic variants at seven loci associated with the abundance of at least one taxon at study-wide significance ($P < 4.0 \times 10^{-11}$; Extended Data Fig. 2a), with only one locus

(*SLC2A9*) previously identified[4]. As several loci associated with the abundances of many species (Supplementary Tables 2 and 3) and none associated with $\alpha$-diversity (Extended Data Fig. 2b), we developed a statistical test to capture pleiotropic effects on many species in an interdependent microbial community[32–34], using principal component analysis (PCA) to enable efficient genome-wide association testing (Fig. 2a and Methods). Similar to a recent approach for GWAS on high-dimensional cell state phenotypes in single-cell RNA-seq data[35], this approach also reduces multiple-testing burden by testing each genetic variant only once.

Applying this approach to SPARK identified four additional human genomic loci (11 total) at which common genetic variation associated with oral microbiome composition ($P < 5 \times 10^{-8}$; Fig. 2b and Extended Data Table 1). The principal component (PC)-based test was well-calibrated (Extended Data Fig. 2c) and top signals were confirmed by multivariate distance matrix regression[33,36] (Extended Data Fig. 2d,e). The association signals tended to distribute across many microbial PCs (mPCs; Extended Data Fig. 2f–p), suggesting that human genetic variants subtly influence many axes of microbial community coordination. Among the ten new loci, eight implicated genes—and in several cases, specific variants—with readily interpretable functions that could explain their associations with microbiome composition.

- Three loci contained genes encoding highly expressed salivary proteins: salivary amylase (encoded by *AMY1*; $P = 1.5 \times 10^{-53}$, top association), submaxillary gland androgen-regulated proteins (*SMR3A* and *SMR3B*; $P = 1.4 \times 10^{-12}$) and basic salivary proline-rich proteins (*PRB1–PRB4*; $P = 1.1 \times 10^{-11}$). These associations seemed to be driven mainly by genetic variants that modify gene expression or copy number (Extended Data Table 1, Extended Data Fig. 3 and Supplementary Note 1). Consistent with these results, heritability-partitioning analysis[37] indicated that genetic effects on oral microbiome composition are enriched at genes specifically expressed in salivary glands ($P = 0.02$, Extended Data Fig. 4a).
- Two loci contained genes with established roles in immune function: the HLA class II genes, which encode proteins that present peptides in adaptive immunity, and *TLR1*, encoding Toll-like receptor 1, that binds bacterial lipoproteins in innate immunity. The strongest association at *TLR1* involved a missense variant (rs5743618; $P = 6.2 \times 10^{-18}$) that produces the I602S substitution known to inhibit trafficking of TLR1 to the cell surface, reducing immune response in a recessive manner[38,39]. Consistent with these reports, I602S associated recessively with microbial abundances ($P = 6.7 \times 10^{-29}$; Extended Data Fig. 4b).
- Two other loci, *ABO* and *FUT2*, encode glycosyltransferases that together determine expression of histo-blood group antigens on epithelial cells and secreted proteins (in addition to the well-known role of *ABO* in determining blood type). This broader role is important to microbial species that interact with mucosal surfaces, such that both loci are known to influence the gut microbiome[10–13], with some bacterial species using A-antigen saccharides as a carbohydrate source[24]. The variants at *ABO* and *FUT2* that associated most strongly with oral microbiome composition were rs2519093 ($P = 9.5 \times 10^{-15}$), which tags the A1 blood group[40], and rs601338 ($P = 1.6 \times 10^{-131}$ additively, $P = 3.0 \times 10^{-188}$ recessively), the common *FUT2* W154X nonsense variant that (in homozygotes) produces the non-secretor phenotype in which bodily fluids lack histo-blood group antigens[41].
- Associations of variants at *PITX1* with oral microbiome composition colocalized with previously reported associations of these variants with dental caries and dentures use ($r^2 = 0.99$ between the top microbiome-associated variant (rs3749751; $P = 3.0 \times 10^{-11}$) and the top dentures-associated single nucleotide polymorphism (SNP) at *PITX1*; Extended Data Fig. 4c; ref. 18). *PITX1* is a developmentally expressed gene which seems to have a role in mandibular tooth morphogenesis (based on a knockout mouse model)[42], suggesting that common genetic variation at *PITX1* might influence tooth morphology and through it, oral microbiota and dental health.

The shared associations of genetic variants at *PITX1* with both oral microbiome composition and dental health phenotypes suggested that other genetic influences on the oral microbiome might similarly influence dental health. To explore this, we performed GWAS of dentures use (a proxy for tooth loss and caries) in the UK Biobank (UKB) cohort ($n = 75,156$ cases, $n = 342,883$ controls)[43]. Three loci—*AMY1*, *FUT2* and *PITX1*—contained variants that associated ($P < 5 \times 10^{-8}$) with both oral microbiome composition and dentures use, and at each of these loci, the association patterns colocalized (Fig. 2b and Extended Data Fig. 4c,d). Moreover, at 8 of the 11 loci influencing oral microbiome composition, the most strongly associated variant also exhibited at least a nominal association ($P < 0.05$) with dentures risk (Extended Data Table 1), suggesting that host genetic effects on oral microbiome composition often have downstream effects on oral health.

Most of these genetic associations seemed to involve effects of human genetic variation on the abundances of several bacterial species, with 167 species–genotype pairs reaching FDR < 0.05 across the 11 loci (Supplementary Table 3). These associations were not driven by compositional effects or by ASD status (Extended Data Fig. 4e,f). The strong associations at *AMY1* and *FUT2* offered an opportunity for detailed investigation of how genetic variation at these loci influences oral microbiomes and oral health. *FUT2* W154X associated with the abundances of 58 of the 439 species (Fig. 2c–e). *FUT2* seemed to be nearly but not completely haplosufficient in these associations, with slightly weaker abundance-modifying effects observed among secretor individuals with a heterozygous W154X genotype compared to those with two wild-type alleles (Fig. 2d and Extended Data Fig. 4g). For several pairs of closely related species, *FUT2* W154X associated with increased abundance of one species and decreased abundance of the other (Fig. 2e and Extended Data Fig. 5), possibly reflecting competition between closely related species for ecological niches.

## Effects of complex variation at the amylase locus

*AMY1* encodes salivary $\alpha$-amylase, an enzyme that breaks down dietary starches into simple sugars. The dramatic copy number expansion of the amylase locus in humans and other animals[44] has attracted much interest for its theorized role in facilitating recent adaptation to starch-based diets[45–47], but its reported association with human body mass index (BMI)[48] and type 2 diabetes[49] has been controversial[50,51]. *AMY1* copy number genotypes in SPARK and UKB (estimated from WGS depth-of-coverage) showed extensive polymorphism[45,50] (2–32 copies per individual; Fig. 3a and Extended Data Fig. 6a) and high mutability ($6.3 \times 10^{-4}$ ($3.7 \times 10^{-4}$–$11 \times 10^{-4}$, 95% confidence interval (CI)) mutations per haplotype per generation, similar to an estimate using coalescent modelling[46]; Extended Data Fig. 6b).

Copy number variation of *AMY1* generated the strongest association of genetic variation at the amylase locus with oral microbiome composition ($P = 1.5 \times 10^{-53}$; Fig. 2b) and associated with the abundances of 42 bacterial species (FDR < 0.05, 22 species at FDR < 0.01 with $P = 5.1 \times 10^{-25}$ to $P = 0.00047$; Fig. 3b). The abundances of these species changed stepwise with *AMY1* copy number, generating a long allelic series with steadily increasing or decreasing abundances (Fig. 3c), congruent with the effect of *AMY1* copy number on the abundance of secreted salivary amylase[45,52]. Two of these associations seemed to confirm associations previously observed in smaller candidate gene studies[49,53] (Supplementary Note 2).

*AMY1* copy number also associated strongly with dentures use in UKB ($P = 5.9 \times 10^{-35}$, surpassed only by the *PITX1* locus; Fig. 2b). Each additional copy of *AMY1* associated with a 2.1% (1.7%–2.4%) increase in the odds of having dentures, corresponding to a 1.4-fold range in odds across people with 2–16 *AMY1* copies (Fig. 3d). This association replicated in the All of Us (AoU) cohort[54] ($n = 230,002$; $P = 3.5 \times 10^{-4}$ for tooth loss, $P = 6.1 \times 10^{-3}$ for caries; Extended Data Fig. 6c–e). Surprisingly, *AMY1* copy number associated with decreased risk of bleeding gums in

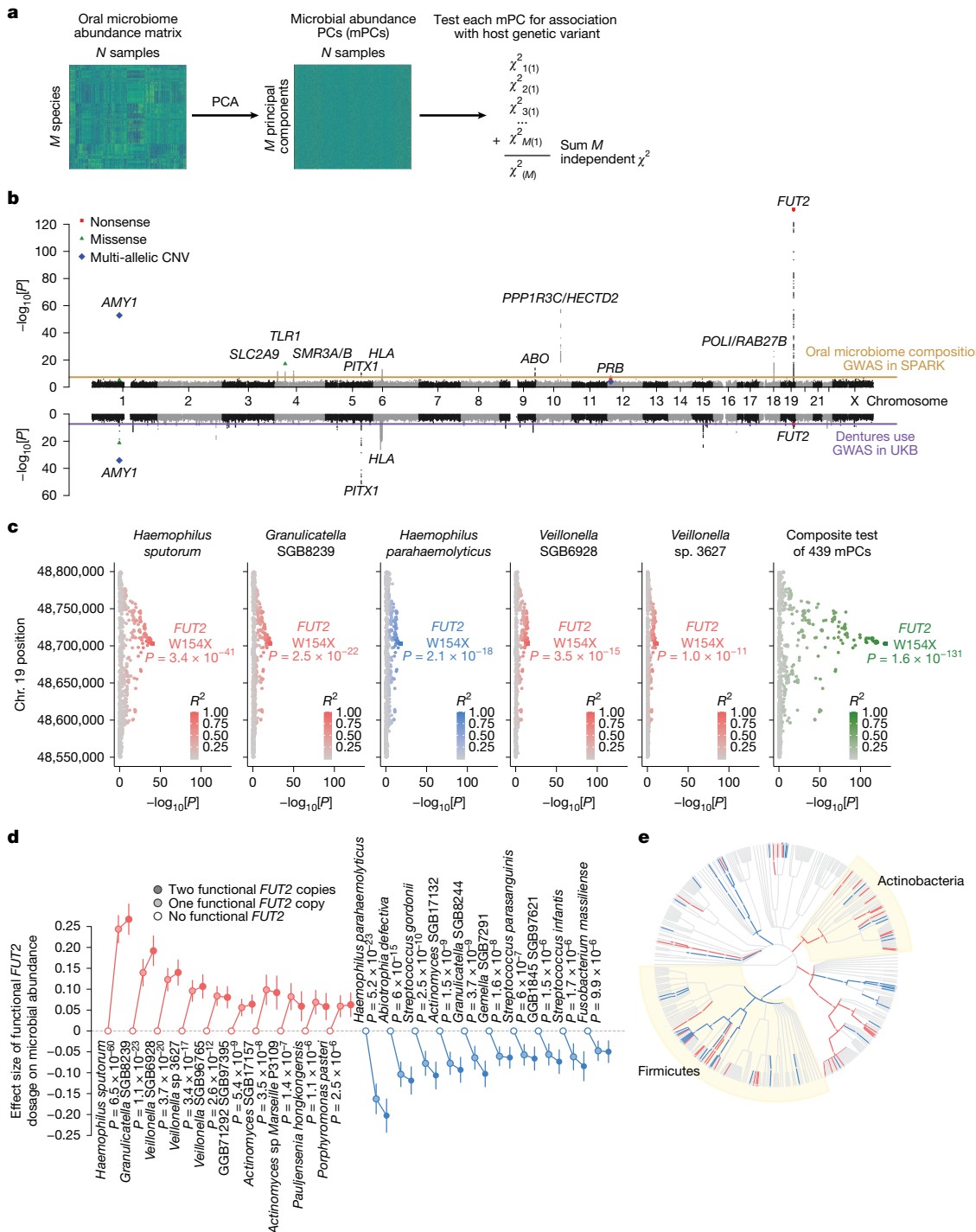

**Fig. 2 | Associations of host genetic variants with oral microbiome composition overlap risk loci for dentures use. a**, Converting relative abundances of *M* microbial species (left) into *M* orthogonal PCs (middle) allows combining chi-squared statistics for a given genetic variant (one per PC) into a single chi-squared test statistic with *M* degrees of freedom (right). **b**, Genome-wide associations with oral microbiome composition in SPARK (top, *n* = 12,519) and dentures use in UKB (bottom, *n* = 418,039). Nonsense (red squares), missense (green triangles) and multi-allelic copy number variants (CNVs) (blue diamonds) are highlighted. **c**, Associations of variants at the *FUT2* locus with relative species abundance for the five microbial species with the strongest associations (left five plots); colour indicates effect direction (plots with red points correspond to species which are more abundant in people with functional *FUT2* (that is, secretors); blue, less abundant) and colour saturation indicates linkage disequilibrium with rs601338 (*FUT2* W154X). Association strengths from the combined test for association with oral microbiome composition show

much greater statistical power (rightmost plot). **d**, Effect sizes (in s.d. units) on relative abundance of microbial species for individuals heterozygous for functional *FUT2* (light-filled circles) and for homozygotes (dark-filled circles) relative to those with no functional *FUT2* (empty circles). For each effect direction, the ten most significantly associated species are shown. *P* values are from a recessive model of *FUT2* W154X genotype. Error bars, 95% CIs. **e**, Microbial taxa whose abundance associated with *FUT2* genotype (FDR < 0.1) shown on the phylogenetic tree of 439 species (red, taxa whose relative abundances increased with functional *FUT2*; blue, decreased). Two significantly associated phyla (Firmicutes and Actinobacteria; *P* = 1.2 × 10$^{-4}$ and 4.0 × 10$^{-5}$, respectively) are highlighted with yellow sectors. At the species level (outermost circle), dot sizes increase with statistical significance. *P* values were computed using one-sided chi-squared test (top half, **b**), two-sided linear regression (bottom half, **b**) or two-sided linear mixed models (**c**, **d**).

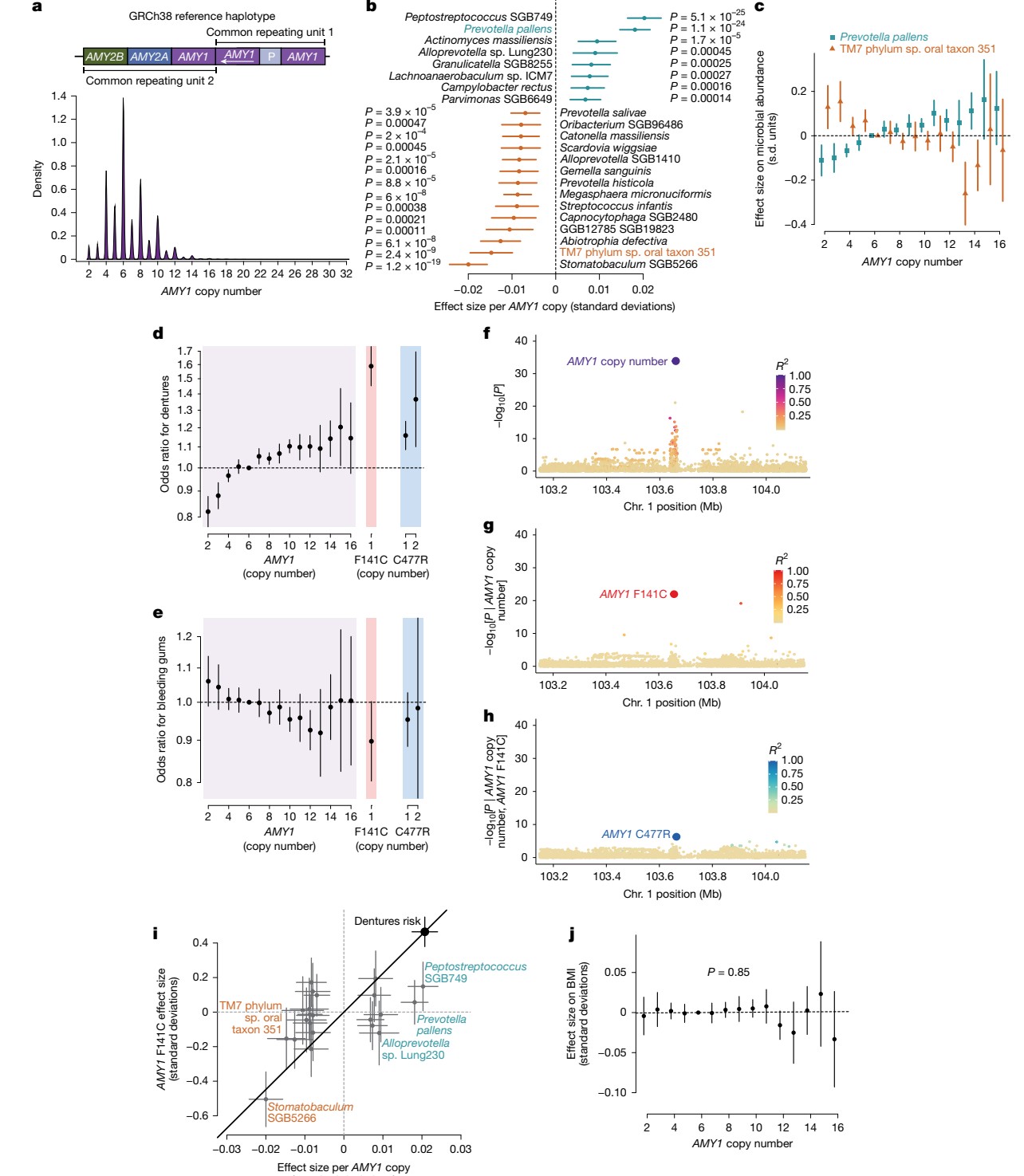

**Fig. 3 | Complex genetic variation of the salivary amylase locus affects the abundance of several oral microbial species and oral health. a**, Distribution of *AMY1* diploid copy number estimates for UKB participants (*n* = 490,415). Inset, diagram of the amylase locus in the human reference genome with common variable cassettes[46,47]. **b**, Effect sizes on relative species abundance for the 16 species most strongly associated with host *AMY1* copy number (FDR < 0.01). **c**, Allelic series of effect sizes of *AMY1* copy number genotypes on normalized abundances of *Prevotella pallens* and TM7 phylum sp. oral taxon 351 (*n* = 12,487). **d**, Odds ratios for risk of dentures use in UKB (*n* = 418,039) across copy number genotypes of *AMY1* (purple), *AMY1* F141C (red) and *AMY1* C477R (blue). **e**, Odds ratios for risk of bleeding gums in UKB (*n* = 418,039). **f**, Associations of variants at the amylase locus with dentures use. Plotted variants include paralogous sequence variants (PSVs) in the *AMY1* region (for which copy numbers of minor

alleles were tested for association). Dot colours indicate linkage disequilibrium (LD) with *AMY1* copy number. **g**, Associations with dentures use conditioned on *AMY1* copy number. **h**, Associations with dentures use additionally conditioned on *AMY1* F141C copy number. **i**, Comparison of effect sizes for *AMY1* copy number versus *AMY1* F141C copy number on relative abundances of 16 microbial species (from **b**, *n* = 12,519) and on risk of dentures use (large black dot, *n* = 418,039). For some species, the relative effect size of *AMY1* copy number versus *AMY1* F141C copy number on abundance differs significantly from this ratio for dentures use (black line). **j**, Effect sizes of *AMY1* copy number genotypes on BMI in UKB (*n* = 418,150). The line drawn is the best fit across *AMY1* copy numbers. Error bars, 95% CIs in all panels. *P* values were computed using two-sided linear mixed models (**b**) and linear regression (**f**–**h**,**j**).

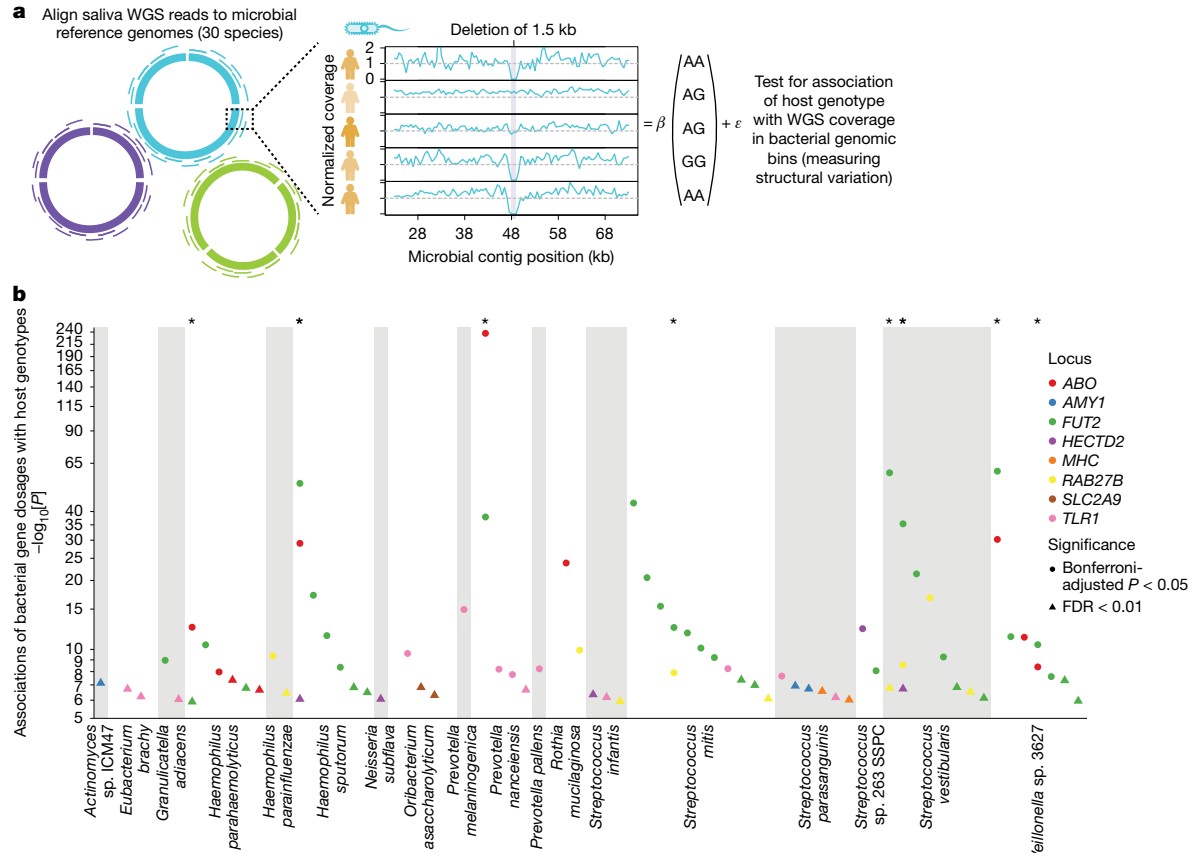

**Fig. 4 | Numerous deletions in oral microbial genomes are selected for by human genetic variation. a**, Approach to identify deletions in microbial reference genomes that associate with 11 human genetic variants that influence oral microbiome composition (Extended Data Table 1). Sequencing reads were remapped to 30 microbial reference genomes, after which normalized WGS coverage was computed across 500 bp genomic bins and truncated to a maximum of 1. **b**, Associations between microbial gene dosage (based on normalized WGS coverage) and human genetic variants. Each dot indicates a microbial genomic region that associated with at least one human genetic variant (circles, Bonferroni-adjusted $P < 0.05$; triangles, FDR < 0.01). Dot colours indicate human loci and regions that associated with more than one human locus are marked with an asterisk (top). $P$ values were computed using two-sided linear regression (**b**).

UKB ($P = 1.5 \times 10^{-6}$; Fig. 3e), even though gingivitis is considered a risk factor for tooth loss[55,56]. However, the bleeding gums and dentures use phenotypes had little genetic overlap[18] and were slightly negatively correlated ($r = -0.07$, s.e. 0.0015), suggesting largely independent pathology. These associations were specific to *AMY1*; the copy number of *AMY2A* and *AMY2B* (encoding pancreatic amylase) did not associate with dental phenotypes (Extended Data Fig. 6f,g).

Beyond the effect of *AMY1* copy number, two missense variants in *AMY1* carried by 1%–2% of UKB participants seemed to confer the largest increases in dentures risk of all common variants in the human genome (OR = 1.59 (1.46–1.73) per copy of *AMY1* F141C ($P = 2.5 \times 10^{-26}$); OR = 1.16 (1.10–1.23) per copy of *AMY1* C477R ($P = 8.3 \times 10^{-8}$); Fig. 3d). These two paralogous sequence variants were typically carried on haplotypes containing three or four copies of *AMY1* and produced the strongest conditional associations with dentures use in two stages of stepwise conditional analysis (Fig. 3f–h). The *AMY1* F141C and C477R variants seemed to confer an increase in dentures risk equivalent to increasing *AMY1* copy number by 22.4 (18.3–26.5) and 7.3 (4.6–9.9) copies, respectively (Fig. 3d). This apparent gain-of-function effect was surprising, as both variants were predicted to be damaging (PolyPhen-2[57] score of 1.0). Analyses of amylase protein expression did not detect effects of *AMY1* F141C on enzymatic activity (Supplementary Note 3, Extended Data Fig. 7a–c and Supplementary Fig. 1). The extended allelic series of *AMY1* copy number and missense variants associated with dentures use provided a set of genetic instruments for evaluating which bacterial species might causally contribute to tooth loss (Fig. 3i and Extended Data Fig. 7d). Reverse causality (that is, dentures use causing changes

in the oral microbiome that associate with *AMY1* variants) seemed to be unlikely based on the concordance of effect sizes in children and adults (Extended Data Fig. 7e).

The UKB and AoU datasets also enabled rigorous evaluation of whether or not *AMY1* copy number influences BMI among modern humans. *AMY1* copy number did not associate with BMI in UKB ($n = 418,150$, $P = 0.85$; Fig. 3j), AoU ($n = 219,879$, $P = 0.30$, Extended Data Fig. 7f) or any genetic ancestry in AoU (Extended Data Fig. 7g–i).

## Genetic associations with bacterial gene dosage

To identify molecular mechanisms by which human genetic variation engages the oral microbiome, we next tested whether the microbial species for which abundances associated with each of the 11 loci might be united by shared biochemical pathway use[58] (Extended Data Fig. 8a, Supplementary Tables 4 and 5 and Supplementary Note 4). This analysis identified an adhesin gene in *Haemophilus sputorum* at which sequencing coverage associated particularly strongly (relative to elsewhere in the *H. sputorum* genome) with *FUT2* W154X, suggesting that the adhesin interacts with FUT2-dependent glycosylation (Extended Data Fig. 8b and Supplementary Note 4). To search for similar molecular interactions between human and bacterial proteins, we tested the 11 lead variants associated with oral microbiome composition (Extended Data Table 1) for association with microbial gene dosages[24] (Fig. 4a). The key conceptual difference between testing human genetic variants for effects on microbial abundances (Fig. 2a) versus microbial gene dosages (Fig. 4a) is that the latter approach searches for effects on relative

fitness of bacterial strains that do or do not contain a genomic region (rather than fitness of a bacterial species). Thus, it highlights microbial genomic regions that may contain genes whose products are involved in a host–microbe genetic interaction (Supplementary Note 5).

To minimize hypothesis testing burden, we searched specifically for associations of the 11 variants with measurements of normalized WGS coverage in 500 base pair (bp) bins tiled across 30 bacterial reference genomes (Fig. 4a, Supplementary Table 6, Methods and Supplementary Note 5). This analysis identified 208 associations involving 68 regions of 18 bacterial genomes in which normalized read depth associated with one or more of the 11 human genetic variants (FDR < 0.01, Fig. 4b and Supplementary Tables 7 and 8). For example, amylase-binding protein orthologues in *Streptococcus parasanguinis*, *abpA* and *abpB*, increased in dosage with higher host *AMY1* copy number ($P = 1.13 \times 10^{-7}$ and $1.80 \times 10^{-7}$, respectively; Extended Data Fig. 8c–g and Supplementary Note 6). Normalized read depth in nearly half of these regions (33/68) associated with secretor status (based on *FUT2* W154X genotype). Eight regions associated with more than one human genetic variant; among them, five regions associated with both secretor status and *ABO\*A1* (Fig. 4b). Effect directions replicated for 202 of the 208 bin-level associations in 10,000 saliva-derived WGS samples from AoU (Extended Data Fig. 8h and Supplementary Table 7).

## ABO A antigen selects for a glycoside hydrolase

Host *ABO\*A1* genotype (based on rs2519093) strongly associated with whether *Prevotella* strains—a prevalent oral genus involved in early biofilm formation[59]—carried a gene encoding a glycoside hydrolase (Fig. 5a–d). *ABO\*A1* genotype associated exceptionally strongly ($P = 4.8 \times 10^{-19}$–$8.3 \times 10^{-241}$) with normalized WGS coverage across a 3 kilobase (kb) segment of the *Prevotella nanceiensis* reference genome (Fig. 5a). This region is annotated as a glycoside hydrolase pseudogene due to an N-terminal truncation in the reference genome. However, assembly of unmapped sequencing reads with mates aligned to the region showed no evidence of such truncation: rather, the reads seemed to originate from a full-length gene, with 95% homology to a glycoside hydrolase found in *Prevotella salivae* (a species not included among the 30 reference genomes analysed).

*ABO\*A1* genotype associated with sequencing coverage in this region only in secretor individuals ($P = 1.7 \times 10^{-307}$ in secretors; $P = 0.44$ in non-secretors), that is, individuals with at least one functional copy of *FUT2*, allowing expression of histo-blood group antigens on epithelial cells and secreted proteins (Fig. 5d). This *FUT2*-dependent effect of host blood group seemed to be driven specifically by A antigen presentation: the fraction of individuals for whom the glycoside hydrolase gene was detectable in saliva-derived DNA increased from 46%–48% in non-secretors and individuals with B or O blood type to 71%–77% in secretors with A or AB blood type (Fig. 5b). This association further reflected the quantity of A antigen predicted by an individual's diploid *ABO* genotype: *ABO\*O*, *B*, *A2* and *A1* alleles exhibited an allelic series of effects on normalized WGS coverage of the glycoside hydrolase gene that was consistent with the increasing abilities of the glycosyltransferases encoded by these alleles to synthesize A antigen (Fig. 5c). The *B* allele imposed a strong opposing effect when present in an individual heterozygous for an *A1* or *A2* allele ($\beta = -0.074$ [$-0.12$, $-0.032$] for *B* relative to *O*, $P = 5.8 \times 10^{-4}$, Fig. 5c), presumably reflecting competition between A and B transferases for available galactose residues on acceptor H antigens (Fig. 5d).

Taken together, these results indicate that the glycoside hydrolase enables *Prevotella* strains that express it to use type A histo-blood group antigens presented on host mucosal cell surfaces or salivary proteins (in secretors) as a carbohydrate source, similar to a recently observed effect in the gut microbiome[24]. We hypothesize that the glycoside hydrolase binds A antigens and cleaves the α1,2-fucosyl group synthesized by FUT2 (Fig. 5d).

Host *ABO* genotypes showed an intriguingly different pattern of association with a genomic region of the most abundant species in SPARK, *Rothia mucilaginosa* ($P = 1.4 \times 10^{-24}$; Fig. 5e,f). Blood groups A, B and AB all associated with absence (rather than presence) of this region of the *R. mucilaginosa* genome, and surprisingly, these associations were observed in non-secretors as well as secretors (Fig. 5e,f). This region contains genes that encode a protein with no annotated domains and a 3-isopropylmalate dehydrogenase functioning in leucine biosynthesis, leaving the mechanism of association unknown.

More broadly, this non-*FUT2*-dependent *ABO* association suggested the possibility that the association of *ABO\*A1* genotype with oral microbiome composition (Fig. 2b) might also be partially independent from secretor status. Indeed, in non-secretors the *ABO\*A1* association with microbiome composition remained significant ($P = 0.004$). This suggests that some effects of *ABO* variation on the oral microbiome come from cells not dependent on FUT2 for H antigen production (for example, blood and endothelial cells, which instead use FUT1). For example, bacteria can produce glycans structurally similar to A or B antigens that can then be recognized by anti-A or anti-B antibodies that are made by plasma cells and infiltrate into the mouth[60].

## Secretor status selects for microbial adhesins

Conversely, many regions in oral microbial genomes associated with secretor status but not *ABO\*A1* genotype (Fig. 4b), and oral microbiome composition associated much more strongly with secretor status than with any variant at *ABO* ($P = 3.0 \times 10^{-188}$ versus $P = 9.4 \times 10^{-15}$; Fig. 2b). This pattern contrasted with host genetic influences on gut microbiomes (which generate stronger associations at *ABO* than at *FUT2*; refs. 11–13), leading us to wonder whether these regions might point to a molecular mechanism by which secretor status influences oral microbiomes independently of *ABO*.

Examining genes in bacterial genomic regions associated with secretor status identified three classes of bacterial proteins that were each implicated by several genes. Proteins with YadA-like domains were encoded by nine genes in three species: *Veillonella* sp. 3627 (*vadA* through *vadF*), *Haemophilus sputorum* (*hadA* and *hadB*) and *Haemophilus parahaemolyticus* (*hadC*) (Fig. 6a–d and Supplementary Table 8). YadA (from *Yersinia pestis*) is a trimeric autotransporter adhesin that aids attachment to host cells by binding components of the extracellular matrix, and some such adhesins are known to recognize host protein glycosylation[61,62], such as the glycosylation added or enabled by FUT2. Five of the seven regions containing these genes were present (that is, not deleted) more often in the oral microbiomes of secretors than non-secretors, consistent with the hypothesis that the adhesins they encode bind histo-blood group antigens on the host cell surface. This was true of *hadC* in *H. parahaemolyticus* despite this species exhibiting lower abundance in secretors (Extended Data Fig. 8i). The genome of *V.* sp. 3627 contained three such regions (containing *vadB* through *vadF*, where *vadC* through *vadE* fall within the same complex region, Extended Data Fig. 8j) whose presence or absence was observed largely independently in different microbiomes (Fig. 6e). Classifying individuals on the basis of which combination of regions was present in their *V.* sp. 3627 population showed increasing enrichment of secretors among individuals with increasing representation of *vadB*–*vadF* genes in *V.* sp. 3627 (Fig. 6e).

*FUT2*-associated genomic regions additionally implicated two other classes of proteins that seemed to have roles in adhesion to host cells. Four proteins with CshA domains (CrpD and CrpE in *Streptococcus mitis*, CrpF and CrpG in *Streptococcus vestibularis*) and six proteins with mucin-binding domains (MucBP, Muc_B2, MucBP_2) (SmdA through SmdE in *S. mitis*, SmdF in *S. vestibularis*) were encoded by genes in *FUT2*-associated bacterial genomic regions (Fig. 6f,g). CshA from *Streptococcus gordonii* binds host fibronectin[63], a heavily glycosylated

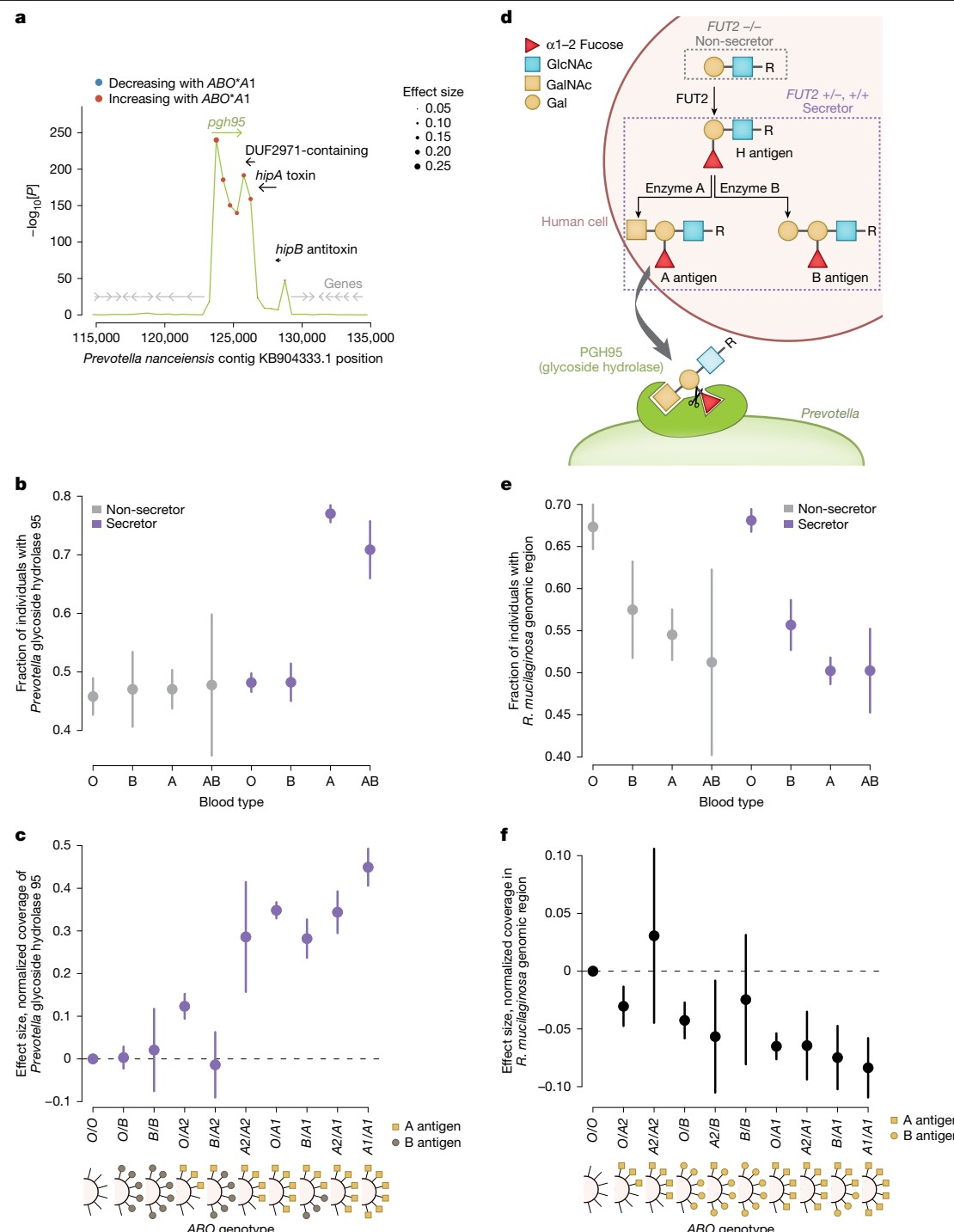

**Fig. 5 | *ABO* variation modulates selection on a glycoside hydrolase gene in *Prevotella* and also engages the oral microbiome in non-secretors.** **a**, Associations of host *ABO*A1* genotype with normalized coverage (truncated at 1) in 500 bp bins of the *P. nanceiensis* genome surrounding a glycoside hydrolase gene (*n* = 10,433). Dot colour, effect direction (red, higher coverage among individuals with A1 blood type); dot size, effect magnitude. Arrows indicate genes: glycoside hydrolase *pgh95* (A3GM_RS0109435, green), other genes overlapping the region associated with *ABO*A1* (black) and nearby genes (grey). *P* values, two-sided linear regression. **b**, Proportion of individuals whose oral microbiomes carry the *pgh95* gene (*n* = 10,433), stratified by blood type and secretor status. **c**, Effect sizes on normalized coverage in the *pgh95* region (123,500–124,000) for genotype combinations of common blood type alleles (*O, B, A2* and *A1*) relative to *O/O* individuals. Analyses were restricted to secretors (*n* = 8,278). Effect sizes seem to reflect expected abundances of A antigens (yellow squares, A antigens; grey circles, B antigens). **d**, Inferred interaction between glycosylation

of host cell proteins and bacterial glycoside hydrolase. Top (pink background), glycosylation patterns of human mucosal cell surface proteins and secreted proteins depend on an individual's combination of *FUT2* and *ABO* genotypes. In individuals with no functional copies of *FUT2* (non-secretors), type I H antigen is not produced, whereas in secretors, type I H antigen is produced and can be further glycosylated into A antigen or B antigen depending on *ABO* genotype (dashed purple outline). These antigens are then presented on mucosal cell surface and secreted proteins. The associations of *ABO* genotypes with presence of the *pgh95* gene in *Prevotella* strains suggest that the bacterial glycoside hydrolase protein (PGH95, green) is specifically targeting secreted type A antigens and cleaving the α1,2-fucosyl group, consistent with high amino acid homology (-75%) with α1,2-fucosidases in the glycoside hydrolase 95 (GH95) family. **e,f**, Analogous to **b,c**, respectively, for the *ABO*-associated region in *R. mucilaginosa* (*n* = 12,475). Error bars, 95% CIs in all panels.

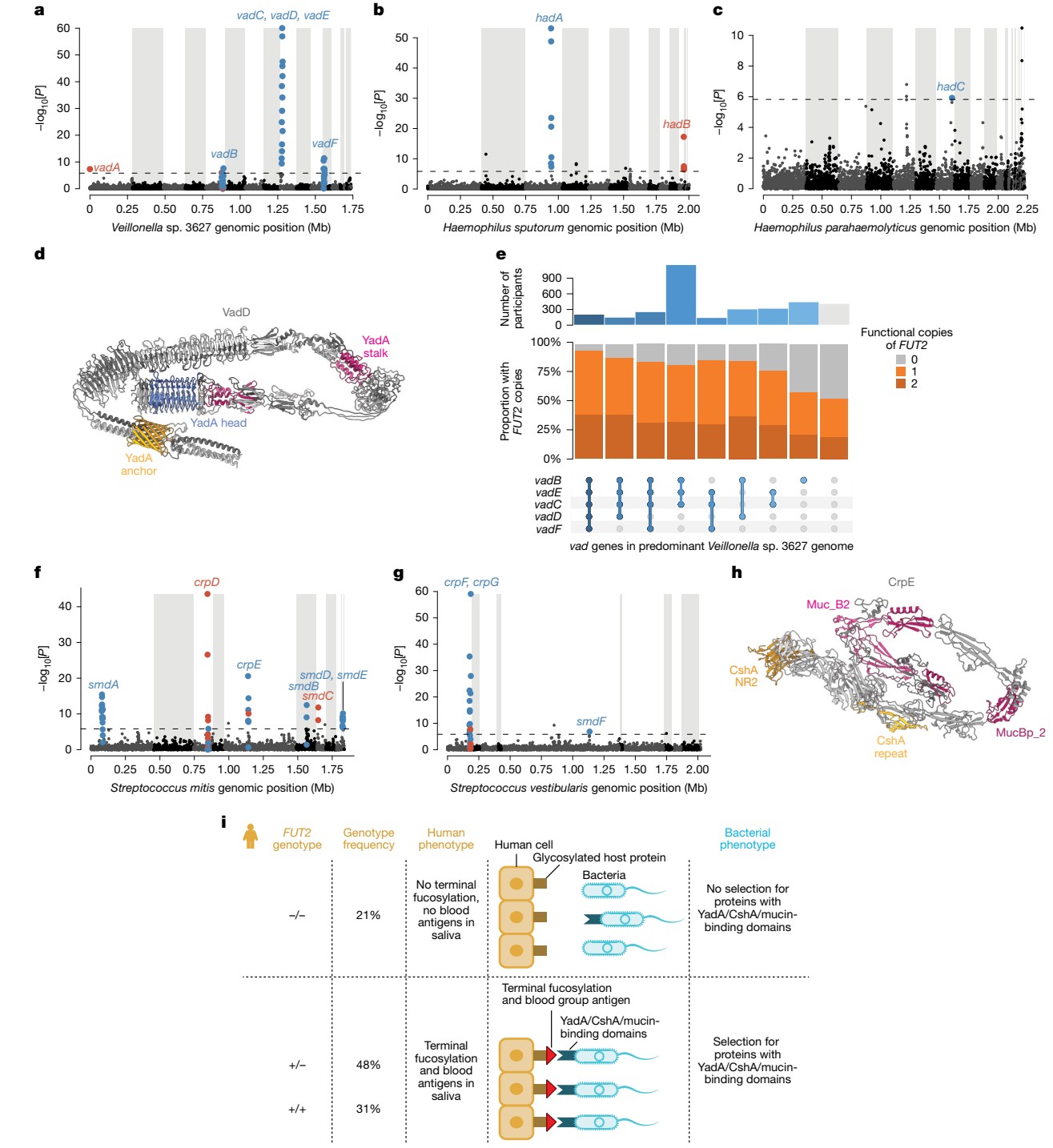

**Fig. 6 | *FUT2*-dependent secretor status selects for three classes of adhesin genes across five microbial species. a**, Associations of secretor status with normalized coverage (truncated at 1) in 500 bp bins of the *V.* sp. 3627 genome (*n* = 7,419). Shading indicates assembled contigs. Significant associations (FDR < 0.01) that overlap genes encoding proteins with YadA-like domains are highlighted (blue, genomic region more often present in secretors; red, absent). Effect directions are also indicated for bins that did not reach significance but were surrounded by significantly associated bins. **b**, Analogous to **a**, for *H. sputorum* (*n* = 8153). **c**, Analogous to **a**, for *H. parahaemolyticus* (*n* = 7,456). **d**, Predicted trimeric structure of VadD (from *V.* sp. 3627), where the head domain (blue) facilitates attachment to host proteins, stalk domains (magenta) flexibility and reach, and anchor domain (gold) translocation to bacterial surface. **e**, Upset plot of the relative proportions of *FUT2* W154X genotypes (non-secretors in grey, secretors in orange) among individuals with each

combination of *vadB*–*vadF* gene deletions in the *V.* sp. 3627 genome. Analysis was restricted to individuals with each gene either primarily present in strains of *V.* sp. 3627 (normalized coverage >0.8) or primarily absent (normalized coverage <0.2). Blue-to-grey shading of sets (bottom) and numbers of individuals per set (top) indicate the number of *vad* genes present. **f**, Analogous to **a**, for *S. mitis* (*n* = 12,479). Highlighted genes encode proteins that contain either a CshA domain (*crp* genes) or mucin-binding domain (*smd* genes). **g**, Analogous to **f**, for *S. vestibularis* (*n* = 11,723). **h**, Predicted structure of a portion of CrpE from *S. mitis*. The CshA NR2 (gold) and mucin-binding domains (magenta) both have lectin activity to their characterized ligands (fibronectin and mucin)[63,64]. **i**, Model of how host *FUT2* genotype selects for bacterial strains expressing proteins with YadA, CshA or mucin-binding domains that can attach to host cell surface proteins based on the availability of histo-blood type antigens. *P* values, two-sided linear regression (**a**–**c**,**f**,**g**).

component of the extracellular matrix. Similarly, mucins have numerous glycosylation sites and are up to 90% carbohydrate by mass[64]. Interestingly, one of these proteins, CrpE in *S. mitis*, seems to contain both a CshA domain and multiple mucin-binding domains (Fig. 6h and Supplementary Table 9), suggesting that these might function in concert to bind the same host protein or a combination of proximal targets multivalently.

These enrichments of genes encoding proteins with YadA, CshA and mucin-binding domains were unlikely to occur by chance: the genomes of *V.* sp. 3627, *H. sputorum* and *H. parahaemolyticus* only contain 12, 4 and 8 genes with YadA domains, respectively (Fisher's exact $P = 7.5 \times 10^{-12}$, $3.5 \times 10^{-5}$ and 0.021), and the genomes of *S. mitis* and *S. vestibularis* only contain two and three genes with CshA domains ($P = 7.4 \times 10^{-5}$ and $2.0 \times 10^{-5}$) and ten and three genes with mucin-binding domains ($P = 3.3 \times 10^{-11}$ and 0.0087). Most of these genes (15/19) were more commonly present in the oral microbiomes of people with functional *FUT2*, suggesting that they might encode bacterial lectins that depend on either fucosylation or sugar moieties added by ABO glycotransferase[65] (Fig. 6i). This convergence of bacterial genomic adaptations to host *FUT2* genotype broadly suggests that commensal bacteria commonly make use of host histo-blood group antigens not only as a carbohydrate source but also for bacterial attachment to host cell surfaces.

## Discussion

Analysis of the largest set of oral microbiome profiles generated to date identified many specific human genetic variants that contribute to the diversity observed across the oral microbiomes of different people[7,8]. The large number of such effects suggests a larger influence of human genetics on the oral microbiome than on the gut microbiome[66,67], perhaps because host cells in the mouth interface more directly with bacteria (in contrast to cells in the gut, which are typically protected by a mucosal barrier). Some of these genetic effects on microbial abundances seem likely to mediate associations of the same human genetic variants with oral health phenotypes, nominating bacterial species that may contribute to dysbiosis. The salivary amylase gene generated the strongest such shared effect on oral microbiomes and health, driven by both *AMY1* copy number variation and missense mutations in *AMY1*. The expansion of salivary amylase copy number in humans and domesticated animals has been hypothesized to be the result of positive selection driven by the advent of agriculture[44-47]. Our observation here that *AMY1* gene copy number variation associates with oral microbial phenotypes that lead to clinically relevant conditions—combined with the high mortality rate of tooth infections before modern dentistry and antibiotics[15]—suggests that *AMY1* copy number may have been under selection as a result of effects on oral health in addition—or in response—to dietary changes.

The numerous associations that these analyses uncovered between human genetic variants and bacterial gene dosages suggest frequent intergenomic adaptation of microbial species to individual human hosts and implicate specific molecular interactions likely to drive such adaptation. Most of these associations involved genes in bacterial species whose overall abundances were unaffected by the same human genetic variants, similar to recent observations of associations of BMI with gut microbial sequence variation[68], suggesting that genomic adaptations enable many bacterial species to survive equally well across variable host genetic environments. By contrast, an association with relative species abundance could imply that the microbial genome is unable to adapt to a particular human variation. The variable gene regions we identified showed some breakpoint heterogeneity (Extended Data Fig. 9a,b) and could either reflect gene dosage variation among circulating strains or recurrent mutations, such as in *Helicobacter pylori*[69]. The large number of such effects suggests that analyses of bacterial gene dosage may be a powerful way to identify host genetic

influences on microbiomes, perhaps because analysing the balance between members of the same species with and without a variable gene controls for strong environmental influences on species abundance.

We note a need for care in conducting GWAS of microbial-abundance phenotypes. We initially observed a strong association ($P = 2.5 \times 10^{-70}$) of oral microbiome composition with variant calls in the ribosomal RNA gene region of the p-arm of chromosome 21; however, these variant calls (which later failed a mappability filter) actually reflected the presence of orthologous bovine rDNA sequences and associated with the abundances of bacterial species used in dairy fermentation, suggesting DNA co-acquisition from recently eaten dairy foods (Supplementary Note 7). Our analytical approach for identifying host–microbe genetic interactions had several limitations that should be ameliorated with larger cohorts and improved microbial reference genomes (Extended Data Fig. 9c–h and Supplementary Note 8). Future datasets will also provide increased power to resolve possible pleiotropy and reverse causality with oral health phenotypes, either through cohorts with human genetic, microbiome and oral health phenotypes[70] or by Mendelian randomization approaches powered by even larger saliva sequencing datasets—which we have shown here provide rich information about how oral microbiomes are shaped by human genetics.

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

## Methods

### Ethics

This research complies with all relevant ethical regulations. The study protocol was determined to be not human subjects research by the Broad Institute Office of Research Subject Protection as all data analysed were previously collected and de-identified.

### Quantification of microbial relative abundances from saliva WGS

We analysed saliva-derived WGS data previously generated for 12,519 individuals from the SPARK cohort of the Simons Foundation Autism Research Initiative (SFARI)[25]. In brief, DNA extracted from saliva samples was prepared with PCR-free methods for 150 bp paired-end sequencing on Illumina NovaSeq 6000 machines. Reads were aligned to human reference build GRCh38 by the New York Genome Center using Centers for Common Disease Genomics project standards. Details of saliva sample collection, DNA extraction and sequencing were described in ref. 26 (which analysed data from sequencing waves WGS1–3 of the SPARK integrated WGS (iWGS) v.1.1 dataset; here we analysed WGS1–5, which included additional samples included in subsequent sequencing waves).

From the CRAM files previously aligned to GRCh38, we extracted all unmapped reads for subsequent generation and analysis of oral microbiome phenotypes. This retrieved a median of 67.5 million unmapped reads per sample ([35.9 million,126.1 million], quartiles), comparable with the total number of reads used for previous metagenomic characterization of human microbiomes[71] and consistent with previous analyses of SPARK samples for oral microbiome profiling[26,27]. Unmapped reads were converted to compressed FASTQ with samtools (v.1.15.1) and then used as input for microbiome profiling using MetaPhlAn (v.4.0.6) with the vOct22 reference database. To evaluate robustness of these oral microbiome profiles, relative abundances of species in SPARK samples were compared with those from samples in the Human Microbiome Project[72] by first subsetting to those profiled in both cohorts and then performing principal coordinate analysis using Bray–Curtis distance (Extended Data Fig. 1b).

From the relative abundance phenotype generated for each species by MetaPhlAn, we estimated the fractions of variance explained by covariates (age, sex, ASD and genetic ancestry PCs; Fig. 1c) using analysis of variance (finding the sum-of-squares for each covariate and dividing by the total sum-of-squares).

### Generation of mPCs

Relative abundance measures of all microbes were first filtered to 439 entries corresponding to microbial species found in at least 10% of SPARK DNA samples. Rank-based inverse normal transformation across individuals was then performed for the abundance of each species. This transformation did not always produce values with mean 0 and variance 1 (due to large fractions of samples with zero abundance), so these were then scaled and centred for use as input to PC analysis to obtain 439 orthogonal microbial abundance principal components (mPCs) representing orthogonal axes of microbial variation.

### Genotyping and quality control of human genetic variants in SPARK

Variant calling in SPARK was previously performed using DeepVariant (v.1.3.0) to produce sample-level VCFs from reads aligned to GRCh38 followed by GLnexus (v.1.4.1) to call variants jointly across the cohort. We performed a series of QC steps on the joint call set, starting by converting half-calls to missing and then excluding variants with >10% missingness using plink2 (v.2.00a3.6LM). Variants were further excluded if they had a minor allele frequency <1% or if they had a Hardy–Weinberg equilibrium exact test $P < 1 \times 10^{-6}$ with mid-$P$ adjustment for excessive heterozygosity, leaving 12,525,098 common variants. Genetic ancestry PCs were generated by LD-pruning variants in 500 kb windows with

$r^2 > 0.1$ and then running plink2 --pca approx. No individuals were filtered for outlier heterozygosity after inspection in each genetic ancestry group. Variants were then filtered to those present in the TOPMed-r3 imputation panel to exclude those in regions of poor mappability to produce a final set of 9,618,621 common variants to test for association with oral microbiome phenotypes.

### mPC-based GWAS of oral microbiome composition

A straightforward way to search for host genetic effects on microbiomes is to test human genetic variants for association with the abundance of each microbial taxon in turn[10–13]. However, we reasoned that a statistical test designed to aggregate evidence of pleiotropic genetic effects on many species in a microbial community could considerably increase statistical power[32–34]. To perform such a test in a scalable manner (efficient enough to test millions of human genetic variants), we made use of the decomposition of the microbial abundance matrix into 439 orthogonal mPCs. Specifically, we tested each genetic variant for association with each rank-based inverse normal transformed mPC, after which we summed the 439 test statistics obtained per variant to compute a single, combined association test for each variant (Fig. 2a; details in next section). Beyond increasing power to detect pleiotropic effects, the approach reduces multiple-testing burden by testing each genetic variant only once. We evaluated applying this approach to a subset of top axes of microbial variation (rather than all 439 mPCs) but did not observe a further increase in power, consistent with many axes of variation contributing association signal in this dataset (Extended Data Fig. 2f–p).

### Details of GWAS of oral microbiome composition

We performed GWAS on each mPC phenotype using the linear mixed model implemented in BOLT-LMM to account for the family structure of the SPARK cohort[30,31,73]. Specifically, we ran BOLT-LMM using the --lmmInfOnly flag (as the non-infinitesimal mixed model provided a negligible increase in statistical power) with the following covariates: sequencing batch, age, age squared, square root of age, sex, percentage of mapped reads and the top ten genetic ancestry PCs. A single father without a recorded age was assigned the average age of other fathers in the dataset. *AMY1* and *PRB1* copy numbers were rescaled to a range of [0,2] and encoded as dosages for association testing.

To test a genetic variant for association with an effect on overall oral microbiome composition, we summed chi-square statistics across the 439 orthogonal mPCs and computed the $P$ value based on a chi-squared distribution with 439 degrees of freedom. We computed $P$ values using a one-sided test, analogous to how in linear regression, one-sided chi-squared test statistics are computed (corresponding to two-sided tests of $z$-statistics).

### MDMR of oral microbiomes with selected genetic variants

We compared our test for genetic effects on oral microbiome composition with multivariate distance matrix regression (MDMR)[33] as implemented in the MDMR R package[36] (v.0.5.2), which finds significant predictors of multivariate outcomes by estimating the attributable amount of dissimilarity between samples. Rank-based inverse normal transformed relative abundances of the 439 most prevalent species (with or without initial centred log-ratio transformation) were used to generate the Euclidean distance matrix. MDMR was then run with the following covariates: sequencing batch, age, age squared, square root of age, sex and percentage of mapped reads. As genetic ancestry PCs frequently produced a singular matrix as a result of multicollinearity with individual variants, the top ten genetic PCs were first regressed from each tested variant (rather than including genetic PCs as covariates). The 11 loci identified from our mPC-based GWAS of oral microbiome composition were tested alongside 1,000 randomly selected variants on chromosome 1.

## Stratified LD score regression for estimating enrichment of heritability at genes with tissue-specific expression

We observed that the same mathematical framework that enables partitioning of heritability by means of stratified LD score regression on summary statistics from GWAS of a single trait[74] could be extended to analyse test statistics for association with oral microbiome composition (based on summing chi-squared test statistics across 439 mPCs). Starting from the representation of expected marginal chi-square association statistic for SNP $i$ based on linkage disequilibrium with variants in categories $C_k$,

$$E[\chi_i^2] = 1 + Na + N \sum_k \tau_k l(i, k)$$

averaging across the 439 chi-square statistics for each variant gives

$$E[\overline{\chi_i^2}] = 1 + N\overline{a} + N \sum_k \overline{\tau_k} l(i, k)$$

such that providing $\overline{\chi_i^2}$ as input to S-LDSC generates enrichments corresponding to $\overline{\tau_k}$. Averaged chi-square statistics per variant were used as input to munge_sumstats.py. LDSC was then run with baseline v.1.2, weights_hm3_no_hla as weights and previously described tissue-specific expression bins derived from Genotype-Tissue Expression (GTEx) project samples[37].

## GWAS of abundances of individual taxa

To avoid test statistic inflation from zero inflation and outlier values, relative abundance measures for each taxon were rank-based inverse normal transformed. Abundances of 1,262 taxa (of any phylogenetic level: species, genus, family and so on) observed in >10% of SPARK samples were then tested for association with host genotypes using BOLT-LMM. The top 20 PCs from PCA on 439 species observed at >10% prevalence (that is, the top 20 mPCs) were used as covariates to control for the largest axes of variation across samples, along with sequencing batch, age, age squared, square root of age, sex, percentage of mapped reads and the top ten genetic ancestry PCs. To test loci that might be associated as dominant/recessive rather than additive, BOLT-LMM was rerun using the --domRecHetTest flag.

To evaluate whether some of these associations could reflect compositional effects rather than being specific to the associated taxa, we computed an alternative set of taxon abundance phenotypes in which we took the centred log-ratio transform of relative abundances in each sample[75] (after replacing zero values observed for a given taxon with the minimum non-zero value for that taxon, to allow computing geometric means). Centred log-ratio transformed values for each taxa were then tested for association with host genotypes using BOLT-LMM with the same covariates as above (Extended Data Fig. 4e).

For estimating effect sizes of specific genotype values such as *FUT2* W154X genotypes (Fig. 2d), *AMY1* copy numbers (Fig. 3c) or *PRB1* copy numbers (Extended Data Fig. 3c), we used linear regression with the same covariates as above, encoding each genotype value (rounded if necessary) as a separate factor. The standard errors estimated by these regressions are slightly underestimated because they do not account for relatedness among SPARK participants, but we determined that this underestimation of standard errors was mild (~7% based on a ~14% inflation of chi-square test statistics computed using linear regression versus a linear mixed model for the 11 genome-wide significant loci (Supplementary Fig. 2)). To compare effect sizes in adults versus unrelated children, one child was randomly selected from each family in SPARK. BOLT-LMM was used to run linear regression on each of these subsets separately.

## GWAS in UK Biobank

Starting from 488,377 individuals in the UKB SNP-array dataset[43], individuals were excluded on the basis of the following criteria: 36,008 were removed to drop one relative in pairs of close relatives with kinship coefficient >0.0884, preferentially keeping individuals if they (1) reported having dentures or (2) reported not having dentures (that is, had a non-missing dentures phenotype); 28,701 were removed for not having European genetic ancestry[76]; 1,469 were removed for not having available TOPMed-imputed genotypes (including for chromosome X); 2,601 were removed for not having available WGS data; and 53 were removed for having withdrawn, leaving 419,545 available individuals for GWAS. For the binary oral health phenotypes (dentures use and bleeding gums), 418,039 had non-missing values. For the quantitative BMI $z$-score phenotype[77], 418,150 had non-missing values.

TOPMed-imputed variants for these individuals were filtered to require minor allele frequency >0.001 and INFO >0.3. BOLT-LMM was run in linear regression mode on these samples and variants with the following covariates: age, age squared, sex, genotype array, assessment centre and top 20 genetic ancestry PCs. For estimating effect sizes of specific copy numbers of *AMY1* (Fig. 3d,e,j), *AMY2A* (Extended Data Fig. 6f) or *AMY2B* (Extended Data Fig. 6g), we performed logistic regression (for oral health phenotypes) or linear regression (for BMI) with the same covariates, encoding each copy number (rounded to the nearest integer) as a separate factor and using the modal copy number as the reference level.

## Phyletic stratification of genetic associations

The phylogenetic tree of all species in the MetaPhlAn 4 database used (v.Oct22), mpa_vOct22_CHOCOPhlAnSGB_202212.nwk, was first subsetted to the tree spanning nodes with primary label among the 439 species seen at >10% prevalence in the SPARK cohort. This tree was used with graphlan (v.1.1.3) for depiction of phylogenetic trees (Figs. 1b and 2e and Extended Data Fig. 5f). For comparisons among the effect sizes of a human genetic variant associated with relative abundances of many species, phylogenetic distances between pairs of species were first computed as a cophenetic distance matrix from this tree. For a given index species A, phylogenetic distances between A and other species B were then compared with either (1) absolute values of effect sizes for species B (that is, $|\beta_B|$, Extended Data Fig. 5b,c,g,h) or (2) effect sizes for species B oriented relative to the effect direction for species A (that is, $sign(\beta_A) \times \beta_B$, Extended Data Fig. 5d,e,i,j).

## Estimation of *AMY1* copy number in all cohorts

In the SPARK (Extended Data Fig. 6a) and AoU v7 cohorts (Extended Data Fig. 6c), *AMY1* copy number was estimated by counting WGS reads that mapped to the duplicated regions that include *AMY1A* (chr. 1: 103638545–103666411), *AMY1B* (chr. 1: 103685558–103713427) and *AMY1C* (chr. 1: 103732687–103760549) in GRCh38 and normalizing against the total number of reads that aligned in either the 0.5 Mb upstream of *AMY2B* or the 0.5 Mb downstream of *AMY1C*. For the UKB cohort ($n = 490,415$ (ref. 78), Fig. 3a), we applied a more comprehensive read-depth normalization pipeline that incorporated sample-specific GC-bias correction inferred from genome-wide alignments (similar to Genome STRiP[79]) before normalizing against read depth in the 0.5 Mb regions flanking the amylase locus. We corrected for slight miscalibration of these diploid copy number estimates by fitting a linear model to identify coefficients that centred peaks of copy number estimates at integers.

Among the UKB participants with WGS available, we identified 5,149 siblings that shared both amylase haplotypes IBD2 (based on at most three mismatching SNP-array genotypes in a 2 Mb window flanking the amylase locus, computed using plink1.9 --genome). Among these IBD2 sibling pairs, 13 pairs were identified as copy number discordant (and likely to reflect a copy number mutation in the past generation) based on (1) having *AMY1* copy number estimates that differed by >1.0 and (2) having *AMY2A* copy number estimates consistent with a duplication or deletion of a commonly variable amylase gene cassette (that is, ±1 *AMY2A* copies for *AMY1* copy number discordances of odd parity and no

difference in *AMY2A* copy number for *AMY1* copy number discordances of even parity). We estimated *AMY1* copy number genotyping accuracy by computing the correlation across IBD2 sibling pairs excluding these 13 copy number discordant pairs.

## Testing *AMY1* copy number for association with dental phenotypes and BMI in All of Us

We defined the binary tooth loss phenotype as 1 for individuals with at least one recorded instance of 'acquired absence of all teeth' (OMOP concept ID: 40481327, code: 441935006) and 0 otherwise. Likewise, the caries phenotype was derived from 'dental caries' (OMOP concept ID: 133228, code: 80967001). For BMI, we generated a normalized *z*-score phenotype from BMI (OMOP concept ID: 903124, code: bmi) by first adjusting for age (in months, determined from time at weight and height measurement) and age squared and then applying inverse rank normal transformation in each sex separately. The BMI *z*-scores for males and females were then merged together.

Individuals with WGS available for genotyping *AMY1* copy number were first filtered to an unrelated subset of samples (iteratively dropping one individual per related pair with kinship score >0.1, from relatedness_flagged_samples.tsv). For oral health phenotypes, we performed logistic regression against *AMY1* copy number including age, age squared, sex and the top 16 genetic ancestry PCs (from ancestry_preds.tsv) as covariates. For BMI, we performed linear regression using only genetic ancestry PCs as covariates as age and sex had already been residualized out. For estimating effect sizes of specific copy numbers of *AMY1* (Extended Data Fig. 6d,e and Extended Data Fig. 7f–i), we performed logistic regression (for oral health phenotypes) or linear regression (for BMI) with the same covariates, encoding each copy number (rounded to the nearest integer) as a separate factor and using the modal copy number of 6 as the reference level (that is, computing the effect size of each copy number relative to copy number 6).

## Paralogous sequence variation in *AMY1*

To identify and genotype paralogous sequence variants (PSVs) from UKB WGS data, we used a read-counting approach similar to our previous work[80]. In brief, reads from each sample that had been aligned to any of the three 27.6 kb regions in GRCh38 corresponding to *AMY1A* (chr. 1: 103638695–103666261), *AMY1B* (chr. 1: 103685708–103713277) and *AMY1C* (chr. 1: 103732837–103760399) were realigned with bwa (v.0.7.17) to the reference sequence of *AMY1A* after filtering out reads with any of the last four SAM flags (-F 0xF00). Read counts supporting each base at each position were tabulated with htsbox (r345) pileup, filtering alignments <50 bp and base calls with quality score <20. Individuals were called heterozygous for a PSV allele (having at least one copy of *AMY1* with each of two alleles) if at least five reads supported the variant allele in that sample and at least five reads did not. PSVs were then filtered to those with heterozygosity >0.002 (resulting in 892 PSVs passing filters). To estimate diploid copy number genotypes for a PSV, we multiplied each individual's diploid *AMY1* copy number by the allelic fraction of the PSV in that individual. In association tests using linear regression with BOLT-LMM, we rounded copy number estimates to integer genotypes.

For follow-up analyses of effect sizes of specific *AMY1* F141C and C477R copy number genotypes, we optimized the assignments of integer copy number genotypes based on manual inspection of histograms of allelic depth-derived PSV copy number estimates (Supplementary Fig. 3). Specifically, we assigned copy numbers for each of F141C and C477R using the thresholds [0,0.25) = CN0, [0.25,1.75) = CN1, [1.75,2.7) = CN2, [2.7,3.5) = CN3 and [3.5,5) = CN4. In UKB, F141C had 416,381 (99.2%), 3,124 (0.74%) and 40 (0.0095%) individuals with 0, 1 and 2 copies, respectively, and C477R had 412,450 (98.3%), 6,527 (1.6%), 547 (0.13%), 19 (0.0045%) and 2 (0.0005%) individuals with 0, 1, 2, 3 and 4 copies, respectively. In SPARK, F141C had 12,459 (99.5%) and 60 (0.48%) individuals with 0 and 1 copies, respectively, and C477R had

12,343 (98.6%), 172 (1.4%) and 4 (0.032%) individuals with 0, 1 and 2 copies, respectively. These threshold-based copy numbers of the alternate alleles were used in logistic regression along with copy numbers of the reference alleles (F141 and C477) and covariates as above.

## Protein expression of AMY1

Plasmid pCAGEN[81] was a gift from C. Cepko (Addgene plasmid no. 11160; http://n2t.net/addgene:11160; RRID Addgene_11160). Codon-optimized sequences encoding reference, F141C and C477R *AMY1* alleles were synthesized and ordered as gBlocks from Integrated DNA Technologies for cloning into pCAGEN downstream of the CAG promoter. Clones were screened for sequence errors before plasmid preparation using Plasmid Plus Midi Kit (Qiagen, catalogue no. 12943). Plasmid pUC19 (New England Biolabs, catalogue no. N3041S) was used as a negative control. A total of 15 µg of each plasmid was lipofected into separate 10 cm plates of HEK293T (Takara, catalogue no. 632180) cells at ~70% confluence using 30 µl of Lipofectamine 3000 (Invitrogen, catalogue no. L3000015). Authentication of HEK293T was done by morphological match for type and verification of SV40T antigen with PCR assay. Lack of mycoplasma contamination was confirmed by Takara as well as inhouse with MycoAlert Mycoplasma Detection Kit (Lonza, catalogue no. LT07-318). Medium was switched to serum-free after 24 h before collection of both supernatant and lysate at 72 h post-lipofection. A total of 10 ml of supernatant was spun at 1,000*g* for 10 min to remove cells and debris before the addition of 100 µl of Halt Protease Inhibitor Cocktail (100×, Thermo Scientific, catalogue no. 78438) and 100 µl of EDTA (0.5 M). Cells were washed with ice-cold PBS before the addition of cold 1 ml of RIPA Lysis and Extraction Buffer (Thermo Scientific, catalogue no. 89900) with 10 µl of Halt Protease Inhibitor Cocktail (100×) and 10 µl of EDTA (0.5 M). After sufficient solubilization of the cells had occurred, 1 µl of Benzonase Nuclease (250 U per µl, Milipore, catalogue no. E1014) and 10 µl of MgCl$_2$ (1 M) were added before incubation at 37 °C, 500 rpm for 30 min. Lysate was then spun at 10,000*g* for 10 min to remove insoluble precipitate. Both supernatant and lysate were stored at −80 °C until further use.

## Western blot of AMY1 in cell culture supernatant and lysate

A total of 7.5 µl of supernatant or purified lysate was first run denatured and reduced in a 10% Mini-PROTEAN TGX Precast Protein Gel (Bio-Rad, catalogue no. 4561036) before wet transfer to nitrocellulose membrane (120 V, 2 h). After evaluation of equal loading by Ponceau S Staining Solution (Thermo Scientific, catalogue no. A40000279), the membrane was blocked for 1 h at room temperature with TBS, 0.1% Tween-20 and 5% w/v non-fat dry milk before washing three times for 5 min each with TBS-T (TBS, 0.1% Tween-20). Amylase antibody (G-10, Santa Cruz Biotechnology, catalogue no. sc-46657, lot no. G0324) was used as the primary antibody at a 1:200 dilution in TBS-T with 5% w/v milk for incubation overnight at 4 °C with rotation. Membrane was washed three times for 5 min each with TBS-T before addition of anti-mouse IgG, HRP-linked antibody (Cell Signaling Technology, catalogue no. 7076, lot no. 39) as secondary at a 1:2,000 dilution in TBS-T with 5% w/v milk for 1 h incubation at room temperature with rotation. Membrane was washed three times for 5 min each with TBS-T before detection with Amersham ECL Prime Western Blotting Detection Reagent (Cytiva, catalogue no. RPN2236) (Extended Data Fig. 7a).

## Purification of AMY1 reference and F141C protein

Purification of amylase from supernatant was done using glycogen, adapted from refs. 82,83. All following steps were conducted on ice or at 4 °C (rotation and centrifugation). In brief, 10 ml of supernatant was initially concentrated using prewet 15 ml Amicon Ultra Centrifugal Filter, 30 kDa MWCO (Milipore, catalogue no. UFC9030) and put up to a total volume of 900 µl with PBS. A total 600 µl of cold ethanol was added slowly to make 40% ethanol v/v. This was then centrifuged at 10,000*g* for 10 min to remove any insoluble precipitate. To the supernatant, 75 µl

of 0.2 M sodium phosphate buffer (pH 8), 75 μl of glycogen (20 mg ml$^{-1}$, Roche, catalogue no. 10901393001) and 100 μl of ethanol were added in that order. This was then incubated for 5 min with end-over-end mixing before centrifugation at 5,000$g$ for 6 min. The glycogen pellets were then washed twice with 10 mM sodium phosphate buffer (pH 8) containing ethanol (40% v/v) before resuspension in 100 μl of 50 mM MOPS buffer (pH 7, Thermo Scientific, catalogue no. J61821-AK) with 5 mM CaCl$_2$. These were then incubated at 37 °C, 500 rpm for 30 min to allow for glycogen digestion. The above purification procedure was repeated as necessary to arrive at purified amylase as evaluated on 10% Mini-PROTEAN TGX Precast Protein Gel with InstantBlue Coomassie Protein Stain (Abcam, catalogue no. ab119211) (Extended Data Fig. 7b).

## Amylase activity assay

Protein was normalized between reference and F141C isoforms by densitometric measurement against linear dilution series ($r^2 = 0.99$ with input). The concentration chosen for the assay was determined by having maximal linear activity over the observation window. For each technical replicate, 10 μl of reference or F141C amylase enzyme diluted in 50 mM MOPS buffer (pH 7), 5 mM CaCl$_2$ and 0.02% w/v BSA (New England Biolabs, catalogue no. B9000) was quickly mixed with 10 μl of BODIPY FL conjugated-starch substrate from EnzChek Ultra Amylase Assay Kit (Invitrogen, catalogue no. E33651) in 50 mM MOPS buffer (pH 7) with 5 mM CaCl$_2$. The reaction was then maintained at 20 °C for 2 h in a Bio-Rad CFX384 Real-Time PCR Detection System with fluorescence reading taken every minute using FAM fluorophore settings with CFX Manager (v.3.1) software. Fluorescence at 30 min relative to initial reading was used as input to a linear model with allele and plate to regress out any run-to-run effects during comparison across technical replicates (Extended Data Fig. 7c).

## Bacterial genome reference panel for analysing bacterial gene dosages

To reduce hypothesis testing burden in association analyses of human genetic variants with bacterial gene dosage phenotypes, we restricted analyses to a set of 30 bacterial genomes representing highly abundant species and species whose abundances we had found to associate with human genetic variants. Specifically, we selected 30 bacterial species with genomes available in GenBank by including: (1) the five most abundant species in SPARK oral microbiomes, (2) species that associated strongly ($P < 4 \times 10^{-11}$) with at least one human genetic variant and (3) the top two associated species for each locus if not already included. We substituted *Stomatobaculum longum* for the related *Stomatobaculum* SGB5266 which would have been included under (2) but lacked a GenBank assembly. We selected a single genome for each of the 30 species using the following criteria. The SGB centroid was prioritized over other genomes if it corresponded to a GenBank assembly. For cases in which the centroid was not a GenBank assembly and multiple genomes corresponded to GenBank assemblies, the reference genome was prioritized if available, or otherwise the highest ranked genome among those listed. A list of these species and GenBank assemblies is included in Supplementary Table 6. A bowtie2 index was then built from these merged genomes.

## Measuring bacterial gene dosages using read-depth phenotypes

We computed WGS coverage-derived phenotypes informative of gene dosage across each of the 30 bacterial reference genomes by first realigning unmapped reads from SPARK saliva WGS to the 30 reference genomes using bowtie2 (v.2.5.1) with the --very-sensitive flag. These alignments were then position-sorted within contigs with samtools (v.1.15.1). For each WGS sample, we quantified read depth in 500 bp bins tiling each of the 30 bacterial reference genomes using mosdepth (v.0.3.6), excluding reads with mapping quality <5. For each sample, for each of the 30 bacterial reference genomes, we then median-normalized the bin-level read-depth measurements across the 500 bp bins of that

reference genome to control for species abundance (such that normalized read-depth measurements had a median of 1 among bins corresponding to each species). If a sample had <0.5× median coverage across bins corresponding to a given species, we set that sample's normalized read-depth measurements for that species to missing to focus downstream analyses on samples with less-noisy measurements. Finally, we truncated median-normalized read-depth values to the interval [0,1], both to focus on deletions in bacterial genomes and to reduce the influence of outlier measurements that might reflect mismapped reads (potentially derived from either duplicated genomic regions or homologous sequences in microbial species not represented among the 30 reference genomes). We reasoned that these bin-level measurements would capture kilobase-scale deletions of bacterial genomes, circumventing the need to predefine a set of structural variant regions (which was difficult because of the limited sequencing coverage of most species). Additional details are provided in Supplementary Note 5.

## Testing bacterial gene dosage phenotypes for association with host genotypes

We used linear regression to test each of the 11 human genetic variants we had found to associate with oral microbiome composition (Extended Data Table 1) for association with normalized read depth (truncated to [0,1]) in each 500 bp bin of each of the 30 bacterial reference genomes. We used an additive model for all variants except *FUT2* W154X and *TLR1* I602S, for which we used a recessive model (corresponding to secretor/non-secretor status for FUT2). We took two precautions to avoid potential confounders. First, for each of the 30 species, we included as covariates the top 20 PCs of the normalized, truncated read-depth matrix for that species (running PCA after centring and scaling each bin to have a mean of 0 and s.d. of 1 across samples) to control for linked gene dosages (for example, differences across strains) that could potentially generate non-causal associations in a manner analogous to population structure in GWAS (Supplementary Note 5). Second, we applied a form of genomic control[84] (applied across the 500 bp bins of each reference genome) to adjust for remaining test statistic inflation. Specifically, for each pairing of a species and a human genetic variant, we computed the adjustment factor

$$\frac{\mathrm{median}(\chi_1^2)}{F^{-1}(0.5)}$$

across the $\chi_1^2$ test statistics for the 500 bp bins of that reference genome, where $F^{-1}(x)$ is the inverse cumulative distribution function for a $\chi_1^2$ random variable. The $\chi_1^2$ test statistics were then divided by this factor. This yielded 208 read-depth bins in 18 species that significantly associated with at least one of the 11 human genetic variants (FDR < 0.01, Supplementary Tables 7 and 8) and resolved to 68 unique microbial regions after merging bins within 1.5 kb of another significantly associated bin (Fig. 4b). We verified that the truncated normalized read-depth phenotypes involved in these associations were broadly reasonably distributed (with most bimodal at 0 and 1 and mean between 0.05 and 0.95; Supplementary Table 7), such that testing these phenotypes for association with common variants (MAF = 0.08–0.45) using linear regression in a cohort of size 12,519 was expected to produce robust test statistics[77].

For combinatorial analysis of deletions of the five *vad* genes in *V.* sp. 3627 that associated in the same direction with *FUT2* genotype (*vadB* through *vadF*) (Fig. 6e), we first selected individuals whose oral microbiomes had evidence of near-complete presence (>0.8 median-normalized coverage) or near-complete absence (<0.2 median-normalized coverage) of each *vad* gene ($n = 3,081$, representing roughly half of the SPARK samples with coverage of the *V.* sp. 3627 genome reaching the >0.5× threshold for analysis). For each common combination (>2.5% of selected individuals) of presence/absence status of *vadB* through *vadF*, the number of individuals with each *FUT2* W154X genotype were then counted.

## Replication of bacterial gene dosage associations in the All of Us cohort

To evaluate the generalizability of the associations we identified in SPARK between human genetic variants and bacterial gene dosages, we applied the same computational pipeline described above to 10,000 randomly selected saliva-derived WGS samples from the AoU v.8 data release. We then attempted to replicate the 208 associations (between human genetic variants and normalized read-depth measurements in 500 bp bins of bacterial genomes) that had reached significance (FDR < 0.01) in SPARK.

## AlphaFold3 prediction of protein structures

To predict protein structures for *Streptococcus parasanguinis* AbpA or AbpB (bound to human AMY1), the *Veillonella* sp. 3627 VadD trimer and *Streptococcus mitis* CrpE, we used AlphaFold3 (ref. 85) for multimer prediction with default reference databases and max template date of 2021-09-29. The TonB domain of AbpB and the signal peptide of VadD were excluded for visualization. To minimize the model size, CrpE was truncated to the region spanning the CshA NR2 domain to the sixth mucin-binding domain (residues 572–2701). Structures were visualized with ChimeraX (v.1.9)[86] and pLDDT, pTM and ipTM values can be found in Supplementary Table 9.

## Genetically derived blood typing in SPARK

We assigned blood types to SPARK participants on the basis of WGS-derived SNP and indel genotypes using a procedure similar to previous work[12]. The genotype of the rs8176746 missense SNP was first used to determine an individual's dosage (that is, allele count) of type B alleles (T allele count) and non-type B dosage (G allele count).

The rs8176719 indel was next used to determine type O1 dosage (deletion allele count), which was subtracted from the non-type B dosage to yield non-type B/O1 dosage, as the rs8176719 deletion allele typically occurs on haplotypes that would otherwise be type A alleles. Although this is true in European, East Asian and American ancestry haplotypes in 1KGP populations, in a small fraction of African (3.2%) and South Asian (0.2%) ancestry haplotypes, the rs8176719 deletion occurs in *cis* with the type B missense allele. As we found 36 SPARK participants who had O1 dosage exceeding non-type B dosage, we subtracted this excess from their type B dosages.

The rs41302905 missense SNP was next used to determine type O2 dosage (T allele count) and subtracted from the non-type B/O1 dosage to yield type A dosage, as it seems to be in *cis* with type A alleles in all 1KGP populations. O1 and O2 dosages were then merged to compute type O dosage.

The rs56392308 indel was next used to determine the type A2 dosage (deletion allele count) and subtracted from the type A dosage to yield type A1 dosage. For seven individuals in which the type A2 dosage exceeded type A dosage, five seemed to be on type O alleles and two on either type O or type B alleles, so this excess was subtracted from their type A2 dosage.

## Enrichment of conserved domains in bacterial genes associated with secretor status

For each of the bacterial species with adhesin genes in dosage variable regions that associated with *FUT2* loss of function (Fig. 6a–c,f,g), protein IDs (WP numbers) for the species were extracted from its RefSeq general feature format (GFF) file. Conserved domains (from National Center for Biotechnology Information (NCBI) conserved domain database) were identified for each protein using a modified version of the provided bwrpsb.pl script (applied to up to 250 proteins at a time). A one-sided Fisher's exact test was used to identify domains enriched among proteins encoded by genes within read-depth bins that associated with *FUT2* genotype.

## GWAS of read depth in selected microbial genomic regions

We used BOLT-LMM to perform GWAS on the five normalized read-depth phenotypes for which we observed the most significant associations with human genetic variants (in our targeted analysis of 11 human genetic variants). Specifically, these phenotypes measured WGS read depth in the following 500 bp bins: *H. sputorum* QEQH01000003.1: 197,000–197,500, *P. nanceiensis* KB904333.1: 123,500–124,000, *S. mitis* MUYN01000003.1: 100,000–100,500, *S. vestibularis* AEKO01000011.1: 186,000–186,500, *V.* sp. 3627 RQVG01000009.1: 13,500–14,000. In each GWAS, we included as covariates the top 20 PCs of the normalized, truncated read-depth matrix for the species under consideration, along with sequencing batch, age, age squared, square root of age, sex, percentage of mapped reads and the top ten human genetic PCs.

We also used BOLT-LMM to perform GWAS on normalized read-depth measurements in 500 bp bins spanning the genome of *R. mucilaginosa* (the most prevalent species observed in SPARK). To avoid test statistic inflation due to non-normality, we first excluded bins for which <10% of samples had non-modal read-depth values, leaving 3,441 bins. We then rank-based inverse normal transformed these bins (with random tie-breaking) to further normalize the phenotypes. We ran BOLT-LMM using the same covariates as above.

## Reporting summary

Further information on research design is available in the Nature Portfolio Reporting Summary linked to this article.

## Data availability

The following data resources are available by application: UKB (http://www.ukbiobank.ac.uk/), All of Us Research Program (https://allofus.nih.gov/) and SFARI SPARK (https://www.sfari.org/resource/spark/). To protect participant confidentiality, approved researchers can obtain access to the SPARK population dataset described in this study (SPARK integrated WGS (iWGS) v.1.1) by applying at https://base.sfari.org. Quantifications of microbial abundances in SPARK generated in this study can also be obtained from SFARI Base (Dataset DS0000116). Summary statistics from GWAS of microbial abundances in SPARK are available from the GWAS Catalog under accessions GCST90709872 to GCST90711133. Summary statistics from mPC-based GWAS of oral microbiome composition are available at Zenodo (https://doi.org/10.5281/zenodo.14559457)[87]. The following data resources are publicly available: Human Microbiome Project (https://hmpdacc.org/), human reference genome build GRCh38 (https://ftp.1000genomes.ebi.ac.uk/vol1/ftp/technical/reference/GRCh38_reference_genome/), MetaPhlAn v.Oct22 reference database (http://cmprod1.cibio.unitn.it/biobakery4/metaphlan_databases/), TOPMed-r3 imputation panel variant list (https://imputation.biodatacatalyst.nhlbi.nih.gov/), LD score resources https://alkesgroup.broadinstitute.org/LDSCORE/), NCBI GenBank (https://www.ncbi.nlm.nih.gov/genbank/) and NCBI Conserved Domain Database (https://www.ncbi.nlm.nih.gov/Structure/cdd/cdd.shtml).

## Code availability

The following publicly available software resources were used: MetaPhlAn (v.4.0.6, http://segatalab.cibio.unitn.it/tools/metaphlan/index.html), DeepVariant (v.1.3.0, https://github.com/google/deepvariant), GLnexus (v.1.4.1, https://github.com/dnanexus-rnd/GLnexus), HUMAnN (v.3.8, https://huttenhower.sph.harvard.edu/humann), GraPhlAn (v.1.1.3, http://segatalab.cibio.unitn.it/tools/graphlan/index.html), mosdepth (v.0.3.6, https://github.com/brentp/mosdepth), bowtie (v.2.5.1, https://bowtie-bio.sourceforge.net/bowtie2/index.shtml), bcftools (v.1.14, http://www.htslib.org/), samtools (v.1.15.1, http://www.htslib.org/), plink (v.1.90b6.26 and v.2.00a3.7, https://www.cog-genomics.org/plink/), BOLT-LMM (v.2.4.1, https://alkesgroup.broadinstitute.org/BOLT-LMM/), qqman (v.0.1.8, https://cran.r-project.org/web/packages/qqman/index.html), MDMR (v.0.5.2, https://cran.r-project.org/web/packages/MDMR/index.html), bedtools

(v.2.27.1, https://bed-tools.readthedocs.io/en/latest/), AlphaFold3 (https://alphafoldserver.com/) and ChimeraX (v.1.9, https://www.cgl.ucsf.edu/chimerax/). Custom code used to generate results in this study is available at Zenodo (https://doi.org/10.5281/zenodo.14559457)[87].

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

**Acknowledgements** We thank M. Florio, W.-S. Lee, B. Bor and C. Huttenhower for helpful discussions. We are grateful to all the families in SPARK, the SPARK clinical sites and SPARK staff. We appreciate obtaining access to SPARK genetic data on SFARI Base. This research was conducted using the UK Biobank Resource under application no. 40709. N.K. was supported by US NIH Fellowship F31 DE034283. R.E.H. and S.A.M. were supported by US NIH grant R01 HG006855. M.L.A.H. was supported by US NIH Fellowship F32 HL160061. R.E.M. was supported by US NIH grant K25 HL150334. S.A.M. was supported by the Howard Hughes Medical Institute. P.-R.L. was supported by US NIH grants R56 HG012698 and R01 HG013110 and a Burroughs Wellcome Fund Career Award at the Scientific Interfaces. The funders had no role in study design, data collection and analysis, the decision to publish or the preparation of the manuscript. The content is solely the responsibility of the authors and does not necessarily represent the official views of the NIH. Computational analyses were performed on the O2 High Performance Compute Cluster supported by the Research Computing Group at Harvard Medical School (http://rc.hms.harvard.edu), the UKB Research Analysis Platform and the All of Us Researcher Workbench. Molecular graphics and analyses were performed with UCSF ChimeraX, developed by the Resource for Biocomputing, Visualization and Informatics at the University of California, San Francisco, with support from National Institutes of Health R01-GM129325 and the Office of Cyber Infrastructure and Computational Biology, National Institute of Allergy and Infectious Diseases. The All of Us Research Program is supported by the NIH, Office of the Director: Regional Medical Centers: 1 OT2 OD026549; 1 OT2 OD026554; 1 OT2 OD026557; 1 OT2 OD026556; 1 OT2 OD026550; 1 OT2 OD026552; 1 OT2 OD026553; 1 OT2 OD026548; 1 OT2 OD026551; 1 OT2 OD026555; IAA no. AOD 16037; Federally Qualified Health Centers: HHSN 263201600085U; Data and Research Center: 5 U2C OD023196; Biobank: 1 U24 OD023121; The Participant Center: U24 OD023176; Participant Technology Systems Center: 1 U24 OD023163; Communications and Engagement: 3 OT2 OD023205; 3 OT2 OD023206; and Community Partners: 1 OT2 OD025277; 3 OT2 OD025315; 1 OT2 OD025337; and 1 OT2 OD025276. In addition, the All of Us Research Program would not be possible without the partnership of its participants.

**Author contributions** N.K., S.A.M. and P.-R.L. conceived the study design. N.K. carried out the computational and statistical analyses. R.E.H., R.E.M. and P.-R.L. developed the read-depth normalization pipeline on UKB. M.L.A.H. helped with performing the genome-wide association for dentures use on UKB. R.E.H. and C.L.U. helped with characterizing variation at the amylase locus. N.K. carried out the in vitro experiments. N.K., S.A.M. and P.-R.L. wrote the paper with contributions from all authors.

**Competing interests** The authors declare no competing interests.

**Additional information**
**Correspondence and requests for materials** should be addressed to Nolan Kamitaki, Steven A. McCarroll or Po-Ru Loh.

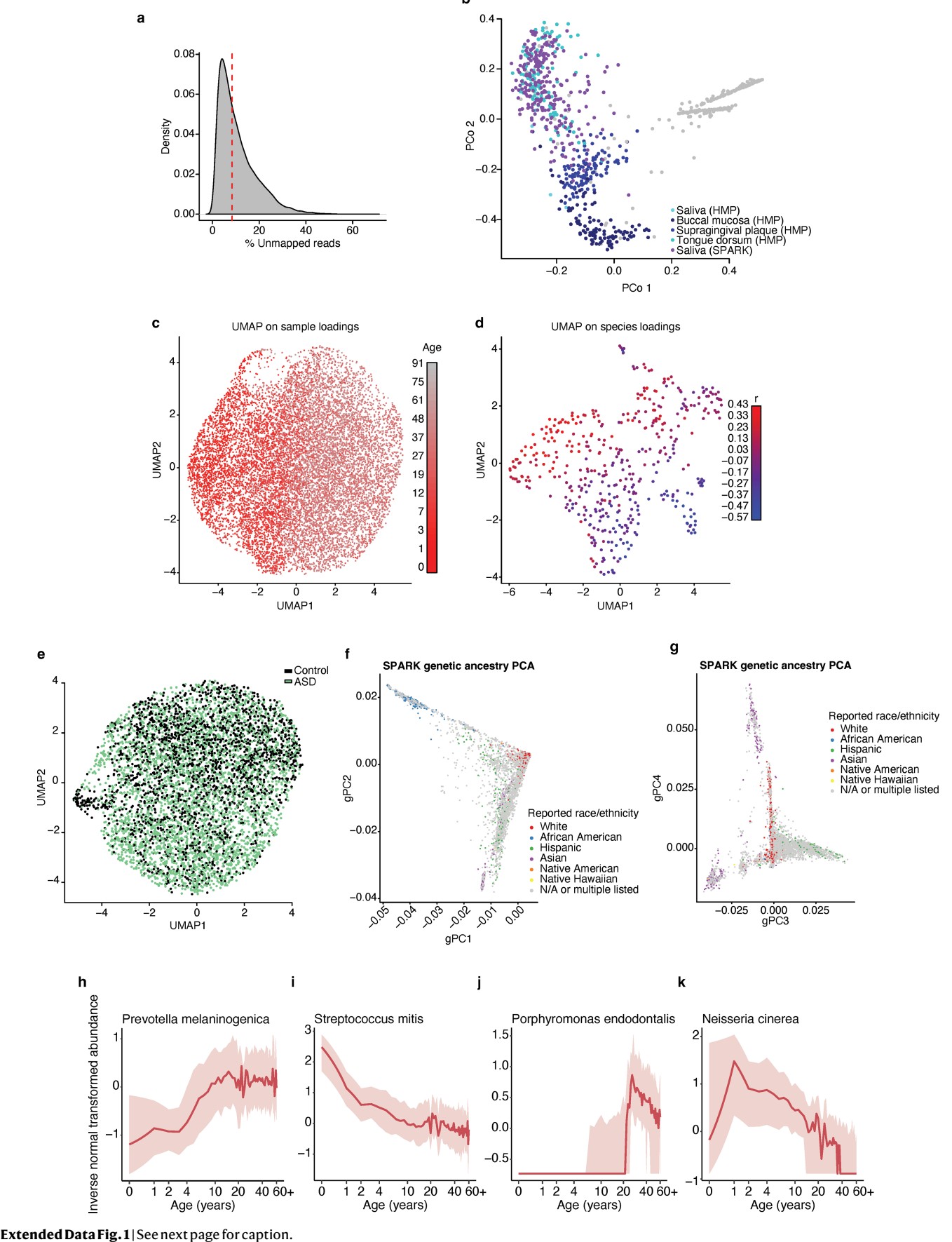

**Extended Data Fig. 1 |** See next page for caption.

**Extended Data Fig. 1 | Oral microbiome composition changes with host age.**
**a**, Distribution of the percent of reads not mapping to the human reference (GRCh38) across SPARK participants (n = 12,519). The median of 8.39% is indicated by the dotted red line. **b**, Principal coordinate analysis of microbiome profiles from the Human Microbiome Project[72] together with a randomly selected subset of SPARK samples (n = 250). PCoA was performed using Bray-Curtis distance on all microbial species profiled in both data sets. The SPARK saliva samples cluster with other samples from oral communities. Points in gray are samples from Human Microbiome Project sites not listed in the legend. **c**, SPARK samples on a UMAP (Uniform Manifold Approximation and Projection) generated from the first 20 principal components of the abundance matrix for the 439 most prevalent species fall on a gradient with respect to host age (color bar on right). **d**, The 439 most prevalent species on a UMAP generated from their loadings onto the first 20 principal components of the abundance matrix fall on a gradient with respect to correlation of relative species abundance with host age (color bar on right). **e**, Among children in SPARK (n = 5,760), a UMAP using the same 20 principal components as in **b** shows minimal stratification by autism spectrum disorder case status. **f,g**, Scatter plots of SPARK participants along axes of top genetic principal components. Individual dots are colored according to self-reported race/ethnicity for individuals who reported a single race/ethnicity. **h-k**, Change in abundances of *Prevotella melaninogenica*, *Streptococcus mitis*, *Porphyromonas endodontalis*, and *Neisseria cinerea* over the age range found in the SPARK cohort (truncated at 60 years old due to limited sampling of elderly individuals). Relative abundances were inverse normal rank-transformed (y-axis). Red curves indicate medians; shading indicates interquartile regions.

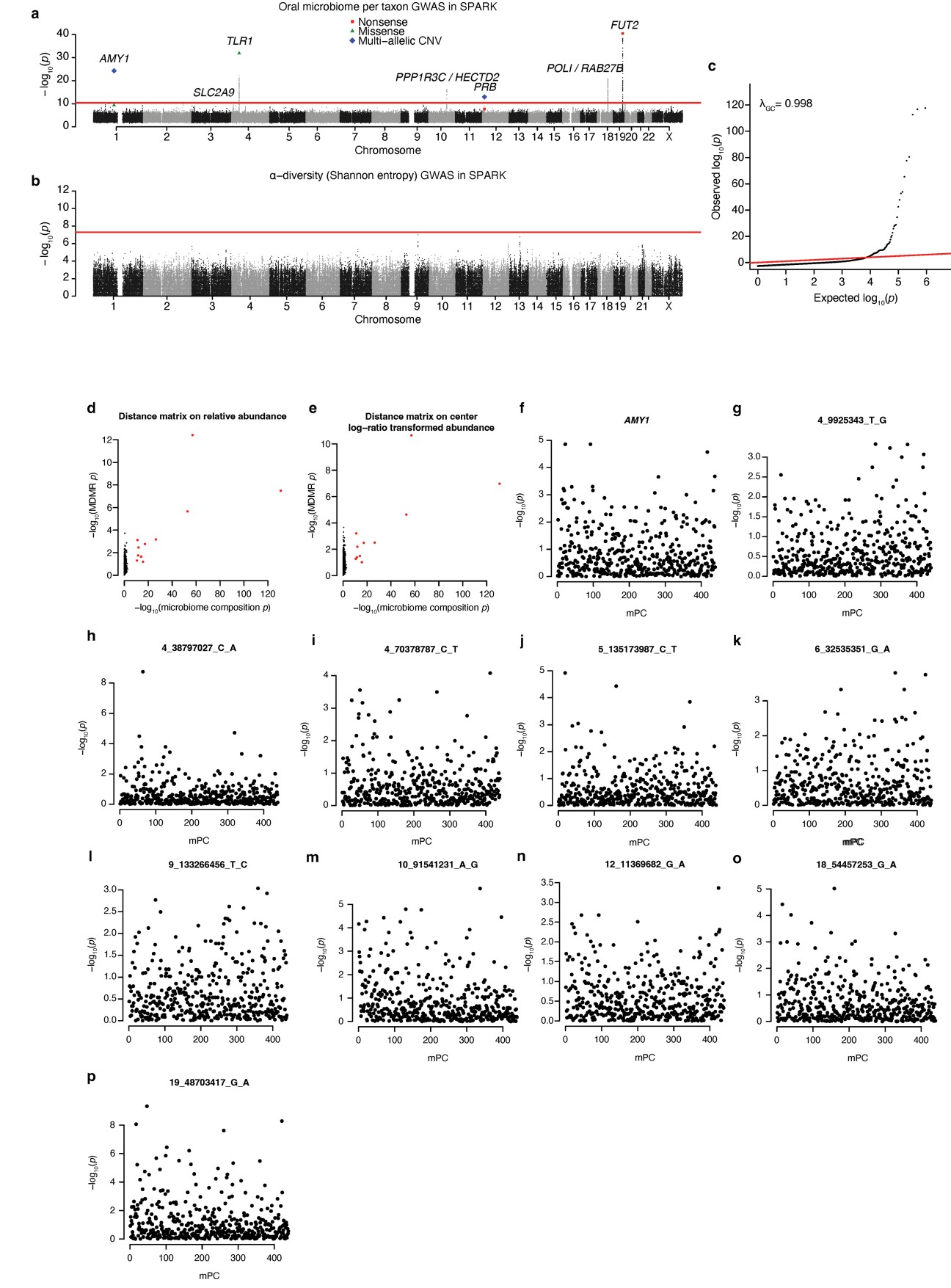

**Extended Data Fig. 2** | See next page for caption.

**Extended Data Fig. 2 | Associations of taxon relative abundances, α-diversity, and microbial composition principal components with human genetic variants. a**, Genome-wide associations with relative abundances of 1,262 taxa observed in >10% of SPARK samples (n = 12,519). For each genetic variant, the most significant $p$-value is shown (across the 1,262 tests); the red line indicates the study-wide significance threshold ($p < 4.0 \times 10^{-11}$). Protein-altering variants and copy number variants of note are highlighted: nonsense (red squares), missense (green triangles), and multi-allelic CNVs (blue diamonds). **b**, Genome-wide associations with α-diversity (Shannon entropy) in SPARK. **c**, Quantile-quantile plot of $p$-values computed by our mPC-based test for associations between human genetic variants and oral microbiome composition (Fig. 2b). The genomic inflation factor $\lambda_{GC}$ was calculated as the median chi-square statistic divided by $F^{-1}(0.5)$, where $F^{-1}(x)$ is the inverse cumulative distribution function for a $\chi^2_{439}$ random variable. **d**, Associations of 11 lead variants identified by the mPC-based test (red) and 1,000 randomly selected variants (black) with dissimilarity of relative abundances for the 439 most prevalent species using multivariate distance matrix regression (MDMR, y-axis) as compared with our mPC-based test (x-axis). **e**, Analogous to d, for dissimilarity after applying the centered log-ratio transform to relative abundance measurements. **f**, Associations of *AMY1* copy number (y-axis) with each of the 439 individual microbial principal components (x-axis). **g-p**, Analogous to f, for the other 10 lead variants identified by the mPC-based GWAS. P-values were computed using two-sided linear mixed models (**a**,**b**,**f-p**), one-sided chi-squared test (**c**; x-axis of **d**,**e**), or one-sided multivariate distance matrix regression (y-axis of **d**,**e**).

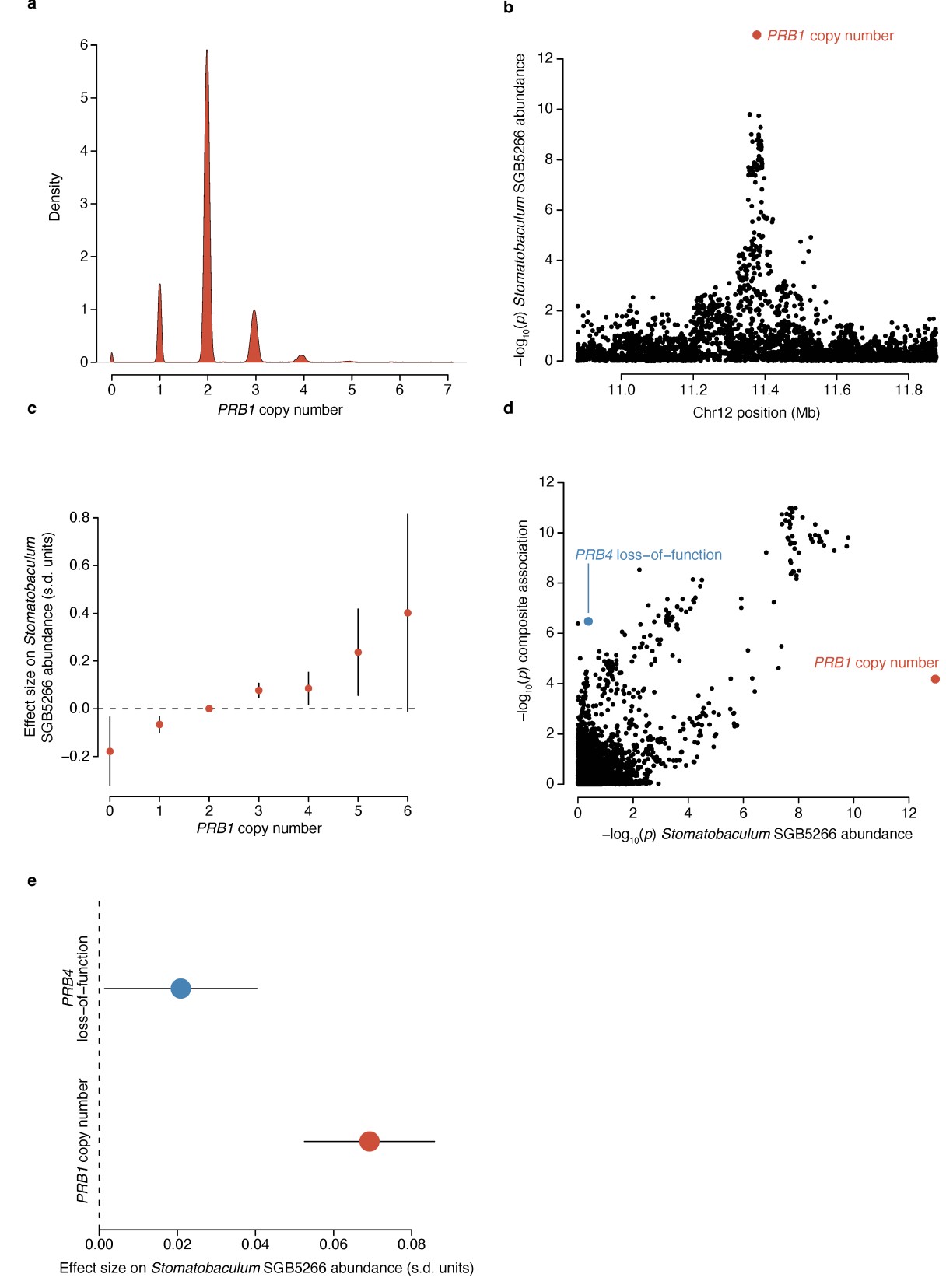

**Extended Data Fig. 3 |** See next page for caption.

**Extended Data Fig. 3 | Complex associations of human genetic variation at the *PRB* locus with oral microbiome phenotypes. a**, Diploid copy number of *PRB1* estimated for SPARK participants from WGS read-depth (n = 12,519). **b**, Associations of genetic variants at the *PRB* locus with relative abundance of *Stomatobaculum* SGB5266. The association of *PRB1* copy number is highlighted (red point). **c**, Allelic series of effect sizes of *PRB1* diploid copy numbers on relative abundance of *Stomatobaculum* SGB5266 (in s.d. units, n = 12,517). **d**, Partial colocalization of associations of genetic variants at the *PRB* locus with *Stomatobaculum* SGB5266 relative abundance (-log10(p), x-axis) and oral microbiome composition (-log10(p), y-axis). *PRB1* copy number (red point) and a common *PRB4* loss-of-function variant (blue point) appear to deviate from a generally concordant pattern of associations, likely reflecting multiple causal effects. **e**, Effect sizes on *Stomatobaculum* SGB5266 relative abundance for each additional copy of *PRB1* compared to each functional copy of *PRB4* (n = 12,519). Error bars, 95% CIs in all panels. P-values were computed using two-sided linear mixed models (**b**,**d**).

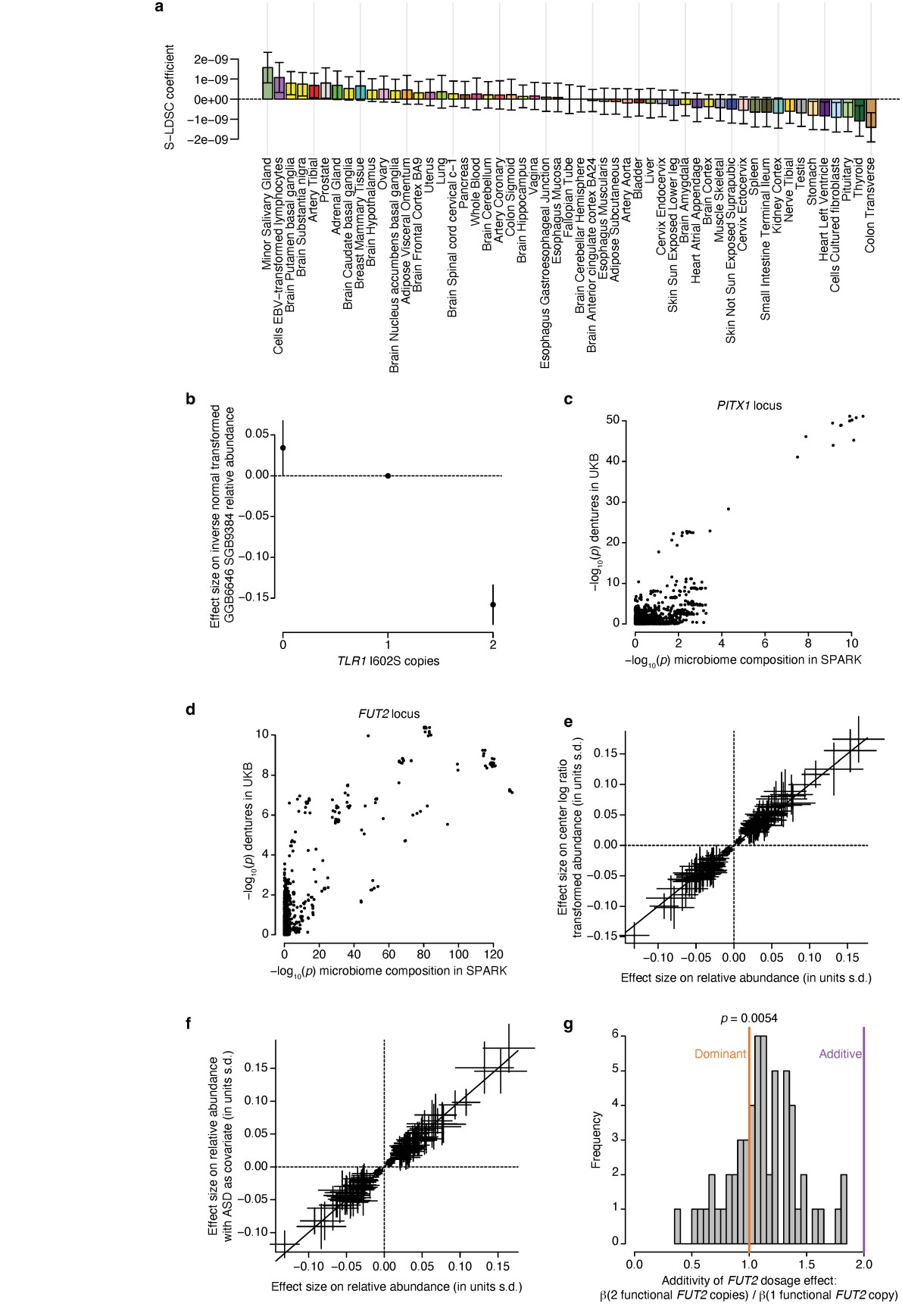

**Extended Data Fig. 4** | See next page for caption.

**Extended Data Fig. 4 | Additional information about associations between human genetic variants and oral microbial abundance phenotypes.**
**a**, Stratified LD-score regression coefficients (which quantify evidence of enrichment of heritability in regions surrounding genes with tissue-specific expression) from S-LDSC analysis of chi-squared statistics from the oral microbiome composition GWAS (n = 12,519). Heritability enrichment was evaluated for genomic regions defined by 53 tissues from the Genotype-Tissue Expression Project (GTEx). **b**, Effect sizes (in s.d. units) on relative abundance of SGB9384 for individuals homozygous for either allele of the *TLR1* I602S missense variant relative to heterozygotes (n = 12,519). **c**, Colocalization of genetic associations at the *PITX1* locus with dentures use (-log10(p), y-axis) and oral microbiome composition (-log10(p), x-axis). **d**, Analogous to c, for the *FUT2* locus. **e**, Consistency of effect sizes for 167 significant variant-species abundance associations computed with or without applying the centered log-ratio transform to relative abundance measurements. **f**, Consistency of effect sizes for 167 significant variant-species abundance associations computed with or without including ASD status as a covariate. **g**, Distribution of relative effect sizes on microbial species abundances for individuals homozygous for the *FUT2* secretor allele relative to heterozygotes, across microbial species whose relative abundance associated with secretor status (FDR < 0.05). Error bars, 95% CIs in all panels. P-values were computed using one-sided chi-squared test (x-axis of **c**,**d**), two-sided linear regression (y-axis of **c**,**d**) or one-sided t-test (**g**).

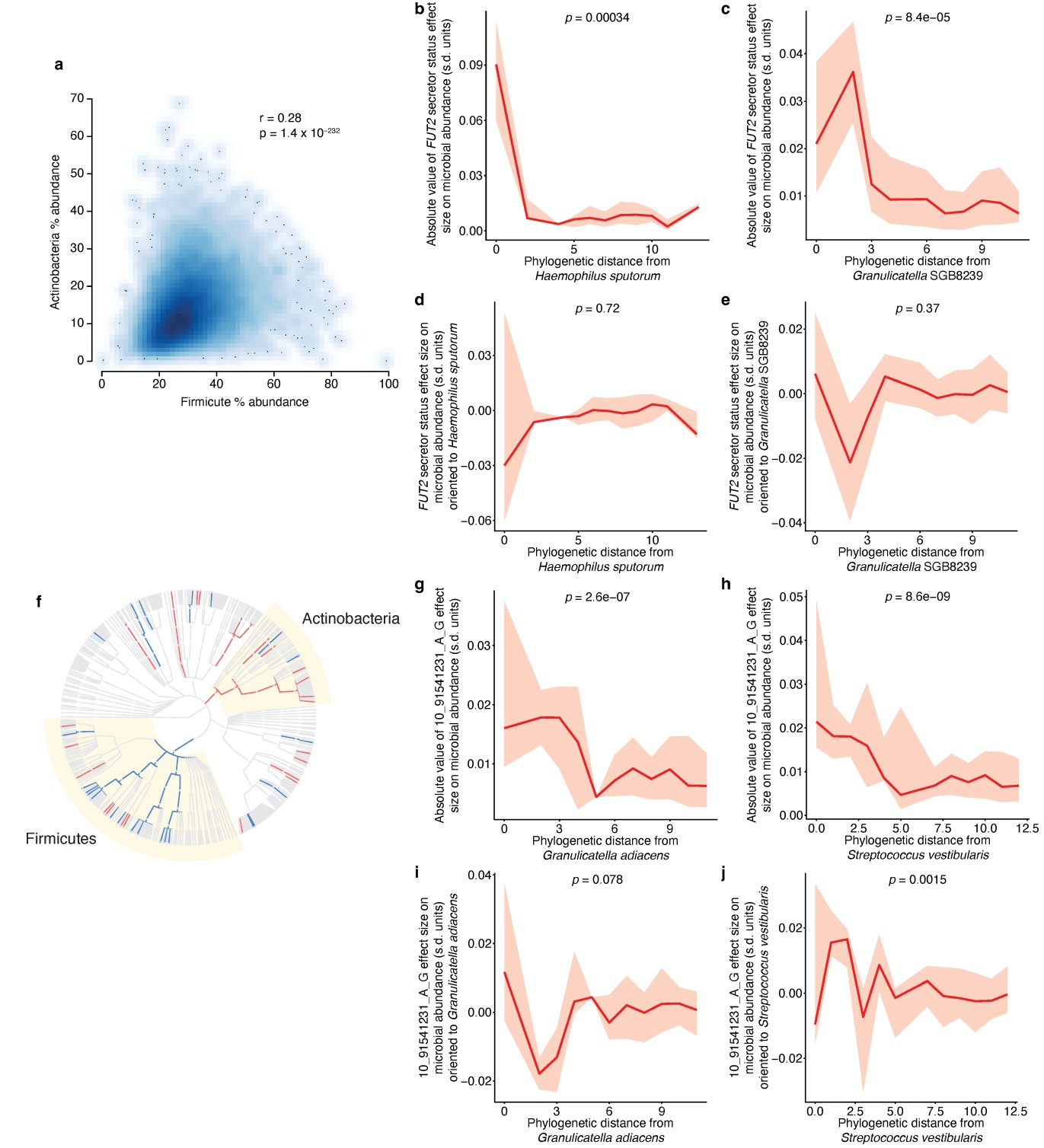

**Extended Data Fig. 5 | Phyletic stratification of effects of two human genetic variants on relative abundances of oral microbial species. a**, Positive correlation of relative abundance of the Actinobacteria phylum relative to the Firmicute phylum across saliva samples from SPARK participants (n = 12,519). **b**, Unsigned effect sizes for associations of secretor status (based on *FUT2* W154X, using a recessive model) with relative abundances of the 439 most prevalent microbial species (y-axis) versus phylogenetic distance from *Haemophilus sputorum* (x-axis). **c**, Analogous to **b**, for *Granulicatella* SGB8239. **d**, Signed effect size of associations of secretor status with relative abundances of microbial species (oriented relative to the effect direction for *H. sputorum*) (y-axis) versus phylogenetic distance from *H. sputorum* (x-axis). **e**, Analogous to **d**, for *G.* SGB8239. **f**, Microbial taxa whose abundance associated with the index variant at the *HECTD2/PPP1R3C* locus (FDR < 0.1) shown on the phylogenetic tree of 439 species (red, taxa whose relative abundances increased with the reference allele; blue, taxa whose relative abundances decreased with the reference allele). Two significantly-associated phyla (Firmicutes and Actinobacteria) are highlighted with yellow sectors. At the species level (outermost circle), dot sizes increase with statistical significance. **g**, Analogous to **b**, for *Granulicatella adiacens* with the *HECTD2/PPP1R3C* index variant. **h**, Analogous to **g**, for *Streptococcus vestibularis*. **i**, analogous to **d**, for *G. adiacens* with the *HECTD2/PPP1R3C* index variant. **j**, analogous to **i**, for *S. vestibularis*. For **b-e**, **g-j**, the red line indicates the median effect size, and the shaded region indicates the interquartile range. P-values were computed using two-sided Pearson's product-moment correlation (**a**) or two-sided linear regression (**b-e**, **g-j**).

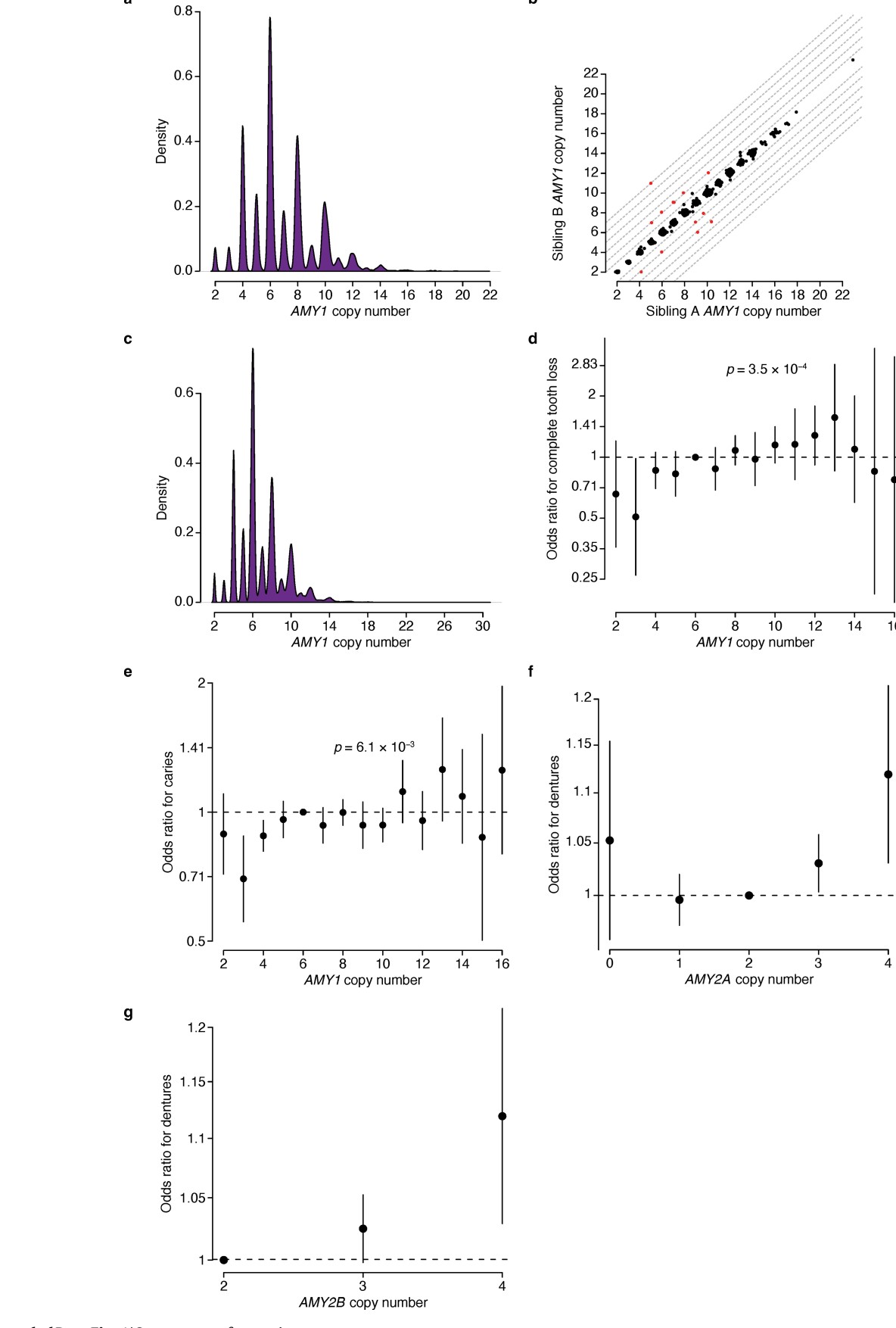

**Extended Data Fig. 6** | See next page for caption.

**Extended Data Fig. 6 | Genotyping and mutability of *AMY1* copy number and replication of associations with oral health. a**, Estimated diploid copy number of *AMY1* for SPARK participants (n = 12,519). **b**, Concordance of *AMY1* copy number estimates between 5,149 sibling pairs in the UKB cohort that share both haplotypes identical-by-descent (IBD2) in the region surrounding the amylase locus. Among the 5,149 IBD2 sibling pairs, 13 had copy number-discordant calls (red points) that tended to differ by two copies (11/13), likely corresponding to de novo duplication or deletion of a copy of the common structural cassette containing two *AMY1* genes in an inverted orientation to each other (Fig. 3a). Several IBD2 sibling pairs with *AMY1* copy number estimates that differed by close to 1 copy appeared more likely to reflect uncertainty in copy number estimates as they lacked a corresponding *AMY2A* duplication or deletion that would be expected to accompany a duplication or deletion of a single copy of *AMY1*. This gives an estimated germline mutation rate of $6.3 \times 10^{-4}$ mutations/meiosis ($[3.5 \times 10^{-4}, 11.1 \times 10^{-4}]$, 95% CIs, similar to recent estimates from haplotype coalescent trees[46]), exceeding the mutation rate of most short tandem repeats[88]. **c**, Analogous to **a**, for the AoU cohort (n = 245,377). **d**, Odds ratios for risk of complete tooth loss in AoU (n = 230,002) per *AMY1* diploid copy number. **e**, Analogous to **d**, for having caries. **f**, Odds ratios for risk of dentures use in UKB (n = 418,039) per *AMY2A* diploid copy number. **g**, Analogous to **f**, for *AMY2B* diploid copy number. Error bars, 95% CIs in all panels. P-values were computed using two-sided linear regression (**d**,**e**).

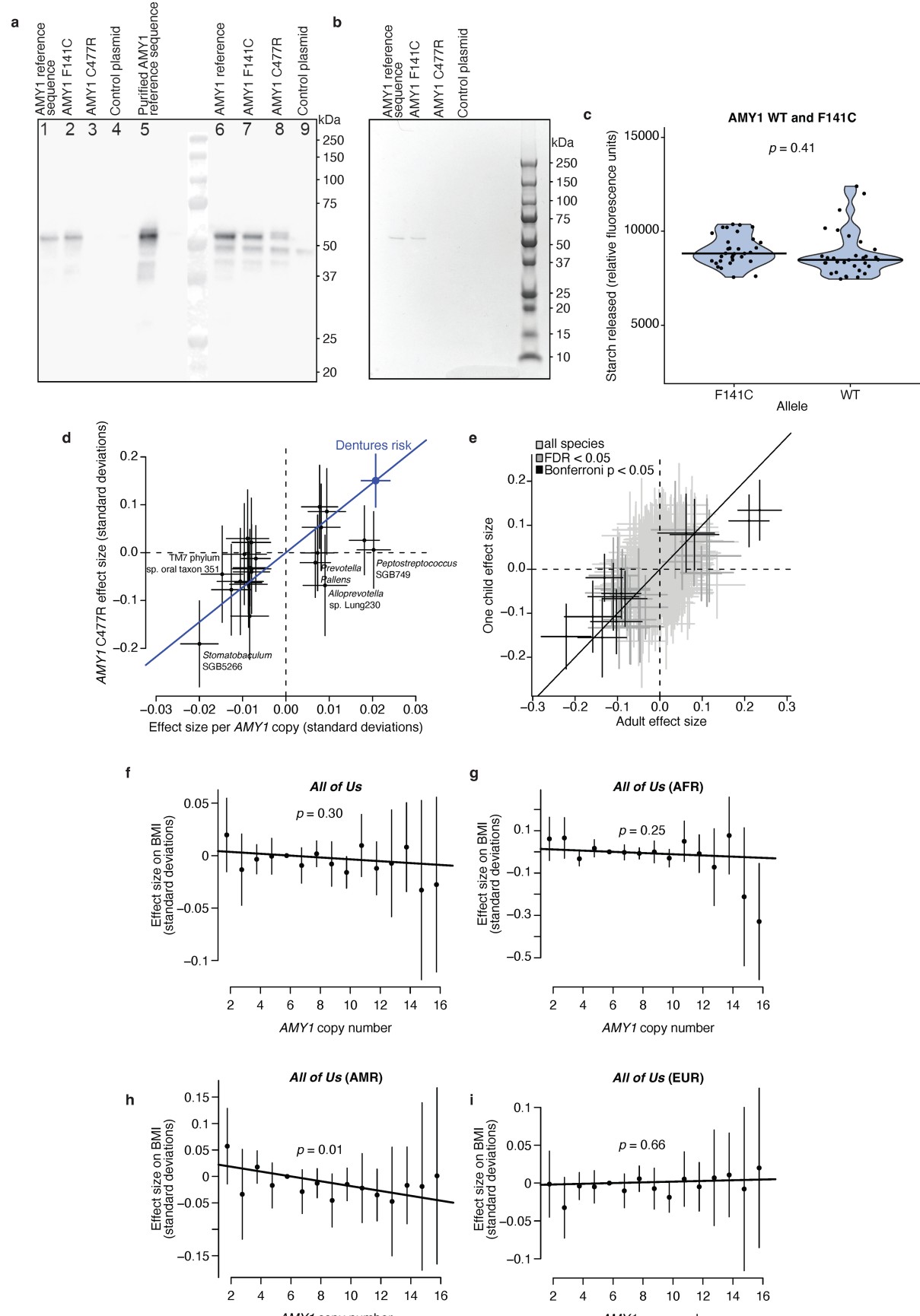

**Extended Data Fig. 7** | See next page for caption.

**Extended Data Fig. 7 | Functional assays of AMY1 coding variation, consistency of microbiome associations, and lack of copy number associations with BMI in AoU. a**, Immunoblotting for amylase transgenically expressed in HEK293T cells. Amylase (~56kD) is found in the supernatant of cells transfected with the reference sequence of *AMY1* (lane 1) and *AMY1* F141C (lane 2), but not *AMY1* C477R (lane 3) or a control plasmid (lane 4). The glycogen-purified protein is recognized by anti-amylase (lane 5). Amylase is found in the lysate of cells transfected with the reference sequence of *AMY1* (lane 6), *AMY1* F141C (lane 7), and less abundantly in cells transfected with *AMY1* C477R (lane 8), but not a control plasmid (lane 9). Cross-reactive protein (~50kD) can be seen in the lysate of all samples and smaller AMY1 fragments in lanes containing AMY1. These results were replicated in 3 independent transformations with similar results. **b**, Glycogen-purified supernatant from cells transfected with the reference sequence of *AMY1* (lane 1) and *AMY1* F141C (lane 2), but not *AMY1* C477R (lane 3) or a control plasmid (lane 4) contains a single protein band at ~56kD. These results were replicated in 3 independent transformations with similar results. **c**, Starch degradation is similar between equivalent mass dilutions of the reference amylase isoform and AMY1 F141C (n = 32 technical replicates for each allele). Starch degradation was measured as change in FAM relative fluorescence units over 30 min (y-axis) after addition of diluted AMY1 to quenched starch substrate. Centers, medians. **d**, Comparison of effect sizes for *AMY1* copy number versus *AMY1* C477R copy number on relative abundances of 16 bacterial species (from Fig. 3b) and on risk of dentures use (blue dot). For some species (e.g., *Stomatobaculum* SGB5266), the relative effect size of *AMY1* copy number versus *AMY1* C477R copy number on abundance is similar to this ratio for dentures use (blue line), whereas for others, it is not (e.g. *Prevotella pallens*). **e**, Concordance of effect size estimates for *AMY1* copy number on relative abundances of 439 microbial species in adults (x-axis) and unrelated children (y-axis). Species whose abundances associated significantly with *AMY1* copy number are indicated in darker gray (FDR < 0.05) or black (Bonferroni p < 0.05). **f**, Effect sizes on BMI per *AMY1* copy number in AoU (n = 219,879). The line drawn is the best fit across *AMY1* copy numbers. As a positive control, we confirmed that the BMI phenotype we tested (see Methods) associated strongly with the BMI-associated SNP rs1421085 at *FTO*[89] ($p = 3.72 \times 10^{-140}$). **g**, Analogous to **f**, for the African/African American ancestry subset of AoU participants (n = 49,296, $p = 0.25$). **h**, Analogous to **f**, for the American Admixed/Latino ancestry subset of AoU participants (n = 38,788, $p = 0.01$). **i**, Analogous to **f**, for the European ancestry subset of AoU participants (n = 122,577, $p = 0.66$). Error bars, 95% CIs in all panels. P-values were computed using two-sided t-test (**c**), two-sided linear mixed models (**e**), or two-sided linear regression (**f-i**).

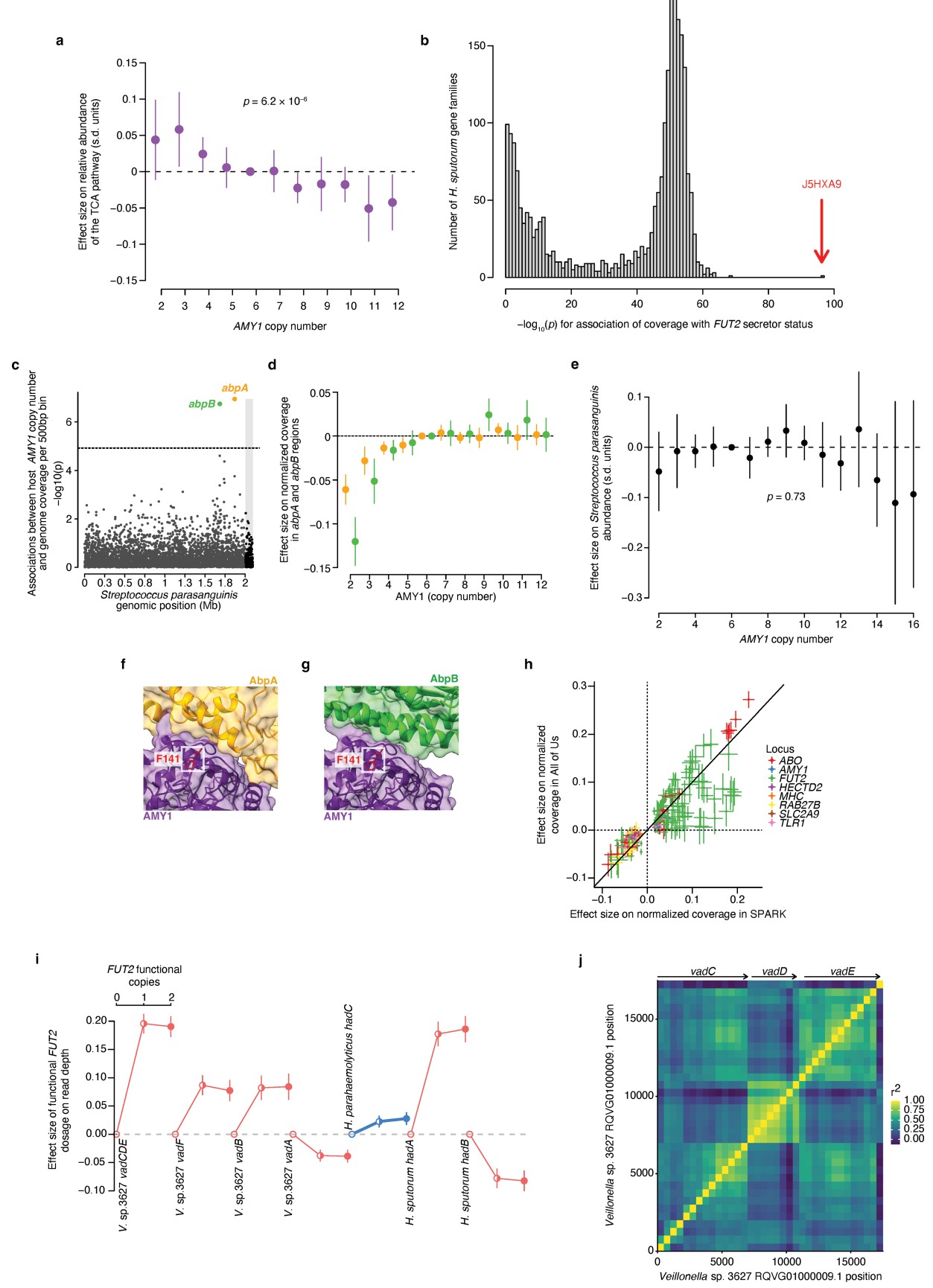

**Extended Data Fig. 8** | See next page for caption.

**Extended Data Fig. 8 | Pathway abundance associations and details of gene dosage associations. a**, Effect sizes of *AMY1* diploid copy numbers on the relative abundance of reads mapping to genes in the tricarboxylic acid cycle (TCA) pathway across microbial species. **b**, Associations (-log10(p), x-axis) of the relative abundance of reads mapping to each of 2,416 *Haemophilus sputorum* gene families with FUT2 secretor status. Read depth measurements were inverse normal transformed across samples for each gene. Gene families generally fell into two modes, one group not associated with FUT2 secretor status and another that associated at significance similar to that of the relative abundance of *H. sputorum*. One outlier (indicated with the red arrow) associated much more strongly and corresponds to a gene encoding a protein with trimeric autotransporter adhesin domain annotations (UniParc ID J5HXA9). **c**, Associations of deletions in the genome of *Streptococcus parasanguinis* (as estimated by normalized coverage) with *AMY1* copy number (n = 12,340). Shading indicates the two assembled contigs of the reference genome. The two significant associations (FDR < 0.01) overlap genes encoding amylase-binding proteins, *abpA* and *abpB*. **d**, Allelic series of effect sizes of *AMY1* copy number on normalized coverage (n = 12,026) in the 500 bp bins overlapping *abpA* (orange) and *abpB* (green). **e**, Effect sizes of *AMY1* diploid copy numbers on relative abundance of *Streptococcus parasanguinis* (n = 12,487). **f**, Protein-protein interaction between human AMY1 (purple) and *S. parasanguinis* AbpA (orange) predicted by AlphaFold3. AMY1 residue F141 is highlighted in red.

**g**, Analogous to **f**, for *S. parasanguinis* AbpB (green). **h**, Replication of 208 associations between human genetic variants and normalized read-depth measurements in 500 bp bins of microbial genomes in the AoU cohort (comprised of individuals age 18 or older). Effect sizes estimated in AoU participants (n = 10,000, y-axis) are plotted against effect sizes estimated in SPARK (n = 12,519, x-axis). Dots correspond to the 208 associations and are colored according to the human genetic locus involved, as in Fig. 4b. **i**, Effect sizes of *FUT2* W154X genotype on normalized coverage at bacterial genes encoding proteins with YadA-like (adhesin) domain annotations (n = 7419, *Veillonella* sp. 3627; n = 8153, *Haemophilus sputorum*; n = 7456, *Haemophilus parahaemolyticus*). Colors indicate the effect direction of *FUT2* genotype on the relative abundance of each species (red, increasing with functional copies of *FUT2*; blue, decreasing with functional copies of *FUT2*). **j**, Correlation matrix of normalized coverage in the region of the *V.* sp. 3627 genome surrounding *vadD* and *vadE* revealed a linked deletion to *vadE* that contained *vadC* and nearly passed FDR < 0.01 ($p = 3.32 \times 10^{-6}$) in association with *FUT2* W154X genotype, causing samples with *vadE* to tend to also contain *vadC* (Fig. 6d) and possibly suggestive of the event introducing *vadD* occurring after the one that produced *vadC* and *vadE*. Arrows indicate *vad* gene locations. Error bars, 95% CIs in all panels. P-values were computed using two-sided linear mixed model (**a**,**e**) or two-sided linear regression (**b**,**c**). Effect sizes were computed with two-sided linear regression (**h**,**i**).

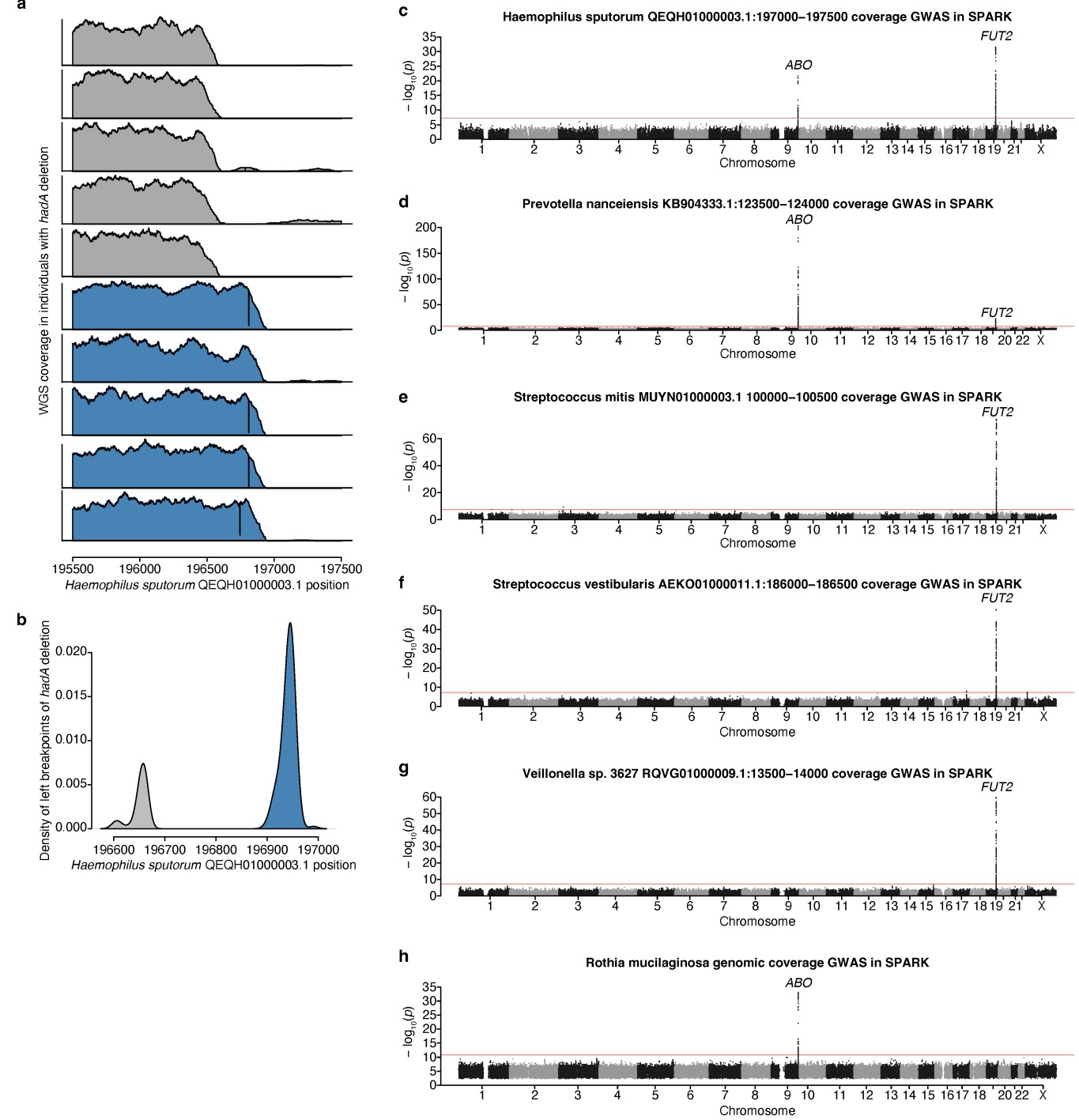

**Extended Data Fig. 9 | Left breakpoints of deletions in the genome of *Haemophilus sputorum* that contain the *hadA* gene are bimodal.**
**a**, Coverage of the QEQH01000003.1 contig in the *Haemophilus sputorum* genome for individuals with distinct left breakpoints of deletions containing *hadA*. **b**, Distribution of left breakpoints for the 200 individuals with highest genomic coverage of *H. sputorum* who had strong evidence of the deletion containing *hadA* (<0.1 median normalized coverage at the strongest associated bin). For each individual, mosdepth was run to measure per-base depth, and left and right breakpoints were identified as the first or last base with 10 consecutive zeros, respectively. **c**, Genome-wide associations for normalized read-depth in the 500 bp bin of *Haemophilus sputorum* QEQH01000003.1:

197000-197500. **d**, Analogous to c, but for *Prevotella nanceiensis* KB904333.1: 123500-124000. **e**, Analogous to c, but for *Streptococcus mitis* MUYN01000 003.1:100000-100500. **f**, Analogous to c, but for *Streptococcus vestibularis* AEKO01000011.1:186000-186500. **g**, Analogous to c, but for *Veillonella* sp. 3627 RQVG01000009.1:13500-14000. **h**, Genome-wide associations for normalized read-depth in 500 bp bins spanning the *Rothia mucilaginosa* genome. For each human genetic variant, the most significant *p*-value is shown (across all 500 bp bins), with the red line indicating the study-wide significance threshold ($p < 1.5 \times 10^{-11}$). P-values were computed using two-sided linear mixed models (**c-h**).

**Extended Data Table 1 | Eleven loci at which human genetic variants associate with oral microbiome composition**

| Locus | Lead variant | rsID | MAF | $p_{microbiome}$ | $p_{dentures}$ | Other information |
|---|---|---|---|---|---|---|
| *AMY1* | *AMY1* copy number | n/a | n/a | $1.5 \times 10^{-53}$ | **$5.9 \times 10^{-35}$** | Determines *AMY1* gene dosage |
| *SLC2A9* | 4_9925343_T_G | rs13129697 | 0.32 | $2.9 \times 10^{-12}$ | **0.021** | *SLC2A9* eQTL and sQTL; replicates oral microbiome association[4] |
| *TLR1* | 4_38797027_C_A | rs5743618 | 0.4 | $6.2 \times 10^{-18}$ | 0.99 | *TLR1* I602S missense; affects immune response[38,39] |
| *SMR3A* / *SMR3B* | 4_70378787_C_T | rs28612397 | 0.12 | $1.4 \times 10^{-12}$ | **$6.0 \times 10^{-3}$** | *SMR3A* and *SMR3B* eQTL; *SMR3B* and *MUC7* sQTL |
| *PITX1* | 5_135173987_C_T | rs3749751 | 0.41 | $3.0 \times 10^{-11}$ | **$7.5 \times 10^{-52}$** | Variant has unknown function; *Pitx1* knockout causes jaw malformation[42] |
| *HLA* | 6_32535351_G_A | rs112652539 | 0.4 | $2.2 \times 10^{-16}$ | **$7.1 \times 10^{-13}$ (rs9271236; $r^2$=0.73)** | eQTL for six *HLA* class II genes |
| *ABO* | 9_133266456_C_T | rs2519093 | 0.18 | $9.4 \times 10^{-15}$ | **0.022** | Tags A1 blood group allele[40] |
| *PPP1R3C* / *HECTD2* | 10_91541231_A_G | rs12260868 | 0.33 | $8.8 \times 10^{-58}$ | **0.048** | *HECTD2* and *PPP1R3C* eQTL |
| *PRB1-PRB4* | 12_11369682_G_A | rs7966710 | 0.25 | $1.1 \times 10^{-11}$ | 0.88 (rs7977399; $r^2$=0.81) | *PRB2* eQTL; *PRB1* mCNV and *PRB4* R39X also associate with oral microbiome composition (**Supplementary Note 1**) |
| *POLI* / *RAB27B* | 18_54457253_G_A | rs17559023 | 0.08 | $4.2 \times 10^{-27}$ | 0.75 | *RAB27B*, *POLI*, and *C18orf54* eQTL; *POLI* sQTL |
| *FUT2* | 19_48703417_G_A | rs601338 | 0.45 | $1.6 \times 10^{-131}$ | **$7.4 \times 10^{-8}$** | *FUT2* W154X; primary determinant of secretor status[41] |

MAF, minor allele frequency. $p_{microbiome}$, p-value for association with oral microbiome composition in SPARK. $p_{dentures}$, p-value for association with dentures risk in UKB (with a proxy variant indicated in parentheses for two variants); nominally significant associations ($p_{dentures} < 0.05$) are indicated in bold.

# Reporting Summary

## Statistics

For all statistical analyses, confirm that the following items are present in the figure legend, table legend, main text, or Methods section.

| n/a | Confirmed | |
|---|---|---|
| ☐ | ☒ | The exact sample size (*n*) for each experimental group/condition, given as a discrete number and unit of measurement |
| ☐ | ☒ | A statement on whether measurements were taken from distinct samples or whether the same sample was measured repeatedly |
| ☐ | ☒ | The statistical test(s) used AND whether they are one- or two-sided<br>*Only common tests should be described solely by name; describe more complex techniques in the Methods section.* |
| ☐ | ☒ | A description of all covariates tested |
| ☐ | ☒ | A description of any assumptions or corrections, such as tests of normality and adjustment for multiple comparisons |
| ☐ | ☒ | A full description of the statistical parameters including central tendency (e.g. means) or other basic estimates (e.g. regression coefficient) AND variation (e.g. standard deviation) or associated estimates of uncertainty (e.g. confidence intervals) |
| ☐ | ☒ | For null hypothesis testing, the test statistic (e.g. *F*, *t*, *r*) with confidence intervals, effect sizes, degrees of freedom and *P* value noted<br>*Give P values as exact values whenever suitable.* |
| ☒ | ☐ | For Bayesian analysis, information on the choice of priors and Markov chain Monte Carlo settings |
| ☒ | ☐ | For hierarchical and complex designs, identification of the appropriate level for tests and full reporting of outcomes |
| ☐ | ☒ | Estimates of effect sizes (e.g. Cohen's *d*, Pearson's *r*), indicating how they were calculated |

*Our web collection on statistics for biologists contains articles on many of the points above.*

## Software and code

Policy information about availability of computer code

| Data collection | CFX Manager (v.3.1) as associated with CFX384 Real-Time PCR Detection System was used to collect fluorescence measurements for amylase enzyme activity. |
|---|---|
| Data analysis | The following publicly available software resources were used: MetaPhlAn (v.4.0.6, http://segatalab.cibio.unitn.it/tools/metaphlan/index.html), DeepVariant (v.1.3.0, https://github.com/google/deepvariant), GLnexus (v.1.4.1, https://github.com/dnanexus-rnd/GLnexus), HUMAnN (v.3.8, https://huttenhower.sph.harvard.edu/humann), GraPhlAn (v.1.1.3, http://segatalab.cibio.unitn.it/tools/graphlan/index.html), mosdepth (v.0.3.6, https://github.com/brentp/mosdepth), bowtie (v.2.5.1, https://bowtie-bio.sourceforge.net/bowtie2/index.shtml), bcftools (v.1.14, http://www.htslib.org/), samtools (v.1.15.1, http://www.htslib.org/), plink (v.1.90b6.26 and v.2.00a3.7,https://www.cog-genomics.org/plink/), BOLT-LMM (v.2.4.1, https://alkesgroup.broadinstitute.org/BOLT-LMM/), qqman (v.0.1.8, https://cran.r-project.org/web/packages/qqman/index.html), MDMR (v.0.5.2, https://cran.r-project.org/web/packages/MDMR/index.html), bedtools (v.2.27.1, https://bedtools.readthedocs.io/en/latest/), AlphaFold3 (v3, https://alphafoldserver.com/), and ChimeraX (v.1.9, https://www.cgl.ucsf.edu/chimerax/). Custom code used to generate results in this study is available via Zenodo at 10.5281/zenodo.14559458 |

For manuscripts utilizing custom algorithms or software that are central to the research but not yet described in published literature, software must be made available to editors and reviewers. We strongly encourage code deposition in a community repository (e.g. GitHub). See the Nature Portfolio guidelines for submitting code & software for further information.

## Data

Policy information about availability of data

All manuscripts must include a data availability statement. This statement should provide the following information, where applicable:

- Accession codes, unique identifiers, or web links for publicly available datasets
- A description of any restrictions on data availability
- For clinical datasets or third party data, please ensure that the statement adheres to our policy

The following data resources are available by application: UKB (http://www.ukbiobank.ac.uk/), All of Us Research Program (https://allofus.nih.gov/), and SFARI SPARK (https://www.sfari.org/resource/spark/). Relative abundances of species in the oral microbiome will be returned to SFARI for release upon request. The following data resources are publicly available: Human Microbiome Project (https://hmpdacc.org/), human reference genome build GRCh38 (https://ftp.1000genomes.ebi.ac.uk/vol1/ftp/technical/reference/GRCh38_reference_genome/), MetaPhlAn vOct22 reference database (http://cmprod1.cibio.unitn.it/biobakery4/metaphlan_databases/), TOPMed-r3 imputation panel (https://imputation.biodatacatalyst.nhlbi.nih.gov/), LD score resources (https://alkesgroup.broadinstitute.org/LDSCORE/), NCBI GenBank (https://www.ncbi.nlm.nih.gov/genbank/ ), and NCBI Conserved Domain Database (https://www.ncbi.nlm.nih.gov/Structure/cdd/cdd.shtml).

## Research involving human participants, their data, or biological material

Policy information about studies with human participants or human data. See also policy information about sex, gender (identity/presentation), and sexual orientation and race, ethnicity and racism.

| Reporting on sex and gender | Sex was used as a covariate in several analyses, but no values directly pertaining to sex are reported. |
|---|---|
| Reporting on race, ethnicity, or other socially relevant groupings | For UK Biobank, using the top 20 ancestry principal components, a subset of individuals that fell within a Euclidean distance (centered at the mean values of each PC for individuals who self-identified as "white") capturing 99% of individuals who self-identified as "white" were used for phenotype associations. For All of Us, analyses were performed either on the entire cohort, or by restricting to released genetically-predicted ancestry as noted. For SPARK, all individuals were included in analyses without restricting to any ancestry.  For all analyses except associations with microbial gene dosage, ancestry principal components were included as covariates in genetic associations. |
| Population characteristics | UK Biobank is a cohort of approximately 500,000 individuals across the United Kingdom between 40 and 69 years of age at time of recruitment (Sudlow et al. 2015 PLOS Medicine). For phenotype associations in the UK Biobank cohort, age, age squared, sex, genotype array, assessment center, and top 20 genetic ancestry PCs were used as covariates. All of Us is a cohort of approximately 245,000 individuals with WGS available (at time of analysis) across the United States older than 18 years of age at time of recruitment (The All of Us Research Program Investigators 2019 N Engl J Med). For oral health associations in the All of Us cohort, age, age squared, sex, and the top 16 genetic ancestry principal components (from ancestry_preds.tsv) were used as covariates. For BMI associations in the All of Us cohort, only genetic ancestry principal components were used as covariates as age and sex had already been residualized out. SFARI SPARK is a cohort of approximately 160,000 families with at least one child with autism spectrum disorder, where 12,519 individuals (at time of analysis) have WGS from saliva available (SPARK Consortium 2018 Neuron). For microbiome associations in the SPARK SFARI cohort, sequencing batch, age, age squared, square root of age, sex, percent of mapped reads, and the top 10 genetic ancestry principal components were used as covariates. |
| Recruitment | Individuals and biosamples were not obtained for this study and their recruitment is as described in prior publications (cited in current work). |
| Ethics oversight | Individuals and biosamples were not obtained for this study and local IRBs at each institution approved the collections and patient-consent materials, as described in the earlier papers on these cohorts (cited in current work). North West-Haydock Research Ethics Committee gave ethical approval for UK Biobank data collection and availability under reference 16/NW/0274. Western IRB of Wayne State University gave ethical approval for Simons Foundation Autism Research Initiative (SPARK) data collection and availability under protocol 20151664. The IRB of the All of Us Research Program gave ethical approval gave ethical approval for All of Us data collection and availability under protocol 2021-02-TN-001. The Office of Research Subject Protection (ORSP) of the Broad Institute waived ethical approval for this work, as this research on de-identified, previously-collected data was determined not to constitute human subjects research and did not require IRB review. Data from the UKB Resource were accessed under application number 40709 and from SFARI SPARK under application 3350.2. |

Note that full information on the approval of the study protocol must also be provided in the manuscript.

# Field-specific reporting

Please select the one below that is the best fit for your research. If you are not sure, read the appropriate sections before making your selection.

☒ Life sciences ☐ Behavioural & social sciences ☐ Ecological, evolutionary & environmental sciences

For a reference copy of the document with all sections, see nature.com/documents/nr-reporting-summary-flat.pdf

# Life sciences study design

All studies must disclose on these points even when the disclosure is negative.

| | |
|---|---|
| Sample size | Starting from 488,377 individuals in the UK Biobank SNP-array data set, individuals were excluded based on the following criteria: 36,008 were removed to drop one relative within pairs of close relatives with kinship coefficient > 0.0884, preferentially keeping individuals if they a) reported having dentures or b) reported not having dentures (i.e., had a non-missing dentures phenotype); 28,701 were removed for not having European genetic ancestry; 1,469 were removed for not having available TOPMed-imputed genotypes (including for chromosome X); 2,601 were removed for not having available WGS data; and 53 were removed for having withdrawn, leaving 419,545 available individuals for genetic association analyses.  For the binary oral health phenotypes (dentures use and bleeding gums), 418,039 had non-missing values. For the quantitative BMI z-score phenotype, 418,150 had non-missing values.<br><br>For the All of Us cohort, 245,377 samples were genotyped for AMY1 copy number from available WGS data and these were then filtered to an unrelated subset of samples (iteratively dropping one individual per related pair with kinship score > 0.1, from relatedness_flagged_samples.tsv). 230,002 individuals had non-missing values for the oral health phenotypes and 219,879 individuals had non-missing values for the BMI z-score phenotype. For replication of microbial gene dosage associations with human genetic variants a random set of 10,000 samples with saliva as the biosample type were chosen.<br><br>For the SFARI SPARK cohort, all 12,519 samples with available WGS data were used for all analyses unless otherwise specified (ex. subsets used with sufficient genomic read depth coverage of a specific microbial species).<br><br>In all cases except for replication of microbial gene dosage associations in All of Us, no sample-size calculation was done to predetermine sample size and the maximum number of available samples were used. For oral health associations (UK Biobank), we expected that the association would be sufficiently powered to allow for associating AMY1 copy number with dentures risk given nearby variants reached genome-wide significance and low r2 (<0.2) between AMY1 copy number and any biallelic tag variants. Additionally, we expected reasonable power to find evidence of colocalized dentures use associations with microbiome composition given the large number of genome-wide significant loci (n=47) seen in a previous GWAS for dentures risk. For BMI associations, we expected reasonable power to replicate previously reported associations with AMY1 copy number, given each of our cohorts (UK Biobank and All of Us) were nearly two orders of magnitude larger than the largest where a significant relationship was observed.  For oral microbiome associations, we expected comparable power to find significantly associated human loci given results from several similarly sized gut microbiome association studies (n=8-16k). For replication of microbial gene dosage associations in All of Us, we chose a sample size of 10,000 to limit computational expense while approximating power of the SFARI SPARK cohort. For enzymatic assays of amylase isoforms, we expected 32 replicates to be sufficient to observe an effect sufficient to explain the genotypic associations (22.4- and 7.3-fold). |
| Data exclusions | Established QC metrics were used to exclude some samples, genotypes, or sequencing data for analysis as described in previously published studies (cited in the current work). Samples from individuals in UK Biobank, All of Us, and SFARI SPARK that requested to be withdrawn at the time of analysis were excluded. |
| Replication | For oral health phenotypes, All of Us (complete tooth loss, caries) served as independent replicate for association with AMY1 copy number as first performed in UK Biobank (dentures use), where caries was previously reported to have high genetic correlation with dentures use. Additionally, although not genome-wide significant, the bleeding gums phenotype in UK Biobank also served as a replication of the AMY1 copy number allelic series with effects from missense variants (F141C, C477R).<br><br>For BMI z-score phenotype, All of Us served as an independent replicate for association with AMY1 copy number as first performed in UK Biobank. Additionally, the same lack of association with AMY1 copy number seen in each genetically-predicted ancestry of All of Us serve as confirmations of non-ancestry specific trends.<br><br>For microbiome composition associations, the colocalization of individually significant microbial species at the same human genetic loci (even at a (taxa)x(human genetic variants) level of Bonferroni correction) each replicate the overall pattern of association, but also often resolved to the same lead index variant and pattern of association (Fig. 2, 3 and Extended Data Fig. 2, 3). Additionally, the comparison of associations for species relative abundance in adults and children separately (performed to assess the plausibility of reverse causality from dentures use) serve as additional replicates of the human genetic effect. The relative effect sizes of F141C and C477R on the relative abundance of different microbial species also affected by AMY1 copy number (Fig. 3i) further replicates these variants as exerting some phenotypic function equivalent to additional copies of AMY1.<br><br>For microbial gene dosage associations, the effects observed in the SFARI SPARK cohort were replicated in an independent set of 10,000 samples from the All of Us cohort (Extended Data Fig. 9c)<br><br>For in vitro amylase enzymatic assays, purified protein from reference sequence and F141C AMY1 isoforms was used in n=32 replicates of enzymatic activity, where all attempts were successful and included in Extended Data Figure 7c. |
| Randomization | For UK Biobank, samples were collected in batches at different assessment centers at locations across the United Kingdom and these were encoded as indicator covariates in phenotype-genotype associations. For SFARI SPARK, samples were collected in sequencing batches (WGS1 through WGS5), where these were encoded as indicator covariates. For All of Us, samples were sequenced in batches at different centers, where these were encoded as indicator covariates. For enzymatic assays, replicates were run as prepared in equally sized groups on plates, where plates were encoded as indicator covariates. No further randomization was done as all samples were used for each analysis. |
| Blinding | For all computational analyses, samples were listed with a randomized ID where association of measured genotype with trait (phenotype such as dentures use or relative abundance of a particular microbial species) was only done at the point of final statistical analysis. Blinding was not done for in vitro amylase enzymatic assay sample plating, as fluorescence quantification was performed simultaneously and identically for all samples on each plate. |

# Reporting for specific materials, systems and methods

We require information from authors about some types of materials, experimental systems and methods used in many studies. Here, indicate whether each material, system or method listed is relevant to your study. If you are not sure if a list item applies to your research, read the appropriate section before selecting a response.

### Materials & experimental systems

| n/a | Involved in the study |
|-----|----------------------|
| ☐ | ☒ Antibodies |
| ☐ | ☒ Eukaryotic cell lines |
| ☒ | ☐ Palaeontology and archaeology |
| ☒ | ☐ Animals and other organisms |
| ☒ | ☐ Clinical data |
| ☒ | ☐ Dual use research of concern |
| ☒ | ☐ Plants |

### Methods

| n/a | Involved in the study |
|-----|----------------------|
| ☒ | ☐ ChIP-seq |
| ☒ | ☐ Flow cytometry |
| ☒ | ☐ MRI-based neuroimaging |

## Antibodies

| | |
|---|---|
| Antibodies used | anti-Amylase Antibody (clone G-10, Santa Cruz Biotechnology, catalog no. sc-46657, lot no. G0324), Anti-mouse IgG, HRP-linked Antibody (Cell Signaling Technology, catalog no. 7076, lot no. 39) |
| Validation | The anti-Amylase Antibody (clone G-10) has been validated for use in Western blotting against human salivary amylase with some user-submitted Western blots (ex. Luti, S. et al. Chronic Training Induces Metabolic and Proteomic Response in Male and Female Basketball Players: Salivary Modifications during In-Season Training Programs. Healthcare (Basel) 11, 241 (2023).). We also confirm a) it recognizes a single band at the expected size (~56 kD) in glycogen-purified cell culture supernatant (EDF 6) which is expected to yield only amylase from previous work, and b) presence of this band in unpurified cell culture supernatant from cells transfected with a plasmid containing amylase coding sequence but not those without. We do note that there is a non-specific band at ~50kD present in the supernatant of cells transfected with a control plasmid not containing AMY1. The anti-mouse antibody (CST 7076) has been validated against CST primary antibodies in Western blots as indicated on the manufacturer's website. |

## Eukaryotic cell lines

Policy information about cell lines and Sex and Gender in Research

| | |
|---|---|
| Cell line source(s) | Lenti-X 293T (HEK293T clone) from Takara Bio USA (lot no. AIY00015, catalog no. 632180) |
| Authentication | Morphological match for type and in-house verification of SV40T antigen with genotyping PCR assay. No other standard authentication methods were performed (such as STR typing). |
| Mycoplasma contamination | Lack of mycoplasma contamination was done by Takara Bio USA as well as by members of receiving lab (McCarroll) |
| Commonly misidentified lines (See ICLAC register) | None were used in this study, HEK293T is a derivative of HEK and has not been listed as commonly misidentified. |

## Plants

| | |
|---|---|
| Seed stocks | *Report on the source of all seed stocks or other plant material used. If applicable, state the seed stock centre and catalogue number. If plant specimens were collected from the field, describe the collection location, date and sampling procedures.* |
| Novel plant genotypes | *Describe the methods by which all novel plant genotypes were produced. This includes those generated by transgenic approaches, gene editing, chemical/radiation-based mutagenesis and hybridization. For transgenic lines, describe the transformation method, the number of independent lines analyzed and the generation upon which experiments were performed. For gene-edited lines, describe the editor used, the endogenous sequence targeted for editing, the targeting guide RNA sequence (if applicable) and how the editor was applied.* |
| Authentication | *Describe any authentication procedures for each seed stock used or novel genotype generated. Describe any experiments used to assess the effect of a mutation and, where applicable, how potential secondary effects (e.g. second site T-DNA insertions, mosiacism, off-target gene editing) were examined.* |

