## [Peer Review File · Nature]

Human and bacterial genetic variation shape oral microbiomes and health

Corresponding Author: Mr Nolan Kamitaki

Version 1:

Reviewer comments:

Referee #1

(Remarks to the Author)

The paper by Kamitaki et al. focuses on the analysis of the relationship between host genetics and the oral microbiome. They analysed microbial and human reads in oral samples from 12,519 individuals and identified 439 common microbial species (present in >10%), which they linked to host genetic variants. Instead of the commonly used approach (testing the association of each genetic variant with each taxon), they used an alternative approach, by performing an association with overall microbial composition. More specifically, they created 439 PCs (corresponding to the number of taxa), associated each genetic variant with all PCs, and performed a meta-analysis across all PCs. I have several critical comments on the method, though I acknowledge that the associated loci point to relevant biological molecules.

Please see my comments below:

1. The methods and sample selection for this study are not clearly described. In the results section, the authors state that they used samples from the SPARK cohort and cite two references. It reads as if the data were generated in these previous studies. However, the numbers do not match: in this paper, the number of samples mentioned is: "Oral microbiomes in 12,519 individuals measured by saliva whole genome sequencing." In the two reference papers, two different sample sizes from SPARK are mentioned: 1. "SPARK cohort: N = 7812 individuals from 2025 US families, avg. family size = 3.9"; 2. "A total of 1,929 saliva samples from 646 families in the NY Genome Center Cohort of the SPARK collection." Concerns: If the authors used published SPARK data, the numbers do not match. Moreover, they should describe which samples came from which study and how they handled the batch effect. If the authors performed the sequencing as part of this study, then they need to add all missing information on sequencing: sample collection, DNA isolation methods, and sequence depth.
2. From previous publications, it seems that SPARK was a family-based design – many samples came from the same family. In this case, genetic analysis should take into account family structure; it is unclear if this was done and how.
3. Identifying 439 common bacteria is surprisingly high. How many taxa were identified in total (e.g., at >1% frequency)? Since this appears to be the first description of the corresponding dataset (see point 1), I expect the authors to present more descriptive statistics, including the total number of identified microbes and their frequencies.
4. Genetic analysis: the authors used a different method. Instead of running GWAS on microbial abundances or presence, they transformed the data into 439 PCs. This method requires proper benchmarking, which is not included in the paper. Here are several comments on the genetic analysis:
 - a. I did not see any standard QC plots for the genetic analysis. What is the population structure (PCA of genetic data)? What are the genotyping methods, quality control measures, and how was family structure accounted for?
 - b. There is limited evidence that the proposed (somewhat unconventional) methodology is valid. There are many concerns: the methodology is not composition-aware at all. There was no GWAS on microbial richness or entropy. One might assume that the actual captured effects of genetic makeup are related more to alpha diversity than to beta diversity.
 - c. The idea behind the approach is ideologically similar to a GWAS on beta diversity – capturing the effects of SNPs on overall composition. This idea was proposed in 2015 (<https://www.biorxiv.org/content/10.1101/031187v1>). This method has never been efficient in gut microbiome studies, as the effects of SNPs on overall microbial composition are limited. This paper does not provide evidence that the authors' approach outperforms a PERMANOVA-based analysis.
 - d. Also, this GWAS provide limited evidence regarding which taxa are affected by the loci. This is important information, and the sample size is sufficient to detect such effects.

e. Importantly, this approach requires benchmarking – there is no evidence of its validity, especially considering the use of relative abundances in microbiome data. The analysis does not address the compositionality problem. This is also important for individual bacterial GWAS where many signals may result from compositional artefacts.

f. In general, if the effect of certain genetic loci is strong, it should be captured in a limited number of PCs. How were associations distributed across PCs at associated loci?

5. “To avoid test statistic inflation from outlier values, relative abundance measures for each taxon were adjusted for age (using linear regression) and then rank-based inverse normal transformed.” This approach is not correct: most microbial taxa distributions suffer not from single outliers, but from zero inflation, e.g., a “distribution with fingers” where multiple samples have the same value. These patterns do not disappear after inverse normal transformation. Moreover, if the original trait is adjusted before rank-based transformation, the jitter introduced by regression-based adjustment, once rank-transformed, can turn into a pseudo-signal that is inversely and strongly associated with the adjusted factors, since the rank-based transformation equalizes distances between closely ranked samples. A more reasonable approach would be to adjust after inverse normal transformation, not before.

6. Figure 2 and corresponding text: What does “more abundant with functional FUT2, less abundant with functional FUT2” mean? Please clarify (also in the figure legend) which microbes are increased/decreased in secretors vs. non-secretors. Top 10 FUT2 associations are shown – please indicate the p-values for these associations. Figure 2C: Please adjust the scale so that the p-value for association with individual taxa is visible, or add the p-value for the strongest association.

7. “To account for relatedness among SPARK participants, we performed genome-wide association analysis on each mPC phenotype using the linear mixed model implemented in BOLT-LMM (using the `--lmmInfOnly` flag, as the non-infinitesimal mixed model provided a negligible increase in statistical power) with the following covariates: sequencing batch, age, age squared, square root of age, sex, percent of mapped reads, and the top 10 genetic ancestry principal components.” – It is unclear how this model accounts for family structure. I do not see family mentioned in the described model. This is particularly important given that many SPARK participants (at least those mentioned in references) come from families with multiple individuals. Again, this highlights the poor description of the cohort and the genetic analysis.

8. SPARK is a family cohort; it would be valuable to estimate the heritability of the oral microbiome using the family structure. Comments on the associations between bacterial gene dosage phenotypes and the 11 human genetic variants:

9. I doubt the term “structural variants (SVs)” is correct for bacterial gene dosages based on read-depth phenotypes. Although these gene dosages might indicate genomic regions that may harbor SVs, additional analyses are needed to confirm and properly characterize them as SVs (with defined boundaries, types, confidence scores, etc.) before using this term.

10. The methods section does not mention or assess the normality of continuous variables used in the linear model for association analysis.

11. Limiting the association analysis only to the 11 human genetic variants previously found to be associated with microbiome composition of 30 microorganisms introduces general bias. For example, other genetic loci may have stronger effects on microbial gene dosage but remain undetected, or other microbes (not included) may affect gene dosage through microbe-microbe interactions.

12. For this analysis, I do not see proper control for species relative abundance. In Supplementary Note 4: “We normalized a given sample’s read depth in each 500bp bin of a given bacterial genome by the median genomic coverage for that species in that sample.” This seems more like a within-sample read depth adjustment rather than an adjustment for species abundance across samples. I would expect the inclusion of species relative abundance as a covariate in the model to ensure that gene dosage effects are independent of between-sample species abundance.

13. A model using only the top 20 principal components of the normalized, truncated read-depth matrix for that species may not be sufficient, as other factors such as age and sex could be confounders. This could lead to spurious associations.

14. Additionally, it would be useful to check the residual population structure after PC correction. Twenty PCs (which seems arbitrary) may be too many or too few, depending on the species. It would be better to determine the optimal number for each case using predefined thresholds (e.g., variance explained).

15. In Supplementary Note 4, the following confounding scenario is described: “Lastly, one concern that we recognize the above adjustments do not directly guard against is the possible scenario in which (i) mismapping occurs unevenly from another species B onto regions of species A’s genome, (ii) species A but not species B is in our reference panel, and (iii) the abundance of species B associates with a human genotype G. In this scenario, an association between genotype G and the abundance of species B could generate an apparent association of genotype G with gene dosage at one or more genomic bins of species A.”

16. In the following paragraph, the authors argue that this confounding effect is unlikely to be a major contributor to the observed gene dosage associations: “While we cannot formally exclude the possibility of this last form of potential confounding, a few lines of evidence suggest that it is not a major contributor to the gene dosage associations that we identified. One reason is that if an association of genotype G with gene dosage in species A were due to mismapping from species B, then the abundance of species B should associate much more strongly with genotype G than the less-powered association of G with gene dosage in species A. On the contrary, the bacterial gene dosage associations we highlighted in this work had association strengths exceeding all but the strongest associations of human genotypes with relative abundance phenotypes.” However, I am not sure that the strength (effect size estimates) of the associations is directly comparable between species abundance and gene dosage measurements, as these are two different methods (with different normalization, transformations, covariates, etc.).

17. The limitations of the analysis methods are not discussed in the manuscript. For example, limitations of the bacterial gene dosage measurement method (in terms of precision, sensitivity, type detection, etc.), limitations due to studying only a subset of associations (11 loci and 30 species), and limitations of the data for this analysis are not addressed. These are only partially covered in Supplementary Note 4.

18. The discussion also lacks recommendations for future analyses or ways to improve the current methodology.

19. The human genetic associations with microbial gene dosages are not replicated in other cohorts with different genetic backgrounds or environmental exposures. This limits the generalizability of the findings.

On the AlphaFold analysis:

20. For the predicted models, I would expect confidence measures of the individual protein structures to be provided (e.g., average and per-residue pLDDT scores), especially for regions involved in protein-protein interactions between human AMY1 and *S. parasanguinis* AbpA and AbpB.

21. For the complex and protein-protein interaction model predictions, TM scores and interface TM (ipTM) scores should be provided in the supplementary material to assess the reliability of the predictions.

22. In the methods section ("AlphaFold2 prediction of protein structures"), please include the version of ChimeraX used to visualize the structures.

Minor comments:

- What is the difference between Supplementary Tables 1 and 2? Please add species names to the strain names in ST1. Also, ST4 appears to be a subset of ST3 and could be merged.
- What is the purpose of scaling and centering after inverse rank transformation? This transformation already produces distributions close to $N(0,1)$.
- Table 1: Add r²IDs, MAFs, and LD between SNPs if they differ between traits. Consider moving the table to the supplement.
- "Tiled across 30 bacterial reference genomes" – why only 30? After reading the methods, I see this is explained, but please include a brief description or reference in the results section.
- "A recent oral microbiome GWAS9 reported two associations in the centromeric region" – centromeric region of which chromosome?
- The introduction would benefit from a paragraph summarizing what was done in this study to orient the reader.
- Figure 1 is labeled "Figure 1," but others are not. Please remove or standardize for consistency.
- In Supplementary Note 4, four steps of transformations/adjustments are described. It would be useful to reference "Supplementary Note 4" in the main methods section ("Measuring bacterial gene dosages using read-depth phenotypes").
- Mendelian Randomization (MR): While I acknowledge that most MR analyses related to the microbiome produce low-confidence results due to moderate genetic effects and little overlap in associated signals, in this case, MR on dental phenotypes and the oral microbiome across all associated loci would still be relevant.

(Remarks on code availability)

Referee #2

(Remarks to the Author)

Kamitaki et al. examine associations between oral microbiome composition and host genetics in a well-powered human cohort ($n > 10,000$ subjects). The authors discover novel genetic mechanisms of microbiome modulation that they establish are linked with oral health. The authors' validations of their approaches are meticulous (e.g. oral expression of identified genes, concordant sibling AMY1 copy numbers, comparing adult vs. child AMY1 microbiome associations to determine likelihood of causality given the link between AMY1 and dentures use) and their methods are sophisticated and well-considered (e.g. adjusting for sequencing batches, host genetic PCs, performing a combined chi-square test to reduce multiple comparisons). Overall, this is a beautifully performed study that reports valuable insights into the host-microbiome relationship of the oral cavity and puts forth a strong analytical framework that can serve as a model for future studies.

My comments/questions:

ASD has been reported to associate with differences in the oral microbiome within the same SPARK cohort (Manghi et al. *Nature Microbiology* 2024). How do the authors reconcile the findings of Manghi et al. with their assertion that ASD did not exert significant impact on the oral microbiota? Manghi et al. found associations also existed between oral microbiota and IQ, as well as microbiota composition and overall microbial load, the latter of which they proposed is correlated with poor dental hygiene. The authors may consider sensitivity analyses that adjust for these variables (including ASD) in order to test whether key associations remain evident or if new associations are uncovered.

Associations of host genomic variants with microbial genomic regions but not species relative abundance is intriguing and is concordant with a study by Nahavi et al. *Nature Medicine* 2023 (<https://doi.org/10.1038/s41591-023-02599-8>) which found that many gut microbial species encoded genomic SNPs that correlated with a host phenotype (BMI) while the relative abundance of that species in the gut microbiota did not. This may be a relevant supporting reference for the authors in their finding that genomic structure may provide an orthologous method by which to examine host-microbial relationships beyond relative abundance.

Is it the case, as for *Prevotella* spp. and ABO*A1 genotype, that *R. mucilaginosus* relative abundance also did not associate with ABO genotype? The phenomenon of finding genomic structural associations in taxa for which no relative abundance association was evident is also relevant for how the authors selected host gene variants and microbial genomes to query in the genomic structural analysis section. These were selected by which genes and microbes exhibited significant associations comparing to microbial relative abundances. While this approach that the authors took was well-reasoned to minimize the number of statistical comparisons, an exploratory analysis expanded to all genes and all microbial taxa may well yield additional novel associations. Could the authors perform this, or a summed chi-square approach as was used for mPCs? The authors may acknowledge in their text that more microbial genomic structural variations can be linked with other untested gene variants even in the same dataset. The present study will likely set precedent for such analyses, and so it may be helpful to underline this concept so that future researchers do not miss informative associations.

The authors seem to present the idea that AMY1 copy number variants arose due to the selective pressure of oral health as

being exclusive of AMY1 copy number variants having arisen due to development of agricultural mass production of starch-laden foods (in the sentence 'AMY1 copy number may instead have been under selection due to effects on oral health'). Is it not likely that both acted as selective pressures over the same timescale in tandem – that is, that oral health was an additional, as-yet unappreciated selective pressure? One can imagine that better digestion of starches and protection from fatal dental infections that arise from these new food sources shaped this locus simultaneously. Would the authors agree with this, and consider revising the sentence to be inclusive of such a possibility? Also, it is unclear to me why a lack of association with BMI is evidence against agriculture being a selective pressure that shaped AMY1 copy number; AMY1-mediated breakdown of starches may contribute to dietary energy extraction but not obesity which may instead be caused by dietary intake of a complex combination of simple sugars, ultra-processed foods, and other dietary/lifestyle factors for which detrimental effects are highly personalized.

(Remarks on code availability)

Version 2:

Reviewer comments:

Referee #1

(Remarks to the Author)

In general, the authors did a comprehensive job in replying to all our comments and replied all important points. Two comments remain:

c. "The idea behind the approach is conceptually similar to a GWAS on beta diversity – capturing the effects of SNPs on overall composition."

Here the question was not about the tool but about the approach – how does your PC-based analysis (essentially MMR/MANOVA) compare with Multivariate Distance Matrix Regression (MDMR; Anderson, 2001; McArdle & Anderson, 2001)? We expected to see this comparison, which could be done, for example, using the available R package (<https://cran.r-project.org/web/packages/MDMR/> – note, we do not have experience running it), or by using other microbiome GWAS tools. It would be valuable to see benchmark results, e.g. for 1,000 samples and 1,000 SNPs, as a demonstration of validity of your approach.

f. "In general, if the effect of certain genetic loci is strong, it should be captured in a limited number of PCs." Looking at the figures, however, we think that most of the time the signal is instead distributed across a large number of PCs (except in the example you mention). The biological relevance of this observation remains unclear to us, especially considering that these PCs do not reflect different bacteria and are orthogonal. We are interested in how the authors explain this phenomenon.

(Remarks on code availability)

Referee #2

(Remarks to the Author)

It is a pity that additional reference genomes cannot be used to query associations between other oral bacteria and human genetic variants, but it is appreciated that the authors remarked on how that may be overcome in future studies in their Discussion (via long-read sequencing). Perhaps this study will stimulate greater interest in doing so. The authors have otherwise addressed my comments.

(Remarks on code availability)

Version 3:

Reviewer comments:

Referee #1

(Remarks to the Author)

"MDMR approach appeared to be considerably less powerful": the benchmark is convincing and a valuable addition to the study, we appreciate that the authors added it. However, we believe that stating that the 'MDMR approach is considerably less powerful' is an overstatement. Instead, we suggest that the authors state in the paper that the top signals are convincingly confirmed by the (more conservative) MDMR method. We have no further comments, great study.

(Remarks on code availability)

Referee #1:

The paper by Kamitaki et al. focuses on the analysis of the relationship between host genetics and the oral microbiome. They analysed microbial and human reads in oral samples from 12,519 individuals and identified 439 common microbial species (present in >10%), which they linked to host genetic variants. Instead of the commonly used approach (testing the association of each genetic variant with each taxon), they used an alternative approach, by performing an association with overall microbial composition. More specifically, they created 439 PCs (corresponding to the number of taxa), associated each genetic variant with all PCs, and performed a meta-analysis across all PCs. I have several critical comments on the method, though I acknowledge that the associated loci point to relevant biological molecules.

We appreciate the detailed, constructive feedback identifying specific areas in which the efficacy and robustness of the method needed validation or the analytical approach could be improved. We have now taken these helpful suggestions and performed several new analyses, including:

- Benchmarking the new method against the commonly used approach of testing the association of each human genetic variant with each taxon (Comment 4).
- Correcting the way in which we applied inverse normal transformation to per-taxon abundance phenotypes (Comment 5). This change considerably improved GWAS power, increasing the number of variant-species associations from 136 to 167.
- Demonstrating that the associations of human genetic effects on microbial abundances and gene dosages were not driven by effects on compositionality or alpha diversity (Comment 4) or by confounders such as age and sex (Comment 13).
- Evaluating the effect of limiting the microbial gene dosage association analyses to 11 lead human genetic variants (Comment 11).
- Replicating the associations of human genetic variants with bacterial gene dosages in an independent cohort (Comment 19).
- Generating supporting data describing distributional metrics for continuous gene dosage phenotypes (Comment 10) and confidence metrics for AlphaFold results (Comments 20 and 21).

In addition to incorporating these new analyses into the manuscript, we have also revised the manuscript text in numerous ways to clarify points of confusion and discuss limitations of the approach, as helpfully suggested in the comments below. We appreciate the thoughtful reading and believe that these improvements have considerably strengthened the manuscript.

Please see my comments below:

1. The methods and sample selection for this study are not clearly described. In the results section, the authors state that they used samples from the SPARK cohort and cite two references. It reads as if the data were generated in these previous studies. However, the numbers do not match: in this paper, the number of samples mentioned is: “Oral microbiomes in 12,519 individuals measured by saliva whole genome sequencing.” In the two reference papers, two different sample sizes from SPARK are mentioned: 1. “SPARK cohort: N = 7812 individuals from 2025 US families, avg. family size = 3.9”; 2. “A total of 1,929 saliva samples from 646 families in the NY Genome Center Cohort of the SPARK collection.”

Concerns: If the authors used published SPARK data, the numbers do not match. Moreover, they should describe which samples came from which study and how they handled the batch effect. If the authors performed the sequencing as part of this study, then they need to add all missing information on sequencing: sample collection, DNA isolation methods, and sequence depth.

Thank you for pointing out that the source of the SPARK WGS data was confusing. We have now clarified in the first paragraph of Methods (p. 35) that all the WGS data we analyzed were previously generated. The reason that the number of samples does not match the numbers in the two referenced papers is that the SPARK WGS data has been accumulating over time, such that the two references we cite (as work we are “building on”; Results, p. 2) pertain to earlier subsets of the 12,519 samples that we analyzed (specifically, WGS1-3 out of WGS1-5 now available). We have now clarified this in Methods and cited Manghi et al. (2024, *Nat Commun*; ref. 29) for details of saliva sample collection, DNA extraction, and sequencing for WGS1-3. This appears to be the best reference for the WGS data production as the data provider (SFARI) has not published a manuscript describing the current set of 12,519 SPARK WGS samples, and the SPARK documentation does not suggest any differences in sample preparation or sequencing of the newer WGS batches.

Regarding handling potential batch effects, yes, we did include WGS batch as a categorical covariate in all relevant analyses; we have noted this in Methods (pp. 36, 37, 46).

2. From previous publications, it seems that SPARK was a family-based design – many samples came from the same family. In this case, genetic analysis should take into account family structure; it is unclear if this was done and how.

Yes, SPARK used a family-based design, so to account for the family structure, we used a linear mixed model (LMM) to perform the genetic association tests. We have now clarified

this in the main text (Results, p. 5) and cited three references (refs. 35–37) that established that LMMs account for familial relatedness in genetic association analyses:

<https://www.nature.com/articles/ng1702>

<https://www.nature.com/articles/ng.548>

<https://www.nature.com/articles/ng.3190>

3. Identifying 439 common bacteria is surprisingly high. How many taxa were identified in total (e.g., at >1% frequency)? Since this appears to be the first description of the corresponding dataset (see point 1), I expect the authors to present more descriptive statistics, including the total number of identified microbes and their frequencies.

We found 645 species at >1% frequency and have added this information to the main text (Results, p. 5) as suggested. We have also added a new Supplementary Table 1 providing the frequency of each observed taxon. The number of species detected is broadly consistent with previous analyses of the SPARK WGS data: we identified a median of 228 species in each person (quartiles, [188–268]), similar to the average of 161 species per sample found in the previous analysis of WGS1-3 by Manghi et al. (2024, *Nat Commun*). The modest increase in detection in our analysis of WGS1-5 is expected given that we used a more recent MetaPhlAn database release (vOct22) containing 30,550 species (compared to 13,519 species in the version of the database that Manghi et al. used for their analysis). Manghi et al. also noted the higher number of species identified from WGS and verified that this was not due to contamination; they concluded that WGS affords higher resolution than the 16S rRNA profiling used for many earlier microbiome cohorts.

4. Genetic analysis: the authors used a different method. Instead of running GWAS on microbial abundances or presence, they transformed the data into 439 PCs. This method requires proper benchmarking, which is not included in the paper.

These points are well taken, and we agree that additional data demonstrating the validity of the new approach and comparing it to the standard approach were needed. We have now undertaken these benchmarks and restructured the manuscript to start by describing results of the standard GWAS approach and then showing the improvement in power obtained using the new approach (Results, p. 5). We have also included a new multi-panel Extended Data Fig. 2 demonstrating the robustness of the new approach in the ways suggested below (see below for details); we appreciate these helpful suggestions.

Here are several comments on the genetic analysis:

a. I did not see any standard QC plots for the genetic analysis. What is the population structure (PCA of genetic data)? What are the genotyping methods, quality control measures, and how was family structure accounted for?

We have now included plots of the first four human genetic principal components as Extended Data Fig. 1f,g (copied below). As expected, the PCs capture population structure in the SPARK data set, separating clusters of individuals in different self-reported race/ethnicity groups. These PCs explained very little variance in microbial abundances (typically <1%: Fig. 1c), so we followed the standard practice of including 10 genetic ancestry PCs as covariates in genome-wide association analyses (Methods, p. 36).

Regarding the genotyping methods and QC, we have created a clearly labeled section of Methods (“Genotyping and quality control of human genetic variants in SPARK”; pp. 35–36) providing this information. Briefly, variant calling had previously been performed using DeepVariant, and we performed standard QC steps on these variant calls (filtering on missingness, MAF, Hardy-Weinberg equilibrium, and presence in a reference panel).

Regarding accounting for family structure, we performed association analyses using a linear mixed model (see response to Comment 2 above).

b. There is limited evidence that the proposed (somewhat unconventional) methodology is valid. There are many concerns: the methodology is not composition-aware at all. There was no GWAS on microbial richness or entropy. One might assume

that the actual captured effects of genetic makeup are related more to alpha diversity than to beta diversity.

We appreciate that these potential issues needed to be addressed and have now performed several new analyses to validate the new methodology.

First, to establish a baseline against which we could validate and compare our results, we ran the standard approach of testing each human genetic variant for association with the relative abundance of each taxon (i.e., one GWAS per taxon observed at >10% frequency, using a stricter Bonferroni significance threshold to account for the increased number of tests). This standard approach identified 7 of the 11 loci found by our new approach (and no additional loci), as we have now explained in Results (p. 5) and the new Extended Data Fig. 2a (copied below), which plots the strongest p-value (among all the per-taxon tests) observed for each human genetic variant.

Second, to assess the impact of compositionality on these results, we ran another set of GWAS on per-taxon relative abundance phenotypes that we transformed using the centered log-ratio (clr) approach, which is commonly used to address compositionality in microbiome analyses (Tsilimigras & Fodor 2016; ref. 87: <https://www.sciencedirect.com/science/article/pii/S1047279716300722>). This alternative analytical approach identified the same 7 of 11 loci, with highly concordant effect size estimates for the 167 variant-species associations that reached significance (FDR<0.05) in the non-clr-transformed analysis (Extended Data Fig. 4e, copied below). These results suggest that the identified associations of human genetic variants with individual bacterial abundances are unlikely to be compositional artifacts, and that these effects are largely robust to transformation. We have described these analyses in Methods (p. 38) and the new Extended Data Fig 4e.

The above two analyses using standard microbiome GWAS techniques provided strong support for 7 of the 11 loci identified by our new approach. The remaining 4 loci that only reached genome-wide significance using our new microbial PC-based approach all have clear biological relevance to the oral microbiome—*ABO*, *HLA*, *SMR3A/B* (submaxillary gland androgen regulated proteins), and *PITX1* (involved in tooth morphology; top locus for dental caries)—strongly suggesting that the microbial PC-based approach achieved improved statistical power relative to the standard approach.

To further ensure that this increase in GWAS power was not an artifact of test statistic miscalibration, we evaluated the distribution of p-values computed by our new approach for variants across the genome (most of which are expected to have no effect on oral microbiome composition). As expected, the bulk of the p-value distribution matched the null (genomic inflation factor $\lambda_{GC} = 0.998$; Extended Data Fig. 2c, copied below).

Finally, to assess whether or not any of the captured effects of genetic makeup on oral microbiome composition could reflect effects on microbial richness or entropy, we have now also run a GWAS for alpha-diversity (Shannon entropy). This GWAS did not identify any significant associations (Extended Data Fig. 2b, copied below), indicating that the human genetic effects we identified are unlikely to be mediated by effects on alpha-diversity.

c. The idea behind the approach is ideologically similar to a GWAS on beta diversity – capturing the effects of SNPs on overall composition. This idea was proposed in 2015 (<https://www.biorxiv.org/content/10.1101/031187v1>). This method has never been efficient in gut microbiome studies, as the effects of SNPs on overall microbial composition are limited. This paper does not provide evidence that the authors' approach outperforms a PERMANOVA-based analysis.

We agree that our approach is ideologically similar to a GWAS on beta diversity, and we agree that the efficacy of the approach relies on SNPs having effects on many bacterial species, which appears to be the case for the oral microbiome but not the gut microbiome. This is an interesting point that we comment on in the Discussion (p. 24); one reason might be that host cells in the mouth interface more directly with bacteria (in contrast to cells in the gut, which are typically protected by the gut mucosal barrier), affording more varied interactions with human surface proteins.

We also agree that it would be of interest to compare our microbial PC-based approach to a PERMANOVA-based analysis. We attempted to run PERMANOVA as suggested, but this turned out to be computationally intractable: a widely-used implementation (adonis2 from the R package vegan) took ~30 minutes to finish 10 permutations for a single SNP, such that computing a genome-wide significant p-value ($p < 5 \times 10^{-8}$) even for a single variant would take >100 years. This limitation of permutation-based methods is actually highlighted in the published version of the 2015 bioRxiv preprint, which notes that statistical approaches that require permutations to assess significance are “computationally prohibitive, particularly when evaluating p-values less than 5×10^{-8} —the standard GWAS p-value threshold—or even lower when testing multiple-diversity matrices” (Hua et al. 2022, *Genes*).

As such, while we agree that it is unclear whether our approach would in theory outperform a PERMANOVA-based analysis, in practice, such an approach is not possible to run on a large GWAS cohort.

d. Also, this GWAS provide limited evidence regarding which taxa are affected by the loci. This is important information, and the sample size is sufficient to detect such effects.

We agree that this information is important, and we have provided this information for the 167 variant-species associations that reached significance ($FDR < 0.05$) in Supplementary Table 3. We have also provided full variant-taxon association data for the 11 lead variants (including taxa that did not reach significance) in Supplementary Table 2.

e. Importantly, this approach requires benchmarking – there is no evidence of its validity, especially considering the use of relative abundances in microbiome data. The analysis does not address the compositionality problem. This is also important for individual bacterial GWAS where many signals may result from compositional artefacts.

We have now performed several new analyses to demonstrate the validity of our microbial PC-based approach, benchmark its performance relative to the standard approach of running one GWAS per taxon, and demonstrate that the identified associations did not result from compositional artifacts (see response to Comment 4b above for details).

f. In general, if the effect of certain genetic loci is strong, it should be captured in a limited number of PCs. How were associations distributed across PCs at associated loci?

We have now provided plots showing how the associations for each of the 11 lead variants were distributed across the 439 microbial abundance PCs (Extended Data Fig. 2d-n, copied below). The extent to which association signals were distributed across few or many PCs was somewhat difficult to evaluate at the sample size of SPARK, but it does appear that some associations (e.g., at *TLR1*, 4_38797027_C_A in panel f) are driven by a small subset of PCs.

5. “To avoid test statistic inflation from outlier values, relative abundance measures for each taxon were adjusted for age (using linear regression) and then rank-based inverse normal transformed.” This approach is not correct: most microbial taxa distributions suffer not from single outliers, but from zero inflation, e.g., a "distribution with fingers" where multiple samples have the same value. These patterns do not disappear after inverse normal transformation. Moreover, if the original trait is adjusted before rank-based transformation, the jitter introduced by regression-based adjustment, once rank-transformed, can turn into a pseudo-signal that is inversely and strongly associated with the adjusted factors, since the rank-based transformation equalizes distances between closely ranked samples. A more reasonable approach would be to adjust after inverse normal transformation, not before.

We agree. This was a mistake, and the approach of adjusting for covariates only after inverse normal transformation makes more sense. We have now rerun all analyses that applied these steps in the opposite order and observed a gain in power (increasing the number of significant variant-species associations from 136 to 167), consistent with the reviewer’s intuition that regression-based adjustment prior to inverse normal transformation had been introducing jitter and reduced the performance of the association test. We have updated all results affected by this change (Fig. 2cde, Fig. 3bci, Fig. 4ef, Fig. 6dh; Extended Data Figures 3bcde, 4bg, 5bcdefghij, 8a, 9ad; and Supplementary Tables 2 through 5).

6. Figure 2 and corresponding text: What does “more abundant with functional FUT2, less abundant with functional FUT2” mean? Please clarify (also in the figure legend) which microbes are increased/decreased in secretors vs. non-secretors. Top 10 FUT2 associations are shown – please indicate the p-values for these associations. Figure 2C: Please adjust the scale so that the p-value for association with individual taxa is visible, or add the p-value for the strongest association.

We appreciate these helpful suggestions and have clarified in the legend of Fig. 2c (p. 7) that red curves correspond to species that are more abundant in individuals with at least one functional copy of *FUT2* (i.e., secretors), and blue curves correspond to species that are less abundant in secretors. We have also annotated Fig. 2c and 2d with p-values.

7. “To account for relatedness among SPARK participants, we performed genome-wide association analysis on each mPC phenotype using the linear mixed model implemented in BOLT-LMM (using the --lmmInfOnly flag, as the non-infinitesimal

mixed model provided a negligible increase in statistical power) with the following covariates: sequencing batch, age, age squared, square root of age, sex, percent of mapped reads, and the top 10 genetic ancestry principal components.” – It is unclear how this model accounts for family structure. I do not see family mentioned in the described model. This is particularly important given that many SPARK participants (at least those mentioned in references) come from families with multiple individuals. Again, this highlights the poor description of the cohort and the genetic analysis.

We have revised this description in Methods (p. 36) to explain that linear mixed models account for family structure in genetic association analyses, as demonstrated by the following references (which we now cite as refs. 35–37):

<https://www.nature.com/articles/ng1702>

<https://www.nature.com/articles/ng.548>

<https://www.nature.com/articles/ng.3190>

8. SPARK is a family cohort; it would be valuable to estimate the heritability of the oral microbiome using the family structure.

We agree that the heritability of oral microbiome phenotypes would be interesting and valuable to estimate, but unfortunately, the sample size of the SPARK data set is insufficient to support such analyses. We are aware of two family-based approaches to estimating heritability that are robust to shared environmental effects: (i) comparing phenotypic correlations of monozygotic vs. dizygotic twins; and (ii) “sib-regression,” which makes use of variation in the amount of genomic sharing across sibling pairs (Visscher et al. 2006, *PLOS Genet*). The SPARK WGS cohort only has 9 monozygotic twin pairs, making approach (i) infeasible. The 2,188 sibling pairs in the SPARK WGS data are also insufficient for approach (ii), which is expected to produce standard errors of 0.2–0.4 at this sample size according to Table 1 of Visscher et al. 2006:

<https://journals.plos.org/plosgenetics/article/figure?id=10.1371/journal.pgen.0020041.t001>

For phenotypes that are minimally impacted by shared environmental effects, heritability can also be roughly estimated as twice the sibling correlation, but this approach would greatly overestimate the heritability of the oral microbiome given the expectation of strong environmental effects within families. In light of these challenges, we did not further pursue this line of inquiry.

Comments on the associations between bacterial gene dosage phenotypes and the 11 human genetic variants:

9. I doubt the term "structural variants (SVs)" is correct for bacterial gene dosages based on read-depth phenotypes. Although these gene dosages might indicate genomic regions that may harbor SVs, additional analyses are needed to confirm and properly characterize them as SVs (with defined boundaries, types, confidence scores, etc.) before using this term.

This is a good point. We have edited the manuscript text to no longer use the terms "structural variants" or "SVs" in describing the associations with bacterial gene dosages.

10. The methods section does not mention or assess the normality of continuous variables used in the linear model for association analysis.

We have now provided distributional metrics for the continuous variables involved in associations between human genetic variants and bacterial read-depth phenotypes (specifically, the fractions of samples with normalized coverage of 0 or 1, and the mean normalized coverage value; Supplementary Table 7) to give a sense of their distribution across the cohort.

At GWAS-scale sample sizes, linear regression is actually quite robust to violation of normality (Schmidt & Finan 2018, *J Clin Epidem*; PMID 29258908), especially when testing common variants. The main issue that arises is using regression to test rare variants for association with unbalanced binary traits (Loh et al. 2018, *Nat Genet*; Zhou et al. 2018, *Nat Genet*; PMID 29892013 and 30104761). Here, we tested common variants (MAF=0.08–0.45) for association with gene dosage phenotypes that were broadly reasonably distributed (with most having min=0, max=1, and mean between 0.05 and 0.95), such that using linear regression in a cohort of size 12,519 was expected to produce robust test statistics. We have now explained this in Methods (p. 44).

Additionally, we have now replicated the large majority of the bacterial gene dosage associations in an independent cohort (*All of Us*; see response to Comment 19 below), confirming the robustness of the statistical tests.

11. Limiting the association analysis only to the 11 human genetic variants previously found to be associated with microbiome composition of 30 microorganisms introduces general bias. For example, other genetic loci may have stronger effects on microbial gene dosage but remain undetected, or other microbes (not included) may affect gene dosage through microbe-microbe interactions.

We completely agree that these are important limitations of testing only 11 human genetic variants (nominated by our GWAS of microbiome composition) for association with gene

dosages of only 30 bacteria. We took this approach for two main reasons: (1) to minimize multiple hypothesis testing burden (by testing 11 variants rather than millions of common variants in the human genome); and (2) because identifying high-quality reference genomes for bacterial species required manual curation and often was not even possible. For example, many of the species genomic bins (SGB) quantified by the MetaPhlan pipeline are derived from metagenomic bins and lack reference genomes. We have now noted these limitations and explained the rationale for our approach in Discussion (p. 25) and a new section of Supplementary Note 4 (p. 11, discussing limitations of the method).

Additionally, to assess the extent to which limiting the analyses to 11 human genetic variants might have resulted in our missing associations of other human genetic loci with bacterial gene dosages, we have now undertaken two sets of GWAS testing all common human genetic variants for association with two sets of exemplar gene dosage phenotypes (Methods, pp. 45–46):

- (i) five GWAS searching for additional human genetic effects on the five bacterial gene dosage phenotypes that associated most strongly with the 11 lead variants; and
- (ii) 3,441 GWAS searching for additional human genetic effects on 3,441 read-depth phenotypes in 500bp bins spanning the genome of the most prevalent species observed in SPARK (*Rothia mucilaginosa*).

Neither of these analyses identified strong associations beyond the main effects we observed at *FUT2* and *ABO*, suggesting that limiting the tests to the 11 variants was a reasonable trade-off to enable detecting effects on more species and gene dosages (Extended Data Fig. 10c-h, copied below).

(i) GWAS of five bacterial gene dosage phenotypes with top human genetic associations:

(ii) GWAS of 3,441 read-depth bins for *R. mucilaginosa* (top p-value per human variant):

We have included these results in Extended Data Fig. 10c-h while also acknowledging the limitations of the approach in Discussion (p. 25) and Supplementary Note 4 (p. 11), noting that future analyses using new methods, an expanded set of microbial reference genomes, and larger cohorts are likely to identify more effects.

12. For this analysis, I do not see proper control for species relative abundance. In Supplementary Note 4: “We normalized a given sample’s read depth in each 500bp bin of a given bacterial genome by the median genomic coverage for that species in that sample.” This seems more like a within-sample read depth adjustment rather than an adjustment for species abundance across samples. I would expect the inclusion of species relative abundance as a covariate in the model to ensure that gene dosage effects are independent of between-sample species abundance.

We apologize for the lack of clarity: this adjustment was actually an adjustment for species abundance across samples, as suggested. The median genomic coverage of a given species in a given sample quantifies the abundance of that species in that sample, such that by normalizing each 500bp bin of that species in that sample by this quantity controls for variation across samples due to varying species abundance (resulting in normalized read-depth measurements that have a median of 1 in each sample across bins of each species). We have clarified this text in Supplementary Note 4 (p. 9).

13. A model using only the top 20 principal components of the normalized, truncated read-depth matrix for that species may not be sufficient, as other factors such as age and sex could be confounders. This could lead to spurious associations.

We repeated the analysis including age and sex as covariates and observed that these covariates had no effect on the 208 associations (no change in effect sizes):

14. Additionally, it would be useful to check the residual population structure after PC correction. Twenty PCs (which seems arbitrary) may be too many or too few, depending on the species. It would be better to determine the optimal number for each case using predefined thresholds (e.g., variance explained).

We appreciate this suggestion, but in thinking through how we would implement such an approach, we realized that using a predefined threshold on variance explained would run into challenges of its own due to the considerable range (across species) of the proportion of variance explained by structure. For example, if the top 20 PCs explained 50% of variance but our threshold was lower (e.g., 10%), the lower threshold would result in using too few PCs. On the other hand, if these 20 PCs only explained 10% of variance and our threshold was higher (e.g., 50%), we might end up including an unreasonably large number of PCs as covariates, reducing statistical power.

While we agree that using a fixed number of PCs (20) is also an imperfect solution, it does have the benefit of ensuring control of any major confounding structure (which is captured by top PCs) while also ensuring preservation of statistical power (as 20 is a small fraction of the SPARK sample size); we have added this rationale to Supplementary Note 4 (pp. 9–10). This approach is commonly taken in GWAS.

Additionally, any residual confounding effects of structure would be ameliorated by the genomic control that we applied as a final step (Methods, pp. 43–44). The robustness of this overall analytical pipeline is borne out by the replication analyses we have now conducted (see response to Comment 19 below).

15. In Supplementary Note 4, the following confounding scenario is described:

“Lastly, one concern that we recognize the above adjustments do not directly guard against is the possible scenario in which (i) mismapping occurs unevenly from another species B onto regions of species A’s genome, (ii) species A but not species B is in our reference panel, and (iii) the abundance of species B associates with a human genotype G. In this scenario, an association between genotype G and the abundance of species B could generate an apparent association of genotype G with gene dosage at one or more genomic bins of species A.”

16. In the following paragraph, the authors argue that this confounding effect is unlikely to be a major contributor to the observed gene dosage associations: “While we cannot formally exclude the possibility of this last form of potential confounding, a few lines of evidence suggest that it is not a major contributor to the gene dosage associations that we identified. One reason is that if an association of genotype G with gene dosage in species A were due to mismapping from species B, then the abundance of species B should associate much more strongly with genotype G than the less-powered association of G with gene dosage in species A. On the contrary, the bacterial gene dosage associations we highlighted in this work had association strengths exceeding all but the strongest associations of human genotypes with relative abundance phenotypes.” However, I am not sure that the strength (effect size estimates) of the associations is directly comparable between species abundance and gene dosage measurements, as these are two different methods (with different normalization, transformations, covariates, etc.).

We apologize for the ambiguity in our word choice: by “strength” of an association, we meant the amount of statistical signal (i.e., p-values) rather than the magnitude of the effect (i.e., effect sizes, which we agree are not comparable across the two methods). We have revised the text in Supplementary Note 4 (pp. 10–11) to clarify and further explain this reasoning:

“One reason is that if an association of genotype G with gene dosage in species A were due to mismapping from species B, then the abundance of species B should have a much more significant association with genotype G than the less-powered association of G with gene dosage in species A. That is, read alignments to a 500bp bin of species A (even if solely due

to mismapping from species B) should provide a noisier quantification of species B's abundance than the quantification computed by MetaPhlAn, which utilizes numerous marker genes throughout the genome of species B. On the contrary, the bacterial gene dosage associations we highlighted in this work had association test statistics exceeding all but the most significant associations of human genotypes with relative abundance phenotypes.”

17. The limitations of the analysis methods are not discussed in the manuscript. For example, limitations of the bacterial gene dosage measurement method (in terms of precision, sensitivity, type detection, etc.), limitations due to studying only a subset of associations (11 loci and 30 species), and limitations of the data for this analysis are not addressed. These are only partially covered in Supplementary Note 4.

We agree that an expanded discussion of the limitations of the analytical approach was warranted, and we have now revised the Discussion (p. 25) and Supplementary Note 4 (p. 11) to note limitations of the bacterial gene dosage measurement method, limitations of studying only 11 loci and 30 species, and limitations of the data set analyzed.

18. The discussion also lacks recommendations for future analyses or ways to improve the current methodology.

We appreciate this helpful suggestion, and we have now revised the Discussion (p. 25) to describe several potential future directions, including expanding the set of bacterial reference genomes considered, using long read sequencing to precisely identify microbial structural variants, and analyzing cohorts sampled from other geographic regions.

19. The human genetic associations with microbial gene dosages are not replicated in other cohorts with different genetic backgrounds or environmental exposures. This limits the generalizability of the findings.

We agree and have now performed replication in the NIH *All of Us* (AoU) data set, which recently released saliva-derived WGS data (in AoU v8). Specifically, we analyzed a randomly selected set of 10,000 saliva WGS samples (to manage computational costs on the AoU platform) using the same pipeline to align unmapped reads to the same reference panel of 30 microbial genomes. We then attempted to replicate the 208 associations that we identified in SPARK between human genetic variants and microbial gene dosages.

The 208 associations strongly replicated in AoU, with broadly concordant effect sizes and concordant effect directions for 202 of the 208 tests (Extended Data Fig. 9c, copied below):

These results support the generalizability of the findings, and we have incorporated them into the revised manuscript (Results, p. 17; Extended Data Fig. 9c). In particular, the fact that the *All of Us* cohort is composed entirely of adults whereas the SPARK cohort is half children provides evidence of the robustness of these effects across cohorts with different environmental exposures.

On the AlphaFold analysis:

20. For the predicted models, I would expect confidence measures of the individual protein structures to be provided (e.g., average and per-residue pLDDT scores), especially for regions involved in protein-protein interactions between human AMY1 and *S. parasanguinis* AbpA and AbpB.

We have included average and per-residue pLDDT scores in a new Supplementary Table 9.

21. For the complex and protein-protein interaction model predictions, TM scores and

interface TM (ipTM) scores should be provided in the supplementary material to assess the reliability of the predictions.

We have now included pTM and ipTM scores in Supplementary Table 9 as well. (Because we had not saved these metrics from our previous AlphaFold2 runs, we reran the structure predictions using AlphaFold3 and updated the structures presented in Figures 4 and 6.) The lower TM and ipTM scores for VadD and the lower TM score for CrpE are probably due in part to the large size of these proteins (PMID: 39990437); these structures are included in Fig. 6 only for illustration (whereas the higher-confidence structures in Fig. 4 for AbpA+AMY1 and AbpB+AMY1 provide evidence of interactions between human amylase and bacterial Abp).

22. In the methods section ("AlphaFold2 prediction of protein structures"), please include the version of ChimeraX used to visualize the structures.

We have added the version of ChimeraX (v1.9) used to visualize the structures (Methods, p. 44).

Minor comments:

• What is the difference between Supplementary Tables 1 and 2? Please add species names to the strain names in ST1. Also, ST4 appears to be a subset of ST3 and could be merged.

We have added species names to the strain names in all tables. ST2 is a subset of ST1 and ST4 is a subset of ST3, where ST2 and ST4 are species or pathways with $FDR < 0.05$ and are provided for ease of lookup for the reader. ST1 and ST3 are included for completeness of presentation.

• What is the purpose of scaling and centering after inverse rank transformation? This transformation already produces distributions close to $N(0,1)$.

This is a good question. The reason is that large fractions of samples with zero abundance sometimes led to inverse rank transformed distributions that deviated from mean 0 and variance 1 (due to the large fraction of ties). We have clarified this in Methods (p. 35).

• Table 1: Add rsIDs, MAFs, and LD between SNPs if they differ between traits. Consider moving the table to the supplement.

We have added rsIDs, MAFs, and LD info to Table 1 as suggested. We have retained the table in the main manuscript for now to facilitate review but anticipate moving it to the supplement to satisfy length limitations.

- **"Tiled across 30 bacterial reference genomes" – why only 30? After reading the methods, I see this is explained, but please include a brief description or reference in the results section.**

We appreciate the suggestion and have added a brief explanation in Results (p. 17).

- **"A recent oral microbiome GWAS9 reported two associations in the centromeric region" – centromeric region of which chromosome?**

We have clarified that these associations were in the centromeric regions of chromosomes 17 and 22 (Discussion, p. 25).

- **The introduction would benefit from a paragraph summarizing what was done in this study to orient the reader.**

We have added a brief paragraph at the end of the Introduction (p. 2) summarizing the analyses.

- **Figure 1 is labeled "Figure 1," but others are not. Please remove or standardize for consistency.**

We have removed this label from Figure 1.

- **In Supplementary Note 4, four steps of transformations/adjustments are described. It would be useful to reference "Supplementary Note 4" in the main methods section ("Measuring bacterial gene dosages using read-depth phenotypes").**

We agree and have added a reference to Supplementary Note 4 in this section of Methods (p. 43).

- **Mendelian Randomization (MR): While I acknowledge that most MR analyses related to the microbiome produce low-confidence results due to moderate genetic effects and little overlap in associated signals, in this case, MR on dental phenotypes and the oral microbiome across all associated loci would still be relevant.**

We agree that it would be interesting to use Mendelian randomization to assess whether human genetic effects on oral microbial phenotypes causally influence dental phenotypes. This was actually one of our initial motivations for this work, but we ultimately concluded that we were underpowered to pursue MR analyses. The reason is that although we were able to identify 11 human genetic variants that associate with oral microbiome composition (via our microbial PC-based approach), we could not use these variants together to perform MR (because in aggregating association signal across PCs by summing chi-square test statistics, information about effect directions is lost, and MR requires signed betas). As such, MR would need to be performed using GWAS on individual taxa. However, GWAS power was weaker for individual taxa, such that no taxon associated with more than a few human loci. We anticipate that future studies using cohorts larger than SPARK will enable MR, and we have noted this in Discussion (p. 25).

Referee #2:

Kamitaki et al. examine associations between oral microbiome composition and host genetics in a well-powered human cohort (n>10,000 subjects). The authors discover novel genetic mechanisms of microbiome modulation that they establish are linked with oral health. The authors' validations of their approaches are meticulous (e.g. oral expression of identified genes, concordant sibling AMY1 copy numbers, comparing adult vs. child AMY1 microbiome associations to determine likelihood of causality given the link between AMY1 and dentures use) and their methods are sophisticated and well-considered (e.g. adjusting for sequencing batches, host genetic PCs, performing a combined chi-square test to reduce multiple comparisons). Overall, this is a beautifully performed study that reports valuable insights into the host-microbiome relationship of the oral cavity and puts forth a strong analytical framework that can serve as a model for future studies.

We appreciate these positive comments and the helpful suggestions below, which we believe have improved the manuscript.

My comments/questions:

ASD has been reported to associate with differences in the oral microbiome within the same SPARK cohort (Manghi et al. Nature Microbiology 2024). How do the authors reconcile the findings of Manghi et al. with their assertion that ASD did not exert significant impact on the oral microbiota? Manghi et al. found associations also existed between oral microbiota and IQ, as well as microbiota composition and overall microbial load, the latter of which they proposed is correlated with poor dental hygiene. The authors may consider sensitivity analyses that adjust for these variables (including ASD) in order to test whether key associations remain evident or if new associations are uncovered.

This is a good point that needed clarification and follow-up. Manghi et al. (ref. 29) actually observed that oral microbiome composition was only weakly informative of ASD status ($r^2=0.01-0.02$ in their analysis), consistent with the minimal associations with microbial abundances that we observed here (median fraction of variance explained = 0.002; Fig. 1c). We have clarified this in the main text (Results, p. 5).

We have also now included a sensitivity analysis exploring the effect of including ASD status as a covariate in GWAS of taxon abundances. We observed near-identical effect sizes for the 167 variant-species associations with or without adjusting for ASD status (Extended Data Fig. 4f, copied below), confirming that ASD does not influence the

associations we identified between human genetic variants and oral microbial abundances.

More broadly, given the modest influences of common genetic variation on ASD (Grove et al. 2019, *Nat Genet*) and the modest association of ASD status with oral microbiome composition (Manghi et al.), we would not expect conditioning on ASD status to have much impact on human genetic associations with oral microbiome phenotypes.

Associations of host genomic variants with microbial genomic regions but not species relative abundance is intriguing and is concordant with a study by Nahavi et al. *Nature Medicine* 2023 (<https://doi.org/10.1038/s41591-023-02599-8>) which found that many gut microbial species encoded genomic SNPs that correlated with a host phenotype (BMI) while the relative abundance of that species in the gut microbiota did not. This may be a relevant supporting reference for the authors in their finding that genomic structure may provide an orthologous method by which to examine host-microbial relationships beyond relative abundance.

Thank you for providing this reference. We agree that this work provides helpful context supporting the idea that microbial genomic variation may vary even in scenarios where the relative abundance of a species does not, and we have now cited it in Discussion (p. 24).

Is it the case, as for *Prevotella* spp. and ABO*A1 genotype, that *R. mucilaginosa* relative abundance also did not associate with ABO genotype? The phenomenon of finding genomic structural associations in taxa for which no relative abundance association was evident is also relevant for how the authors selected host gene variants and microbial genomes to query in the genomic structural analysis section. These were selected by which genes and microbes exhibited significant associations comparing to microbial relative abundances. While this approach that the authors took was well-reasoned to minimize the number of statistical comparisons, an exploratory analysis expanded to all genes and all microbial taxa may well yield additional novel associations. Could the authors perform this, or a summed chi-square approach as was used for mPCs? The authors may acknowledge in their text that more microbial genomic structural variations can be linked with other untested gene variants even in the same dataset. The present study will likely set precedent for such analyses, and so it may be helpful to underline this concept so that future researchers do not miss informative associations.

Yes, just as with *Prevotella* spp. and ABO, the relative abundance of *Rothia mucilaginosa* did not associate with ABO genotype ($p=0.95$), representing another example of a bacterial gene dosage association in the absence of a relative abundance association.

It is a good point that these observations raise the question of whether limiting the association analyses of microbial gene dosages to 11 human genetic variants significantly associated with microbial relative abundances—rather than performing GWAS on microbial gene dosage phenotypes—might have resulted in our missing associations involving other human genetic loci. Similarly, restricting these analyses to 30 microbes, most of which were selected based on associations of relative abundances with the 11 human genetics variants, could have limited our ability to discover associations.

We agree that exploratory analyses assessing the impact of these choices are of interest. As such, we have now undertaken two sets of exploratory GWAS to assess the extent to which expanding the analyses to consider all common human genetic variants (rather than the 11 prioritized variants) might uncover more associations with bacterial gene dosages (Methods, p. 45):

- (i) five GWAS searching for additional human genetic effects on the five bacterial gene dosage phenotypes that associated most strongly with the 11 lead variants; and
- (ii) 3,441 GWAS searching for additional human genetic effects on 3,441 read-depth phenotypes in 500bp bins spanning the genome of the most prevalent species observed in SPARK (*Rothia mucilaginosa*).

Somewhat surprisingly, neither of these analyses identified strong associations beyond the main effects we observed at *FUT2* and *ABO* (Extended Data Fig. 10c-h, copied below).

(i) GWAS of five bacterial gene dosage phenotypes with top human genetic associations:

(ii) GWAS of 3,441 read-depth bins for *R. mucilaginosa* (top p-value per human variant):

We were a bit surprised by these results, as the strengths of the signals at *FUT2* and *ABO* had led us to envision that these bacterial gene dosage phenotypes might also associate with human genetic variants at several other untested loci. These results, which we now discuss in Discussion (p. 25), suggest that limiting the tests to the 11 variants was a reasonable trade-off to enable detecting effects on more species and gene dosages.

We also agree that expanding the analyses to include more microbial taxa would be very interesting; however, this is quite challenging because identifying high-quality reference genomes for bacterial species requires manual curation and often is not even possible. For example, many of the species genomic bins (SGB) quantified by the MetaPhlan pipeline are derived from metagenomic bins and lack reference genomes. We have now noted these limitations and acknowledged in the manuscript text (as suggested) that expanding the approach to consider more microbial genomic structural variations and more human genetic variants may well uncover more effects, even in the same SPARK data set (Discussion, p. 25).

The authors seem to present the idea that *AMY1* copy number variants arose due to the selective pressure of oral health as being exclusive of *AMY1* copy number variants having arisen due to development of agricultural mass production of starch-laden foods in the sentence ‘*AMY1* copy number may instead have been under selection due to effects on oral health’). Is it not likely that both acted as selective pressures over the same timescale in tandem – that is, that oral health was an additional, as-yet unappreciated selective pressure? One can imagine that better digestion of starches and protection from fatal dental infections that arise from these new food sources shaped this locus simultaneously. Would the authors agree with this, and consider revising the sentence to be inclusive of such a possibility? Also, it is unclear to me why a lack of association with BMI is evidence against agriculture being a selective pressure that shaped *AMY1* copy number; *AMY1*-mediated breakdown of starches may contribute to dietary energy extraction but not obesity which may instead be caused by dietary intake of a complex combination of simple sugars, ultra-processed foods, and other dietary/lifestyle factors for which detrimental effects are highly personalized.

We appreciate this line of reasoning and agree that these selective pressures could have shaped the locus simultaneously, so as suggested, we have revised the indicated sentence in Discussion (p. 24) to be inclusive of this possibility.

We have also revised this sentence to no longer mention the lack of association with BMI. We agree that in principle, *AMY1* copy number could have been shaped by selection due to agriculture but have no effect on BMI variation in present-day humans, which may be driven more by consumption of simple carbohydrates and by lifestyle factors.

Referee #1:

In general, the authors did a comprehensive job in replying to all our comments and replied all important points. Two comments remain:

c. "The idea behind the approach is conceptually similar to a GWAS on beta diversity – capturing the effects of SNPs on overall composition."

Here the question was not about the tool but about the approach – how does your PC-based analysis (essentially MMR/MANOVA) compare with Multivariate Distance Matrix Regression (MDMR; Anderson, 2001; McArdle & Anderson, 2001)? We expected to see this comparison, which could be done, for example, using the available R package (<https://cran.r-project.org/web/packages/MDMR/> – note, we do not have experience running it), or by using other microbiome GWAS tools. It would be valuable to see benchmark results, e.g. for 1,000 samples and 1,000 SNPs, as a demonstration of validity of your approach.

We agree that this benchmark is valuable, and we appreciate the suggestion of comparing our approach to MDMR. We have now performed this benchmark, running the MDMR R package on a distance matrix generated from the same input that we used to produce PCs (i.e., relative abundances of the 439 most prevalent species across the entire SPARK cohort). We computed association tests for the 11 lead variants identified by our approach ($p = 3.0 \times 10^{-11}$ to 1.6×10^{-131}) and 1,000 randomly selected variants. The MDMR approach appeared to be considerably less powerful ($p = 0.062$ to 3.9×10^{-13} for the 11 lead variants) :

We also benchmarked MDMR using a distance matrix that we generated after applying the centered log-ratio transform to relative abundances to account for compositionality. Again, the MDMR approach appeared to be considerably less powerful, with the CLR transform having little effect on the results, consistent with our other benchmarks (Extended Data Fig. 4e):

We have included these plots as Extended Data Fig. 2d,e, and we have added a description of the analyses in Results and Methods.

f. "In general, if the effect of certain genetic loci is strong, it should be captured in a limited number of PCs."

Looking at the figures, however, we think that most of the time the signal is instead distributed across a large number of PCs (except in the example you mention). The biological relevance of this observation remains unclear to us, especially considering that these PCs do not reflect different bacteria and are orthogonal. We are interested in how the authors explain this phenomenon.

We agree that for most human genetic loci, the signal is quite diffusely spread across many PCs rather than strongly captured by a few. We suspect this indicates that these human genetic variants have subtle influences on many bacterial species, resulting in modest associations with many axes of variation in microbiome composition. We have added this suggested explanation to Results.

Referee #2 (Remarks to the Author):

It is a pity that additional reference genomes cannot be used to query associations between other oral bacteria and human genetic variants, but it is appreciated that the authors remarked on how that may be overcome in future studies in their Discussion (via long-read sequencing). Perhaps this study will stimulate greater interest in doing so. The authors have otherwise addressed my comments.

We appreciate your comments and likewise hope that more complete collections of reference genomes will become available in the near future to facilitate identification of additional host-microbe genetic interactions.